# Linear Last-iterate Convergence in Constrained Saddle-point Optimization

**Chen-Yu Wei, Chung-Wei Lee, Mengxiao Zhang, Haipeng Luo**
University of Southern California
{chenyu.wei,leechung,mengxiao.zhang,haipengl}@usc.edu

## Abstract

Optimistic Gradient Descent Ascent (OGDA) and Optimistic Multiplicative Weights Update (OMWU) for saddle-point optimization have received growing attention due to their favorable last-iterate convergence. However, their behaviors for simple bilinear games over the probability simplex are still not fully understood — previous analysis lacks explicit convergence rates, only applies to an exponentially small learning rate, or requires additional assumptions such as the uniqueness of the optimal solution.

In this work, we significantly expand the understanding of last-iterate convergence for OGDA and OMWU in the constrained setting. Specifically, for OMWU in bilinear games over the simplex, we show that when the equilibrium is unique, *linear* last-iterate convergence is achieved with a learning rate whose value is set to a *universal constant*, improving the result of (Daskalakis & Panageas, 2019b) under the same assumption. We then significantly extend the results to more general objectives and feasible sets for the projected OGDA algorithm, by introducing a sufficient condition under which OGDA exhibits concrete last-iterate convergence rates with a constant learning rate whose value only depends on the smoothness of the objective function. We show that bilinear games over *any polytope* satisfy this condition and OGDA converges exponentially fast even *without the unique equilibrium assumption*. Our condition also holds for strongly-convex-strongly-concave functions, recovering the result of (Hsieh et al., 2019). Finally, we provide experimental results to further support our theory.

## 1 Introduction

Saddle-point optimization in the form of $\min_{\boldsymbol{x}} \max_{\boldsymbol{y}} f(\boldsymbol{x}, \boldsymbol{y})$ dates back to (Neumann, 1928), where the celebrated minimax theorem was discovered. Due to advances of Generative Adversarial Networks (GANs) (Goodfellow et al., 2014) (which itself is a saddle-point problem), the question of how to find a good approximation of the saddle point, especially via an efficient iterative algorithm, has recently gained significant research interest. Simple algorithms such as Gradient Descent Ascent (GDA) and Multiplicative Weights Update (MWU) are known to cycle and fail to converge even in simple bilinear cases (see e.g., (Bailey & Piliouras, 2018) and (Cheung & Piliouras, 2019)).

Many recent works consider resolving this issue via simple modifications of standard algorithms, usually in the form of some extra gradient descent/ascent steps. This includes Extra-Gradient methods (EG) (Liang & Stokes, 2019; Mokhtari et al., 2020b), Optimistic Gradient Descent Ascent (OGDA) (Daskalakis et al., 2018; Gidel et al., 2019; Mertikopoulos et al., 2019), Optimistic Multiplicative Weights Update (OMWU) (Daskalakis & Panageas, 2019b; Lei et al., 2021), and others. In particular, OGDA and OMWU are suitable for the repeated game setting where two players repeatedly propose $\boldsymbol{x}_t$ and $\boldsymbol{y}_t$ and receive only $\nabla_{\boldsymbol{x}} f(\boldsymbol{x}_t, \boldsymbol{y}_t)$ and $\nabla_{\boldsymbol{y}} f(\boldsymbol{x}_t, \boldsymbol{y}_t)$ respectively as feedback, with the goal of converging to a saddle point or equivalently a Nash equilibrium using game theory terminology. One notable benefit of OGDA and OMWU is that they are also *no-regret* algorithms with important applications in online learning, especially when playing against adversarial opponents (Chiang et al., 2012; Rakhlin & Sridharan, 2013).

Despite considerable progress, especially those for the unconstrained setting, the behavior of these algorithms for the constrained setting, where $\boldsymbol{x}$ and $\boldsymbol{y}$ are restricted to closed convex sets $\mathcal{X}$ and

$\mathcal{Y}$ respectively, is still not fully understood. This is even true when $f$ is a bilinear function and $\mathcal{X}$ and $\mathcal{Y}$ are simplex, known as the classic two-player zero-sum games in normal form, or simply *matrix games*. Indeed, existing convergence results on the last iterate of OGDA or OMWU for matrix games are unsatisfactory — they lack explicit convergence rates (Popov, 1980; Mertikopoulos et al., 2019), only apply to exponentially small learning rate thus not reflecting the behavior of the algorithms in practice (Daskalakis & Panageas, 2019b), or require additional conditions such as uniqueness of the equilibrium or a good initialization (Daskalakis & Panageas, 2019b).

Motivated by this fact, in this work, we first improve the last-iterate convergence result of OMWU for matrix games. Under the same unique equilibrium assumption as made by Daskalakis & Panageas (2019b), we show *linear* convergence with a concrete rate in terms of the Kullback-Leibler divergence between the last iterate and the equilibrium, using a learning rate whose value is set to a *universal constant*.

We then significantly extend our results and consider OGDA for general constrained and smooth convex-concave saddle-point problems, without the uniqueness assumption. Specifically, we start with proving an average duality gap convergence of OGDA at the rate of $\mathcal{O}(1/\sqrt{T})$ after $T$ iterations. Then, to obtain a more favorable last-iterate convergence in terms of the distance to the set of equilibria, we propose a general sufficient condition on $\mathcal{X}, \mathcal{Y}$, and $f$, called *Saddle-Point Metric Subregularity* (SP-MS), under which we prove concrete last-iterate convergence rates, all with a constant learning rate and without further assumptions.

Our last-iterate convergence results of OGDA greatly generalize that of (Hsieh et al., 2019, Theorem 2), which itself is a consolidated version of results from several earlier works. The key implication of our new results is that, by showing that matrix games satisfy our SP-MS condition, we provide by far the most general last-iterate guarantee with a linear convergence for this problem using OGDA. Compared to that of OMWU, the convergence result of OGDA holds more generally even when there are multiple equilibria.

More generally, the same linear last-iterate convergence holds for any bilinear games over polytopes since they also satisfy the SP-MS condition as we show. To complement this result, we construct an example of a bilinear game with a non-polytope feasible set where OGDA provably does not ensure linear convergence, indicating that *the shape of the feasible set matters*.

Finally, we also provide experimental results to support our theory. In particular, we observe that OGDA generally converges faster than OMWU for matrix games, despite the facts that both provably converge exponentially fast and that OMWU is often considered more favorable compared to OGDA when the feasible set is the simplex.

## 2 RELATED WORK

**Average-iterate convergence.** While showing last-iterate convergence has been a challenging task, it is well-known that the average-iterate of many standard algorithms such as GDA and MWU enjoys a converging duality gap at the rate of $O(1/\sqrt{T})$ (Freund & Schapire, 1999). A line of works show that the rate can be improved to $O(1/T)$ using the "optimistic" version of these algorithms such as OGDA and OMWU (Rakhlin & Sridharan, 2013; Daskalakis et al., 2015; Syrgkanis et al., 2015). For tasks such as training GANs, however, average-iterate convergence is unsatisfactory since averaging large neural networks is usually prohibited.

**Extra-Gradient (EG) algorithms.** The saddle-point problem fits into the more general variational inequality framework (Harker & Pang, 1990). A classic algorithm for variational inequalities is EG, first introduced in (Korpelevich, 1976). Tseng (1995) is the first to show last-iterate convergence for EG in various settings such as bilinear or strongly-convex-strongly-concave problems. Recent works significantly expand the understanding of EG and its variants for unconstrained bilinear problems (Liang & Stokes, 2019), unconstrained strongly-convex-strongly-concave problems (Mokhtari et al., 2020b), and more (Zhang et al., 2019; Lin et al., 2020; Golowich et al., 2020b).

The original EG is not applicable to a repeated game setting where only one gradient evaluation is possible in each iteration. Moreover, unlike OGDA and OMWU, EG is shown to have linear regret against adversarial opponents, and thus it is not a no-regret learning algorithm (Bowling, 2005;

Golowich et al., 2020a). However, there are "single-call variants" of EG that address these issues. In fact, some of these versions coincide with the OGDA algorithm under different names such as modified Arrow–Hurwicz method (Popov, 1980) and "extrapolation from the past" (Gidel et al., 2019). Apart from OGDA, other single-call variants of EG include Reflected Gradient (Malitsky, 2015; Cui & Shanbhag, 2016; Malitsky & Tam) and Optimistic Gradient (Daskalakis et al., 2018; Mokhtari et al., 2020a). These variants are all equivalent in the unconstrained setting but differ in the constrained setting. To the best of our knowledge, none of the existing results for any single-call variant of EG covers the constrained bilinear case (which is one of our key contributions).

**Error Bounds and Metric Subregularity**    To derive linear convergence for variational inequality problems, *error bound* method is a commonly used technique (Pang, 1997; Luo & Tseng, 1993). For example, it is a standard approach to studying the last-iterate convergence of EG algorithms (Tseng, 1995; Hsieh et al., 2020; Azizian et al., 2020). An error bound method is associated with an error function that gives every point in the feasible set a measure of sub-optimality that is lower bounded by the distance of the point to the optimal set up to some problem dependent constant. If such a error function exists, linear convergence can be obtained. The choice of the error function depends on the feasible region, the objection function, and the algorithm. Common error functions include natural residual functions (Iusem et al., 2017; Malitsky, 2019) and gap functions (Larsson & Patriksson, 1994; Solodov & Tseng, 2000; Chen et al., 2017). Our method to derive the last-iterate convergence for OGDA can also be viewed as an error bound method.

*Metric subregularity* is another important concept to derive linear convergence via some Lipschitz behavior of a set-valued operator (Leventhal, 2009; Liang et al., 2016; Alacaoglu et al., 2019; Latafat et al., 2019). Metric subregularity is closely related to error bound methods (Kruger, 2015). In fact, as we prove in Appendix F, one special case of our condition SP-MS (that allows us to show linear convergence) is *equivalent* to metric subregularity of an operator defined in terms of the normal cone of the feasible set and the gradient of the objective. This is also the reason why we call our condition Saddle-Point Metric Subregularity. Although metric subregularity has been extensively used in the literature, to the best of our knowledge, our work is the first to use this condition to analyze OGDA.

**OGDA and OMWU.**    Recently, last-iterate convergence for OGDA has been proven in various settings such as convex-concave problems (Daskalakis et al., 2018), unconstrained bilinear problems (Daskalakis & Panageas, 2018; Liang & Stokes, 2019), strongly-convex-strongly-concave problems (Mokhtari et al., 2020b), and others (e.g. (Mertikopoulos et al., 2019)).

However, the behavior of OGDA and OMWU for the constrained bilinear case, or even the special case of classic matrix games, appears to be much more mysterious and less understood. Cheung & Piliouras (2020) provide an alternative view on the convergence behavior of OMWU by studying volume contraction in the dual space. Daskalakis & Panageas (2019b) show last-iterate convergence of OMWU for matrix games under a uniqueness assumption and without a concrete rate. Although it is implicitly suggested in (Daskalakis & Panageas, 2019b;a) that a rate of $O(1/T^{1/9})$ is possible, it is still not clear how to choose the learning rate appropriately from their analysis. As mentioned, our results for OMWU significantly improve theirs, with a clean linear convergence rate using a constant learning rate under the same uniqueness assumption, while our results for OGDA further remove the uniqueness assumption.

## 3    Notations and Preliminaries

We consider the following constrained saddle-point problem: $\min_{\boldsymbol{x} \in \mathcal{X}} \max_{\boldsymbol{y} \in \mathcal{Y}} f(\boldsymbol{x}, \boldsymbol{y})$, where $\mathcal{X}$ and $\mathcal{Y}$ are closed convex sets, and $f$ is a continuous differentiable function that is convex in $\boldsymbol{x}$ for any fixed $\boldsymbol{y}$ and concave in $\boldsymbol{y}$ for any fixed $\boldsymbol{x}$. By the celebrated minimax theorem (Neumann, 1928), we have $\min_{\boldsymbol{x} \in \mathcal{X}} \max_{\boldsymbol{y} \in \mathcal{Y}} f(\boldsymbol{x}, \boldsymbol{y}) = \max_{\boldsymbol{y} \in \mathcal{Y}} \min_{\boldsymbol{x} \in \mathcal{X}} f(\boldsymbol{x}, \boldsymbol{y})$.

The set of minimax optimal strategy is denoted by $\mathcal{X}^* = \operatorname{argmin}_{\boldsymbol{x} \in \mathcal{X}} \max_{\boldsymbol{y} \in \mathcal{Y}} f(\boldsymbol{x}, \boldsymbol{y})$, and the set of maximin optimal strategy is denoted by $\mathcal{Y}^* = \operatorname{argmax}_{\boldsymbol{y} \in \mathcal{Y}} \min_{\boldsymbol{x} \in \mathcal{X}} f(\boldsymbol{x}, \boldsymbol{y})$. It is well-known that $\mathcal{X}^*$ and $\mathcal{Y}^*$ are convex, and any pair $(\boldsymbol{x}^*, \boldsymbol{y}^*) \in \mathcal{X}^* \times \mathcal{Y}^*$ is a Nash equilibrium satisfying $f(\boldsymbol{x}^*, \boldsymbol{y}) \leq f(\boldsymbol{x}^*, \boldsymbol{y}^*) \leq f(\boldsymbol{x}, \boldsymbol{y}^*)$ for any $(\boldsymbol{x}, \boldsymbol{y}) \in \mathcal{X} \times \mathcal{Y}$.

For notational convenience, we define $\mathcal{Z} = \mathcal{X} \times \mathcal{Y}$ and similarly $\mathcal{Z}^* = \mathcal{X}^* \times \mathcal{Y}^*$. For a point $\boldsymbol{z} = (\boldsymbol{x}, \boldsymbol{y}) \in \mathcal{Z}$, we further define $f(\boldsymbol{z}) = f(\boldsymbol{x}, \boldsymbol{y})$ and $F(\boldsymbol{z}) = (\nabla_{\boldsymbol{x}} f(\boldsymbol{x}, \boldsymbol{y}), -\nabla_{\boldsymbol{y}} f(\boldsymbol{x}, \boldsymbol{y}))$.

Our goal is to find a point $z \in \mathcal{Z}$ that is close to the set of Nash equilibria $\mathcal{Z}^*$, and we consider three ways of measuring the closeness. The first one is the *duality gap*, defined as $\alpha_f(z) = \max_{y' \in \mathcal{Y}} f(x, y') - \min_{x' \in \mathcal{X}} f(x', y)$, which is always non-negative since $\max_{y' \in \mathcal{Y}} f(x, y') \geq f(x, y) \geq \min_{x' \in \mathcal{X}} f(x', y)$.

The second one is the distance between $z$ and $\mathcal{Z}^*$. Specifically, for any closed set $\mathcal{A}$, we define the projection operator $\Pi_{\mathcal{A}}$ as $\Pi_{\mathcal{A}}(a) = \operatorname{argmin}_{a' \in \mathcal{A}} \|a - a'\|$ (throughout this work $\|\cdot\|$ represents $L_2$ norm). The squared distance between $z$ and $\mathcal{Z}^*$ is then defined as $\operatorname{dist}^2(z, \mathcal{Z}^*) = \|z - \Pi_{\mathcal{Z}^*}(z)\|^2$.

The third one is only for the case when $\mathcal{X}$ and $\mathcal{Y}$ are probability simplices, and $z^* = (x^*, y^*)$ is the unique equilibrium. In this case, we use the sum of Kullback-Leibler divergence $\operatorname{KL}(x^*, x) + \operatorname{KL}(y^*, y)$ to measure the closeness between $z = (x, y)$ and $z^*$, where $\operatorname{KL}(x, x') = \sum_i x_i \ln \frac{x_i}{x'_i}$. With a slight abuse of notation, we use $\operatorname{KL}(z, z')$ to denote $\operatorname{KL}(x, x') + \operatorname{KL}(y, y')$.

**Other notations.** We denote the $(d-1)$-dimensional probability simplex as $\Delta_d = \{u \in \mathbb{R}^d_+ : \sum_{i=1}^d u_i = 1\}$. For a convex function $\psi$, the corresponding *Bregman divergence* is defined as $D_\psi(u, v) = \psi(u) - \psi(v) - \langle \nabla\psi(v), u - v \rangle$. If $\psi$ is $\gamma$-strongly convex in a domain, then $D_\psi(u, v) \geq \frac{\gamma}{2} \|u - v\|^2$ for any $u, v$ in that domain. For $u \in \mathbb{R}^d$, we define $\operatorname{supp}(u) = \{i : u_i > 0\}$.

**Optimistic Gradient Descent Ascent (OGDA).** Starting from an arbitrary point $(\widehat{x}_1, \widehat{y}_1) = (x_0, y_0)$ from $\mathcal{Z}$, OGDA with step size $\eta > 0$ iteratively computes the following for $t = 1, 2, \ldots$,

$$x_t = \Pi_{\mathcal{X}}(\widehat{x}_t - \eta\nabla_x f(x_{t-1}, y_{t-1})), \qquad \widehat{x}_{t+1} = \Pi_{\mathcal{X}}(\widehat{x}_t - \eta\nabla_x f(x_t, y_t)),$$
$$y_t = \Pi_{\mathcal{Y}}(\widehat{y}_t + \eta\nabla_y f(x_{t-1}, y_{t-1})), \qquad \widehat{y}_{t+1} = \Pi_{\mathcal{Y}}(\widehat{y}_t + \eta\nabla_y f(x_t, y_t)).$$

Note that there are several slightly different versions of the algorithm in the literature, which differ in the timing of performing the projection. Our version is the same as those in (Chiang et al., 2012; Rakhlin & Sridharan, 2013). It is also referred to as "single-call extra-gradient" in (Hsieh et al., 2019), but it does not belong to the class of "extra-gradient" methods discussed in (Tseng, 1995; Liang & Stokes, 2019; Golowich et al., 2020b) for example.

Also note that OGDA only requires accessing $f$ via its gradient. In fact, only one gradient at the point $(x_t, y_t)$ is needed for iteration $t$. This aspect makes it especially suitable for a repeated game setting, where in each round, one player proposes $x_t$ while another player proposes $y_t$. With only the information of the gradient from the environment ($\nabla_x f(x_t, y_t)$ for the first player and $\nabla_y f(x_t, y_t)$ for the other), both players can execute the algorithm.

**Optimistic Multiplicative Weights Update (OMWU).** When the feasible sets $\mathcal{X}$ and $\mathcal{Y}$ are probability simplices $\Delta_M$ and $\Delta_N$ for some integers $M$ and $N$, OMWU is another common iterative algorithm to solve the saddle-point problem. For simplicity, we assume that it starts from the uniform distributions $(\widehat{x}_1, \widehat{y}_1) = (x_0, y_0) = \left(\frac{1_M}{M}, \frac{1_N}{N}\right)$, where $1_d$ is the all-one vector of dimension $d$. Then OMWU with step size $\eta > 0$ iteratively computes the following for $t = 1, 2, \ldots$,

$$x_{t,i} = \frac{\widehat{x}_{t,i} \exp(-\eta(\nabla_x f(x_{t-1}, y_{t-1}))_i)}{\sum_j \widehat{x}_{t,j} \exp(-\eta(\nabla_x f(x_{t-1}, y_{t-1}))_j)}, \quad \widehat{x}_{t+1,i} = \frac{\widehat{x}_{t,i} \exp(-\eta(\nabla_x f(x_t, y_t))_i)}{\sum_j \widehat{x}_{t,j} \exp(-\eta(\nabla_x f(x_t, y_t))_j)},$$
$$y_{t,i} = \frac{\widehat{y}_{t,i} \exp(\eta(\nabla_y f(x_{t-1}, y_{t-1}))_i)}{\sum_j \widehat{y}_{t,j} \exp(\eta(\nabla_y f(x_{t-1}, y_{t-1}))_j)}, \quad \widehat{y}_{t+1,i} = \frac{\widehat{y}_{t,i} \exp(\eta(\nabla_y f(x_t, y_t))_i)}{\sum_j \widehat{y}_{t,j} \exp(\eta(\nabla_y f(x_t, y_t))_j)}.$$

**OMWU and OGDA as Optimistic Mirror Descent Ascent.** OMWU and OGDA can be viewed as special cases of *Optimistic Mirror Descent Ascent*. Specifically, let *regularizer* $\psi(u)$ denote the negative entropy $\sum_i u_i \ln u_i$ for the case of OMWU and (half of) the $L_2$ norm square $\frac{1}{2}\|u\|^2$ for the case of OGDA (so that $D_\psi(u, v)$ is $\operatorname{KL}(u, v)$ and $\frac{1}{2}\|u - v\|^2$ respectively). Then using the shorthands $z_t = (x_t, y_t)$ and $\widehat{z}_t = (\widehat{x}_t, \widehat{y}_t)$ and recalling the notation defined earlier: $\mathcal{Z} = \mathcal{X} \times \mathcal{Y}$ and $F(z) = (\nabla_x f(x, y), -\nabla_y f(x, y))$, one can rewrite OMWU/OGDA compactly as

$$z_t = \operatorname*{argmin}_{z \in \mathcal{Z}} \left\{ \eta\langle z, F(z_{t-1}) \rangle + D_\psi(z, \widehat{z}_t) \right\}, \tag{1}$$

$$\widehat{z}_{t+1} = \operatorname*{argmin}_{z \in \mathcal{Z}} \left\{ \eta\langle z, F(z_t) \rangle + D_\psi(z, \widehat{z}_t) \right\}. \tag{2}$$

By the standard regret analysis of Optimistic Mirror Descent, we have the following important lemma, which is readily applied to OMWU and OGDA when $\psi$ is instantiated as the corresponding regularizer. The proof is mostly standard (see e.g., (Rakhlin & Sridharan, 2013, Lemma 1)). For completeness, we include it in Appendix B.

**Lemma 1.** *Consider update rules Eq. (1) and Eq. (2) and define $\mathrm{dist}_p^2(\boldsymbol{z}, \boldsymbol{z}') = \|\boldsymbol{x} - \boldsymbol{x}'\|_p^2 + \|\boldsymbol{y} - \boldsymbol{y}'\|_p^2$. Suppose that $\psi$ satisfies $D_\psi(\boldsymbol{z}, \boldsymbol{z}') \geq \frac{1}{2}\mathrm{dist}_p^2(\boldsymbol{z}, \boldsymbol{z}')$ for some $p \geq 1$, and $F$ satisfies $\mathrm{dist}_q^2(F(\boldsymbol{z}), F(\boldsymbol{z}')) \leq L^2 \mathrm{dist}_p^2(\boldsymbol{z}, \boldsymbol{z}')$ for $q \geq 1$ with $\frac{1}{p} + \frac{1}{q} = 1$. Also, assume that $\eta \leq \frac{1}{8L}$. Then for any $\boldsymbol{z} \in \mathcal{Z}$ and any $t \geq 1$, we have*

$$\eta F(\boldsymbol{z}_t)^\top (\boldsymbol{z}_t - \boldsymbol{z}) \leq D_\psi(\boldsymbol{z}, \widehat{\boldsymbol{z}}_t) - D_\psi(\boldsymbol{z}, \widehat{\boldsymbol{z}}_{t+1}) - D_\psi(\widehat{\boldsymbol{z}}_{t+1}, \boldsymbol{z}_t) - \tfrac{15}{16}D_\psi(\boldsymbol{z}_t, \widehat{\boldsymbol{z}}_t) + \tfrac{1}{16}D_\psi(\widehat{\boldsymbol{z}}_t, \boldsymbol{z}_{t-1}).$$

## 4 CONVERGENCE RESULTS FOR OMWU

In this section, we show that for a two-player zero-sum matrix game with a unique equilibrium, OMWU with a constant learning rate converges to the equilibrium exponentially fast. The assumption and the algorithm are the same as those considered in (Daskalakis & Panageas, 2019b), but our analysis improves theirs in two ways. First, we do not require the learning rate to be exponentially smaller than some problem-dependent quantity. Second, we explicitly provide a linear convergence rate. In Section 5, we further remove the uniqueness assumption and significantly generalize the results by studying OGDA.

In a matrix game we have $\mathcal{X} = \Delta_M$, $\mathcal{Y} = \Delta_N$, and $f(\boldsymbol{z}) = \boldsymbol{x}^\top \boldsymbol{G} \boldsymbol{y}$ for some matrix $\boldsymbol{G} \in [-1, 1]^{M \times N}$. To show the last-iterate convergence of OMWU, we first apply Lemma 1 with $D_\psi(\boldsymbol{u}, \boldsymbol{v}) = \mathrm{KL}(\boldsymbol{u}, \boldsymbol{v})$, $\boldsymbol{z} = \boldsymbol{z}^*$ (the unique equilibrium of the game matrix $\boldsymbol{G}$) and $(p, q) = (1, \infty)$. The constant $L$ can be chosen as 1 since $\mathrm{dist}_\infty^2(F(\boldsymbol{z}), F(\boldsymbol{z}')) = \max_i |(\boldsymbol{G}(\boldsymbol{y} - \boldsymbol{y}'))_i|^2 + \max_j |(\boldsymbol{G}^\top(\boldsymbol{x} - \boldsymbol{x}'))_j|^2 \leq \|\boldsymbol{y} - \boldsymbol{y}'\|_1^2 + \|\boldsymbol{x} - \boldsymbol{x}'\|_1^2 = \mathrm{dist}_1^2(\boldsymbol{z}, \boldsymbol{z}')$. Also notice that $F(\boldsymbol{z}_t)^\top (\boldsymbol{z}_t - \boldsymbol{z}^*) = f(\boldsymbol{x}_t, \boldsymbol{y}_t) - f(\boldsymbol{x}^*, \boldsymbol{y}_t) + f(\boldsymbol{x}_t, \boldsymbol{y}^*) - f(\boldsymbol{x}_t, \boldsymbol{y}_t) = f(\boldsymbol{x}_t, \boldsymbol{y}^*) - f(\boldsymbol{x}^*, \boldsymbol{y}_t) \geq 0$ by the optimality of $\boldsymbol{z}^*$. Therefore, we have when $\eta \leq \frac{1}{8}$,

$$\mathrm{KL}(\boldsymbol{z}^*, \widehat{\boldsymbol{z}}_{t+1}) \leq \mathrm{KL}(\boldsymbol{z}^*, \widehat{\boldsymbol{z}}_t) - \mathrm{KL}(\widehat{\boldsymbol{z}}_{t+1}, \boldsymbol{z}_t) - \tfrac{15}{16}\mathrm{KL}(\boldsymbol{z}_t, \widehat{\boldsymbol{z}}_t) + \tfrac{1}{16}\mathrm{KL}(\widehat{\boldsymbol{z}}_t, \boldsymbol{z}_{t-1}).$$

Defining $\Theta_t = \mathrm{KL}(\boldsymbol{z}^*, \widehat{\boldsymbol{z}}_t) + \frac{1}{16}\mathrm{KL}(\widehat{\boldsymbol{z}}_t, \boldsymbol{z}_{t-1})$ and $\zeta_t = \mathrm{KL}(\widehat{\boldsymbol{z}}_{t+1}, \boldsymbol{z}_t) + \mathrm{KL}(\boldsymbol{z}_t, \widehat{\boldsymbol{z}}_t)$, we rewrite the above as

$$\Theta_{t+1} \leq \Theta_t - \tfrac{15}{16}\zeta_t. \tag{3}$$

From Eq. (3) it is clear that the quantity $\Theta_t$ is always non-increasing in $t$ due to the non-negativity of $\zeta_t$. Furthermore, the more the algorithm moves between round $t$ and round $t+1$ (that is, the larger $\zeta_t$ is), the more $\Theta_t$ decreases.

To establish the rate of convergence, a natural idea is to relate $\zeta_t$ back to $\Theta_t$ or $\Theta_{t+1}$. For example, if we can show $\zeta_t \geq c\Theta_{t+1}$ for some constant $c > 0$, then Eq. (3) implies $\Theta_{t+1} \leq \Theta_t - \frac{15c}{16}\Theta_{t+1}$, which further gives $\Theta_{t+1} \leq \left(1 + \frac{15c}{16}\right)^{-1}\Theta_t$. This immediately implies a linear convergence rate for $\Theta_t$ as well as $\mathrm{KL}(\boldsymbol{z}^*, \widehat{\boldsymbol{z}}_t)$ since $\mathrm{KL}(\boldsymbol{z}^*, \widehat{\boldsymbol{z}}_t) \leq \Theta_t$.

Moreover, notice that to find such $c$, it suffices to find a $c' > 0$ such that $\zeta_t \geq c'\mathrm{KL}(\boldsymbol{z}^*, \widehat{\boldsymbol{z}}_{t+1})$. This is because it will then give $\zeta_t \geq \frac{1}{16}\mathrm{KL}(\widehat{\boldsymbol{z}}_{t+1}, \boldsymbol{z}_t) + \frac{15}{16}\zeta_t \geq \frac{1}{16}\mathrm{KL}(\widehat{\boldsymbol{z}}_{t+1}, \boldsymbol{z}_t) + \frac{15c'}{16}\mathrm{KL}(\boldsymbol{z}^*, \widehat{\boldsymbol{z}}_{t+1}) \geq \min\{1, \frac{15c'}{16}\}\Theta_{t+1}$, and thus $c \triangleq \min\{1, \frac{15c'}{16}\}$ satisfies the condition.

From the discussion above, we see that to establish the linear convergence of $\mathrm{KL}(\boldsymbol{z}^*, \widehat{\boldsymbol{z}}_t)$, we only need to show that there exists some $c' > 0$ such that $\mathrm{KL}(\widehat{\boldsymbol{z}}_{t+1}, \boldsymbol{z}_t) + \mathrm{KL}(\boldsymbol{z}_t, \widehat{\boldsymbol{z}}_t) \geq c'\mathrm{KL}(\boldsymbol{z}^*, \widehat{\boldsymbol{z}}_{t+1})$. The high-level interpretation of this inequality is that when $\widehat{\boldsymbol{z}}_{t+1}$ is far from the equilibrium $\boldsymbol{z}^*$ (i.e., $\mathrm{KL}(\boldsymbol{z}^*, \widehat{\boldsymbol{z}}_{t+1})$ is large), the algorithm should have a large move between round $t$ and $t+1$ making $\mathrm{KL}(\widehat{\boldsymbol{z}}_{t+1}, \boldsymbol{z}_t) + \mathrm{KL}(\boldsymbol{z}_t, \widehat{\boldsymbol{z}}_t)$ large.

In our analysis, we use a two-stage argument to find such a $c'$. In the first stage, we only show that $\mathrm{KL}(\widehat{\boldsymbol{z}}_{t+1}, \boldsymbol{z}_t) + \mathrm{KL}(\boldsymbol{z}_t, \widehat{\boldsymbol{z}}_t) \geq c''\mathrm{KL}(\boldsymbol{z}^*, \widehat{\boldsymbol{z}}_{t+1})^2$ for some $c'' > 0$, and use it to argue a slower convergence rate $\mathrm{KL}(\boldsymbol{z}^*, \widehat{\boldsymbol{z}}_t) = \mathcal{O}\left(\frac{1}{t}\right)$. Then in the second stage, we show that after $\widehat{\boldsymbol{z}}_t$ and $\boldsymbol{z}_t$ become close enough to $\boldsymbol{z}^*$, we have $\mathrm{KL}(\widehat{\boldsymbol{z}}_{t+1}, \boldsymbol{z}_t) + \mathrm{KL}(\boldsymbol{z}_t, \widehat{\boldsymbol{z}}_t) \geq c'\mathrm{KL}(\boldsymbol{z}^*, \widehat{\boldsymbol{z}}_{t+1})$ for some $c' > 0$.

This kind of two-stage argument might be reminiscent of that used by Daskalakis & Panageas (2019b); however, the techniques we use are very different. Specifically, Daskalakis & Panageas (2019b) utilize tools of "spectral analysis" similar to (Liang & Stokes, 2019) and show that the OMWU update can be viewed as a "contraction mapping" with respect to a matrix whose eigenvalue is smaller than 1. Our analysis, on the other hand, leverages analysis of online mirror descent, starting from the "one-step regret bound" (Lemma 1) and making use of the two negative terms that are typically dropped in the analysis. Importantly, our analysis does not need an exponentially small learning rate required by (Daskalakis & Panageas, 2019b). Thus, unlike their results, our learning rate is kept as a universal constant in all stages. The arguments above are formalized below:

**Lemma 2.** *Consider a matrix game* $f(\boldsymbol{x}, \boldsymbol{y}) = \boldsymbol{x}^\top \boldsymbol{G} \boldsymbol{y}$ *with* $\mathcal{X} = \Delta_M$, $\mathcal{Y} = \Delta_N$, *and* $\boldsymbol{G} \in [-1, 1]^{M \times N}$. *Assume that there exists a unique Nash equilibrium* $\boldsymbol{z}^*$ *and* $\eta \leq \frac{1}{8}$. *Then, there exists a constant* $C_1 > 0$ *that depends on* $\boldsymbol{G}$ *such that for any* $t \geq 1$, *OMWU ensures*

$$\mathrm{KL}(\widehat{\boldsymbol{z}}_{t+1}, \boldsymbol{z}_t) + \mathrm{KL}(\boldsymbol{z}_t, \widehat{\boldsymbol{z}}_t) \geq \eta^2 C_1 \mathrm{KL}(\boldsymbol{z}^*, \widehat{\boldsymbol{z}}_{t+1})^2.$$

*Also, there is a constant* $\xi > 0$ *that depends on* $\boldsymbol{G}$ *(defined in Definition 2) such that as long as* $\max\{\|\boldsymbol{z}^* - \widehat{\boldsymbol{z}}_t\|_1, \|\boldsymbol{z}^* - \boldsymbol{z}_t\|_1\} \leq \frac{\eta \xi}{10}$, *then*

$$\mathrm{KL}(\widehat{\boldsymbol{z}}_{t+1}, \boldsymbol{z}_t) + \mathrm{KL}(\boldsymbol{z}_t, \widehat{\boldsymbol{z}}_t) \geq \eta^2 C_2 \mathrm{KL}(\boldsymbol{z}^*, \widehat{\boldsymbol{z}}_{t+1})$$

*for another constant* $C_2 > 0$ *that depends on* $\boldsymbol{G}$.

With Lemma 2 and the earlier discussion, the last-iterate convergence rate of OMWU is established:

**Theorem 3.** *For a matrix game* $f(\boldsymbol{x}, \boldsymbol{y}) = \boldsymbol{x}^\top \boldsymbol{G} \boldsymbol{y}$ *with a unique Nash equilibrium* $\boldsymbol{z}^*$, *OMWU with a learning rate* $\eta \leq \frac{1}{8}$ *guarantees* $\mathrm{KL}(\boldsymbol{z}^*, \boldsymbol{z}_t) \leq C_3 (1 + C_4)^{-t}$, *where* $C_3, C_4 > 0$ *are some constants depending on the game matrix* $\boldsymbol{G}$.

Proofs for this section are deferred to Appendix D, where all problem-dependent constants are specified as well.[1] To the best of our knowledge, Theorem 3 gives the first last-iterate convergence result for OMWU with a concrete linear rate. We note that the uniqueness assumption is critical for our analysis, and whether this is indeed necessary for OMWU is left as an important future direction.

## 5 CONVERGENCE RESULTS FOR OGDA

In this section, we provide last-iterate convergence results for OGDA, which are much more general than those in Section 4. We propose a general condition subsuming many well-studied cases, under which OGDA enjoys a concrete last-iterate convergence guarantee in terms of the $L_2$ distance between $\boldsymbol{z}_t$ and $\mathcal{Z}^*$. The results in this part can be specialized to the setting of bilinear games over simplex, but the unique equilibrium assumption made in Section 4 and in (Daskalakis & Panageas, 2019b) is no longer needed.

Throughout the section we make the assumption that $f$ is $L$-smooth:

**Assumption 1.** *For any* $\boldsymbol{z}, \boldsymbol{z}' \in \mathcal{Z}$, $\|F(\boldsymbol{z}) - F(\boldsymbol{z}')\| \leq L\|\boldsymbol{z} - \boldsymbol{z}'\|$ *holds.*[2]

To introduce our general condition, we first provide some intuition by applying Lemma 1 again. Letting $\psi(\boldsymbol{u}) = \frac{1}{2}\|\boldsymbol{u}\|^2$ in Lemma 1, we get that for OGDA, for any $\boldsymbol{z} \in \mathcal{Z}$ and any $t \geq 1$,

$$2\eta F(\boldsymbol{z}_t)^\top (\boldsymbol{z}_t - \boldsymbol{z}) \leq \|\widehat{\boldsymbol{z}}_t - \boldsymbol{z}\|^2 - \|\widehat{\boldsymbol{z}}_{t+1} - \boldsymbol{z}\|^2 - \|\widehat{\boldsymbol{z}}_{t+1} - \boldsymbol{z}_t\|^2 - \tfrac{15}{16}\|\boldsymbol{z}_t - \widehat{\boldsymbol{z}}_t\|^2 + \tfrac{1}{16}\|\widehat{\boldsymbol{z}}_t - \boldsymbol{z}_{t-1}\|^2.$$

Now we instantiate the inequality above with $\boldsymbol{z} = \Pi_{\mathcal{Z}^*}(\widehat{\boldsymbol{z}}_t) \in \mathcal{Z}^*$. Since $\boldsymbol{z} = \Pi_{\mathcal{Z}^*}(\widehat{\boldsymbol{z}}_t)$ is an equilibrium, we have $F(\boldsymbol{z}_t)^\top (\boldsymbol{z}_t - \boldsymbol{z}) \geq f(\boldsymbol{x}_t, \boldsymbol{y}_t) - f(\boldsymbol{x}, \boldsymbol{y}_t) + f(\boldsymbol{x}_t, \boldsymbol{y}) - f(\boldsymbol{x}_t, \boldsymbol{y}_t) = f(\boldsymbol{x}_t, \boldsymbol{y}) - f(\boldsymbol{x}, \boldsymbol{y}_t) \geq 0$ by the convexity/concavity of $f$ and the optimality of $\boldsymbol{z}$, and thus

$$\|\widehat{\boldsymbol{z}}_{t+1} - \Pi_{\mathcal{Z}^*}(\widehat{\boldsymbol{z}}_t)\|^2 \leq \|\widehat{\boldsymbol{z}}_t - \Pi_{\mathcal{Z}^*}(\widehat{\boldsymbol{z}}_t)\|^2 - \|\widehat{\boldsymbol{z}}_{t+1} - \boldsymbol{z}_t\|^2 - \tfrac{15}{16}\|\boldsymbol{z}_t - \widehat{\boldsymbol{z}}_t\|^2 + \tfrac{1}{16}\|\widehat{\boldsymbol{z}}_t - \boldsymbol{z}_{t-1}\|^2.$$

---

[1] One might find that the constant $C_3$ is exponential in some problem-dependent quantity $T_0$. However, this is simply a loose bound in exchange for more concise presentation — our proof in fact shows that when $t < T_0$, the convergence is of a slower $1/t$ rate, and when $t \geq T_0$, the convergence is linear without this large constant.

[2] This is equivalent to the condition $\mathrm{dist}_q^2(F(\boldsymbol{z}), F(\boldsymbol{z}')) \leq L^2 \mathrm{dist}_p^2(\boldsymbol{z}, \boldsymbol{z}')$ in Lemma 1 with $p = 2$, hence the same notation $L$.

Further noting that the left-hand side is lower bounded by $\text{dist}^2(\widehat{z}_{t+1}, \mathcal{Z}^*)$ by definition, we arrive at

$$\text{dist}^2(\widehat{z}_{t+1}, \mathcal{Z}^*) \leq \text{dist}^2(\widehat{z}_t, \mathcal{Z}^*) - \|\widehat{z}_{t+1} - z_t\|^2 - \tfrac{15}{16}\|z_t - \widehat{z}_t\|^2 + \tfrac{1}{16}\|\widehat{z}_t - z_{t-1}\|^2.$$

Similarly, we define $\Theta_t = \|\widehat{z}_t - \Pi_{\mathcal{Z}^*}(\widehat{z}_t)\|^2 + \tfrac{1}{16}\|\widehat{z}_t - z_{t-1}\|^2$, $\zeta_t = \|\widehat{z}_{t+1} - z_t\|^2 + \|z_t - \widehat{z}_t\|^2$, and rewrite the above as

$$\Theta_{t+1} \leq \Theta_t - \tfrac{15}{16}\zeta_t. \tag{4}$$

As in Section 4, our goal now is to lower bound $\zeta_t$ by some quantity related to $\text{dist}^2(\widehat{z}_{t+1}, \mathcal{Z}^*)$, and then use Eq. (4) to obtain a convergence rate for $\Theta_t$. In order to incorporate more general objective functions into the discussion, in the following Lemma 4, we provide an intermediate lower bound for $\zeta_t$, which will be further related to $\text{dist}^2(\widehat{z}_{t+1}, \mathcal{Z}^*)$ later.

**Lemma 4.** *For any $t \geq 0$ and $z' \in \mathcal{Z}$ with $z' \neq \widehat{z}_{t+1}$, OGDA with $\eta \leq \frac{1}{8L}$ ensures*

$$\|\widehat{z}_{t+1} - z_t\|^2 + \|z_t - \widehat{z}_t\|^2 \geq \frac{32}{81}\eta^2 \frac{\left[F(\widehat{z}_{t+1})^\top(\widehat{z}_{t+1} - z')\right]_+^2}{\|\widehat{z}_{t+1} - z'\|^2}, \tag{5}$$

*where $[a]_+ \triangleq \max\{a, 0\}$, and similarly, for $z' \neq z_{t+1}$,*

$$\|\widehat{z}_{t+1} - z_{t+1}\|^2 + \|z_t - \widehat{z}_{t+1}\|^2 \geq \frac{32}{81}\eta^2 \frac{\left[F(z_{t+1})^\top(z_{t+1} - z')\right]_+^2}{\|z_{t+1} - z'\|^2}. \tag{6}$$

We note that a direct consequence of Lemma 4 is an "average duality gap" guarantee for OGDA when $\mathcal{Z}$ is bounded:

$$\frac{1}{T}\sum_{t=1}^{T} \alpha(z_t) = \frac{1}{T}\sum_{t=1}^{T} \max_{x' \in \mathcal{X}, y' \in \mathcal{Y}} \left(f(x_t, y') - f(x', y_t)\right) = \mathcal{O}\left(\frac{D}{\eta\sqrt{T}}\right) \tag{7}$$

where $D \triangleq \sup_{z, z' \in \mathcal{Z}} \|z - z'\|$ is the diameter of $\mathcal{Z}$ (the duality gap may be undefined when $\mathcal{Z}$ is unbounded). We are not aware of any previous work that gives this result for the constrained case. See Appendix E for the proof of Eq. (7) and comparisons with previous works.

However, to obtain last-iterate convergence results, we need to make sure that the right-hand side of Eq. (5) is large enough. Motivated by this fact, we propose the following general condition on $f$ and $\mathcal{Z}$ to achieve so.

**Definition 1** (Saddle-Point Metric Subregularity (SP-MS))**.** *The SP-MS condition is defined as: for any $z \in \mathcal{Z}\backslash\mathcal{Z}^*$ with $z^* = \Pi_{\mathcal{Z}^*}(z)$,*

$$\sup_{z' \in \mathcal{Z}} \frac{F(z)^\top(z - z')}{\|z - z'\|} \geq C\|z - z^*\|^{\beta+1} \tag{SP-MS}$$

*holds for some parameter $\beta \geq 0$ and $C > 0$.*

We call this condition Saddle-Point Metric Subregularity because the case with $\beta = 0$ is equivalent to one type of metric subregularity in variational inequality problems, as we prove in Appendix F. The condition is also closely related to other error bound conditions that have been identified for variational inequality problems (e.g., Tseng (1995); Gilpin et al. (2008); Malitsky (2019)). Although these works have shown that under similar conditions their algorithms exhibit linear convergence, to the best of our knowledge, there is no previous work that analyzes OGDA or other no-regret learning algorithms using such conditions.

SP-MS covers many standard settings studied in the literature. The first and perhaps the most important example is bilinear games with a polytope feasible set, which in particular includes the classic two-player matrix games considered in Section 4.

**Theorem 5.** *A bilinear game $f(x, y) = x^\top G y$ with $\mathcal{X} \subseteq \mathbb{R}^M$ and $\mathcal{Y} \subseteq \mathbb{R}^N$ being polytopes and $G \in \mathbb{R}^{M \times N}$ satisfies SP-MS with $\beta = 0$.*

We emphasize again that different from Lemma 2, Theorem 5 does not require a unique equilibrium. Note that we have not provided the concrete form of the parameter $C$ in the theorem (which depends on $\mathcal{X}$, $\mathcal{Y}$, and $G$), but it can be found in the proof (see Appendix G).[3] The next example shows that strongly-convex-strongly-concave problems are also special cases of our condition.

**Theorem 6.** *If $f$ is strongly convex in $\boldsymbol{x}$ and strongly concave in $\boldsymbol{y}$, then SP-MS holds with $\beta = 0$.*

Next, we provide a toy example where SP-MS holds with $\beta > 0$.

**Theorem 7.** *Let $\mathcal{X} = \mathcal{Y} \triangleq \{(a, b) : 0 \leq a, b \leq 1, \ a + b = 1\}$, $n > 2$ be an integer, and $f(\boldsymbol{x}, \boldsymbol{y}) = x_1^{2n} - x_1 y_1 - y_1^{2n}$. Then SP-MS holds with $\beta = 2n - 2$.*

With this general condition, we are now able to complete the loop. For any value of $\beta$, we show the following last-iterate convergence guarantee for OGDA.

**Theorem 8.** *For any $\eta \leq \frac{1}{8L}$, if SP-MS holds with $\beta = 0$, then OGDA guarantees linear last-iterate convergence:*

$$\text{dist}^2(\boldsymbol{z}_t, \mathcal{Z}^*) \leq 64\text{dist}^2(\widehat{\boldsymbol{z}}_1, \mathcal{Z}^*)(1 + C_5)^{-t}; \tag{8}$$

*on the other hand, if the condition holds with $\beta > 0$, then we have a slower convergence:*

$$\text{dist}^2(\boldsymbol{z}_t, \mathcal{Z}^*) \leq 32 \left[ \left(1 + 4\left(\frac{4}{\beta}\right)^{\frac{1}{\beta}}\right) \text{dist}^2(\widehat{\boldsymbol{z}}_1, \mathcal{Z}^*) + 2\left(\frac{2}{C_5\beta}\right)^{\frac{1}{\beta}} \right] t^{-\frac{1}{\beta}}, \tag{9}$$

*where $C_5 \triangleq \min\left\{ \frac{16\eta^2 C^2}{81}, \frac{1}{2} \right\}$.*

We defer the proof to Appendix I and make several remarks. First, note that based on a convergence result on $\text{dist}^2(\boldsymbol{z}_t, \mathcal{Z}^*)$, one can immediately obtain a convergence guarantee for the duality gap $\alpha_f(\boldsymbol{z}_t)$ as long as $f$ is also Lipschitz. This is because $\alpha_f(\boldsymbol{z}_t) \leq \max_{\boldsymbol{x}', \boldsymbol{y}'} f(\boldsymbol{x}_t, \boldsymbol{y}') - f(\boldsymbol{x}^*, \boldsymbol{y}') + f(\boldsymbol{x}', \boldsymbol{y}^*) - f(\boldsymbol{x}', \boldsymbol{y}_t) \leq \mathcal{O}(\|\boldsymbol{x}_t - \boldsymbol{x}^*\| + \|\boldsymbol{y}_t - \boldsymbol{y}^*\|) = \mathcal{O}\left(\sqrt{\text{dist}^2(\boldsymbol{z}_t, \mathcal{Z}^*)}\right)$, where $(\boldsymbol{x}^*, \boldsymbol{y}^*) = \Pi_{\mathcal{Z}^*}(\boldsymbol{z}_t)$. While this leads to stronger guarantees compared to Eq. (7), we emphasize that the latter holds even without the SP-MS condition.

Second, our results significantly generalize (Hsieh et al., 2019, Theorem 2) which itself is a consolidated version of several earlier works and also shows a linear convergence rate of OGDA under a condition stronger than our SP-MS with $\beta = 0$ as discussed earlier. More specifically, our results show that linear convergence holds for a much broader set of problems. Furthermore, we also show slower sublinear convergence rates for any value of $\beta > 0$, which is also new as far as we know. In particular, we empirically verify that OGDA indeed does not converge exponentially fast for the toy example defined in Theorem 7 (see Appendix A).

Last but not least, the most significant implication of Theorem 8 is that it provides by far the most general linear convergence result for OGDA for the classic two-player matrix games, or more generally bilinear games with polytope constraints, according to Theorem 5 and Eq. (8). Compared to recent works of (Daskalakis & Panageas, 2018; 2019b) for matrix games (on OGDA or OMWU), our result is considerably stronger: 1) we do not require a unique equilibrium while they do; 2) linear convergence holds for any initial points $\widehat{\boldsymbol{z}}_1$, while their result only holds if the initial points are in a small neighborhood of the unique equilibrium (otherwise the convergence is sublinear initially); 3) our only requirement on the step size is $\eta \leq \frac{1}{8L}$,[4] while they require an exponentially small $\eta$, which does not reflect the behavior of the algorithms in practice. Even compared with our result in Section 4, we see that for OGDA, the unique equilibrium assumption is not required, and we do not have an initial phase of sublinear convergence as in Lemma 2. In Appendix A, we empirically show that OGDA often outperforms OMWU when both are tuned with a constant learning rate.

---

[3]After the first version of this paper, we found that (Gilpin et al., 2008, Lemma 3) gives a simpler proof for our Theorem 5. Although their lemma only focuses on the case where the feasible sets are probability simplices, it can be directly extended to the case of polytopes.

[4]In fact, any $\eta < \frac{1}{2L}$ is enough to achieve linear convergence rate for OGDA, as one can verify by going over our proof. We use $\eta \leq \frac{1}{8L}$ simply for consistency with the results for OMWU (where $\eta$ cannot be set any larger due to technical reasons).

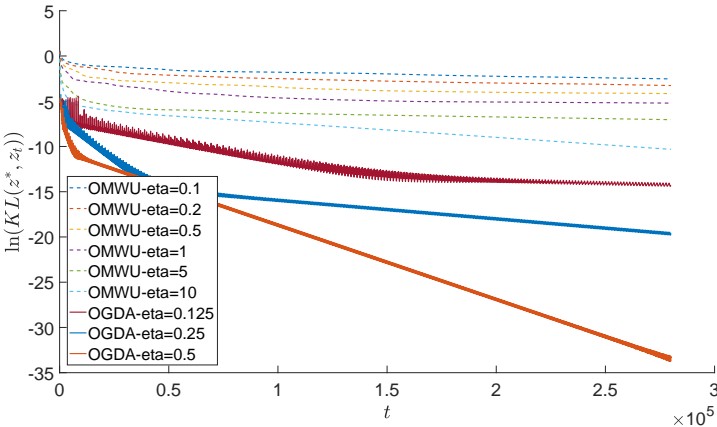

Figure 1: Experiments of OGDA and OMWU with different learning rates for a matrix game $f(\boldsymbol{x}, \boldsymbol{y}) = \boldsymbol{x}^\top \boldsymbol{G} \boldsymbol{y}$. "OGDA/OMWU-eta=$\eta$" represents the curve of OGDA/OMWU with learning rate $\eta$. The configuration order in the legend is consistent with the order of the curves. For OMWU, $\eta \geq 11$ makes the algorithm diverge. The plot confirms the linear convergence of OMWU and OGDA, although OGDA is generally observed to converge faster than OMWU.

One may wonder what happens if a bilinear game has a non-polytope constraint. It turns out that in this case, SP-MS may only hold with $\beta > 0$, due to the following example showing that linear convergence provably does not hold for OGDA when the feasible set has a curved boundary.

**Theorem 9.** *There exists a bilinear game with a non-polytope feasible set such that SP-MS holds with $\beta = 3$, and $\mathrm{dist}^2(\boldsymbol{z}_t, \mathcal{Z}^*) = \Omega(1/t^2)$ holds for* OGDA.

This example indicates that the shape of the feasible set plays an important role in last-iterate convergence, which may be an interesting future direction to investigate, This is also verified empirically in our experiments (see Appendix A).

## 6 EXPERIMENTS FOR MATRIX GAMES

In this section, we provide empirical results on the performance of OGDA and OMWU for matrix games on probability simplex.[5] We include more empirical results in other settings in Appendix A. We set the size of the game matrix to be $32 \times 32$, then generate a random matrix with each entry $G_{ij}$ drawn uniformly at random from $[-1, 1]$, and finally rescale its operator norm to 1. With probability 1, the game has a unique Nash Equilibrium (Daskalakis & Panageas, 2019b).

We compare the performances of OGDA and OMWU. For both algorithms, we choose a series of different learning rates and compare their performances, as shown in Figure 1. The $x$-axis represents time step $t$, and the $y$-axis represents $\ln(\mathrm{KL}(\boldsymbol{z}^*, \boldsymbol{z}_t))$ (we observe similar results using $\mathrm{dist}^2(\boldsymbol{z}^*, \boldsymbol{z}_t)$ or the duality gap as the measure; see Appendix A.1). Note that here we approximate $\boldsymbol{z}^*$ by running OGDA for much more iterations and taking the very last iterate. We also verify that the iterates of OMWU converge to the same point as OGDA.

From Figure 1, we see that all curves eventually become a straight line, supporting our linear convergence results. Generally, the slope of the straight line is larger for a larger learning rate $\eta$. However, the algorithm diverges when $\eta$ exceeds some value (such as 11 for the case of OMWU). Comparing OMWU and OGDA, we see that OGDA converges faster, which is also consistent with our theory if one compares the bounds in Theorem 3 and Theorem 8 (with the value of the constants revealed in the proofs). We find this observation interesting, since OMWU is usually considered more favorable for problems defined over the simplex, especially in terms of regret minimization. Our experiments suggest that, however, in terms of last-iterate convergence, OGDA might perform even better than OMWU.

---

[5]Note that in this case the projection step of OGDA can be implemented efficiently in $O(M \ln M + N \ln N)$ time (Wang & Carreira-Perpiñán, 2013).

ACKNOWLEDGMENTS

The authors would like to thank the anonymous reviewers for providing highly constructive comments which bring about significant improvement of the result during the rebuttal phase. CL would like to thank Yu-Guan Hsieh for many helpful discussions on error bounds and metric subregularity. The authors are supported by NSF Awards IIS-1755781 and IIS-1943607.

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

## A    MORE EXPERIMENT RESULTS

### A.1    MORE EMPIRICAL RESULTS FOR MATRIX GAMES

Here, we provide more plots for the same matrix game experiment described in Section 6. Specifically, the left plot in Figure 2 shows the convergence with respect to $\ln \|\boldsymbol{z}_t - \boldsymbol{z}^*\|$, while the right plot shows the convergence with respect to the logarithm of the duality gap $\ln(\alpha_f(\boldsymbol{z}_t)) = \ln\left(\max_j(\boldsymbol{G}^\top \boldsymbol{x}_t)_j - \min_i(\boldsymbol{G}\boldsymbol{y}_t)_i\right)$. One can see that the plots are very similar to those in Figure 1.

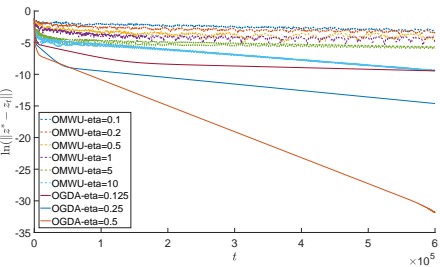
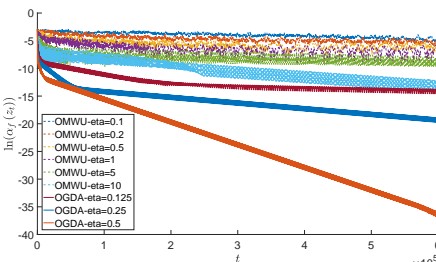

Figure 2: Experiments of OGDA and OMWU with different learning rates on a matrix game $f(\boldsymbol{x}, \boldsymbol{y}) = \boldsymbol{x}^\top \boldsymbol{G}\boldsymbol{y}$, where we generate $\boldsymbol{G} \in \mathbb{R}^{32 \times 32}$ with each entry $G_{ij}$ drawn uniformly at random from $[-1, 1]$ and then rescale $\boldsymbol{G}$'s operator norm to 1. "OGDA/OMWU-eta=$\eta$" represents the curve of OGDA/OMWU with learning rate $\eta$. The configuration order in the legend is consistent with the order of the curves. For OMWU, $\eta \geq 11$ makes the algorithm diverge. The plot confirms the linear convergence of OMWU and OGDA, although OGDA is generally observed to converge faster than OMWU.

### A.2    MATRIX GAME ON CURVED REGIONS

Next, we conduct experiments on a bilinear game similar to the one constructed in the proof of Theorem 9. Specifically, the bilinear game is defined by

$$f(\boldsymbol{x}, \boldsymbol{y}) = x_2 y_1 - x_1 y_2, \quad \mathcal{X} = \mathcal{Y} \triangleq \{(a, b), 0 \leq a \leq \tfrac{1}{2}, 0 \leq b \leq \tfrac{1}{2^n}, a^n \leq b\}.$$

For any positive integer $n$, the equilibrium point of this game is $(0, 0)$ for both $\boldsymbol{x}$ and $\boldsymbol{y}$. Note that in Theorem 9, we prove that OGDA only converges at a rate no better than $\Omega(1/t^2)$ in this game when $n = 2$.

Figure 3 shows the empirical results for various values of $n$. In this figure, we plot $\|\boldsymbol{z}_t - \boldsymbol{z}^*\|$ versus time step $t$ in log-log scale. Note that in a log-log plot, a straight line with slope $s$ implies a convergence rate of order $\mathcal{O}(t^s)$, that is, a sublinear convergence rate. It is clear from Figure 3 that OGDA indeed converges sublinearly for all $n$, supporting our Theorem 9.

### A.3    STRONGLY-CONVEX-STRONGLY-CONCAVE GAMES

In this section, we use the same experiment setup for strongly-convex-strongly-concave games in (Lei et al., 2021), where

$$f(\boldsymbol{x}, \boldsymbol{y}) = x_1^2 - y_1^2 + 2x_1 y_1, \quad \text{and} \quad \mathcal{X} = \mathcal{Y} \triangleq \{(a, b), 0 \leq a, b \leq 1, \ a + b = 1\}.$$

The equilibrium point is $(0, 1)$ for both $\boldsymbol{x}$ and $\boldsymbol{y}$. In Figure 4, we present the log plot of $\|\boldsymbol{z}_t - \boldsymbol{z}^*\|$ versus time step $t$ and compare OGDA with OMWU using different learning rates as in Appendix A.1. The straight line of OGDA implies that OGDA algorithm converges exponentially fast, supporting Theorem 6 and Theorem 8. Also note that here, OGDA outperforms OMWU, which is different from the empirical results shown in (Lei et al., 2021). We hypothesize that this is because they use a different version of OGDA.

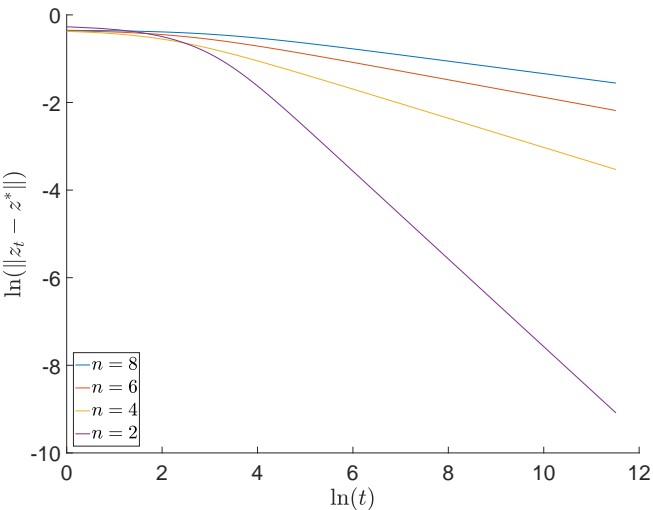

Figure 3: Experiments of OGDA on matrix games with curved regions where $f(\boldsymbol{x}, \boldsymbol{y}) = x_2 y_1 - x_1 y_2$, $\mathcal{X} = \mathcal{Y} \triangleq \{(a, b), 0 \leq a \leq \frac{1}{2}, 0 \leq b \leq \frac{1}{2^n}, a^n \leq b\}$, and $n = 2, 4, 6, 8$. This figure is a log-log plot of $\|\boldsymbol{z}_t - \boldsymbol{z}^*\|$ versus $t$, and it indicates sublinear convergence rates of OGDA in all these games.

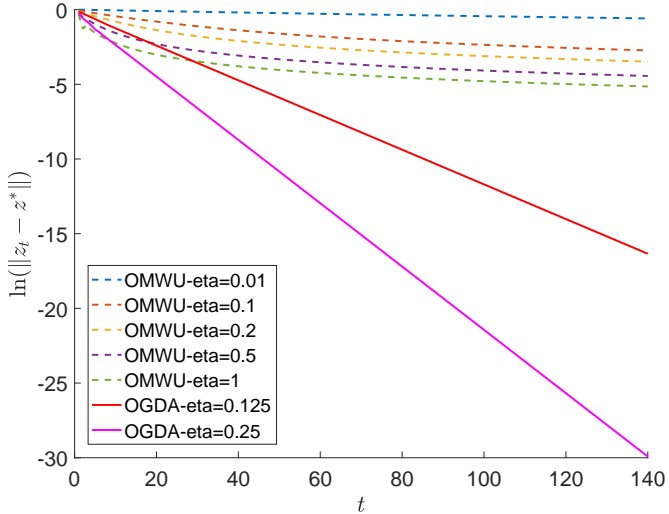

Figure 4: Experiments on a strongly-convex-strongly-concave game where $f(\boldsymbol{x}, \boldsymbol{y}) = x_1^2 - y_1^2 + 2x_1 y_1$ and $\mathcal{X} = \mathcal{Y} \triangleq \{(a, b), 0 \leq a, b \leq 1, a + b = 1\}$. The figure is showing $\ln \|\boldsymbol{z}_t - \boldsymbol{z}^*\|$ versus the time step $t$. The result shows that OGDA enjoys linear convergence and outperforms OMWU in this case.

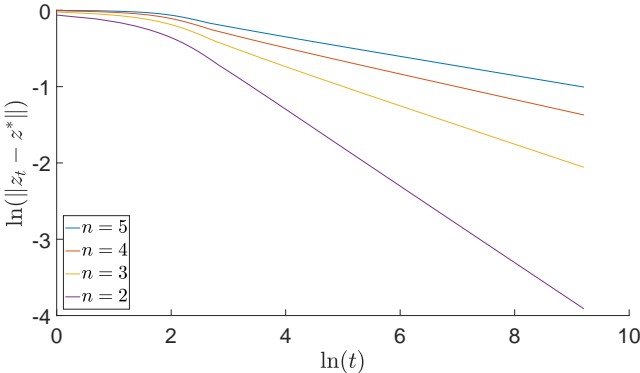

Figure 5: Experiments of OGDA on a set of games satisfying SP-MS with $\beta > 0$, where $f(\boldsymbol{x}, \boldsymbol{y}) = x_1^{2n} - x_1 y_1 - y_1^{2n}$ for some integer $n \geq 2$ and $\mathcal{X} = \mathcal{Y} \triangleq \{(a, b), 0 \leq a, b \leq 1, \ a + b = 1\}$. The result shows that OGDA converges to the Nash equilibrium with sublinear rates in these instances.

### A.4 An Example with $\beta > 0$ for SP-MS

We also consider the toy example in Theorem 7, where $f(\boldsymbol{x}, \boldsymbol{y}) = x_1^{2n} - x_1 y_1 - y_1^{2n}$ for some integer $n \geq 2$ and $\mathcal{X} = \mathcal{Y} \triangleq \{(a, b), 0 \leq a, b \leq 1, \ a + b = 1\}$. The equilibrium point is $(0, 1)$ for both $\boldsymbol{x}$ and $\boldsymbol{y}$. We prove in Theorem 7 that SP-MS does not hold for $\beta = 0$ but does hold for $\beta = 2n - 2$.

The point-wise convergence result is shown in Figure 5, which is again a log-log plot of $\|\boldsymbol{z}_t - \boldsymbol{z}^*\|$ versus time step $t$. One can observe that the convergence rate of OGDA is sublinear, supporting our theory again.

### A.5 Matrix Games with Multiple Nash Equilibria

Finally, we provide empirical results for OGDA and OMWU in matrix games with multiple Nash equilibria, even though theoretically we only prove linear convergence results for OMWU assuming that the Nash equilibrium is unique. We consider the following game matrix

$$
G = \begin{bmatrix} 0 & -1 & 1 & 0 & 0 \\ 1 & 0 & -1 & 0 & 0 \\ -1 & 1 & 0 & 0 & 0 \\ -1 & 1 & 0 & 2 & -1 \\ -1 & 1 & 0 & -1 & 2 \end{bmatrix}.
$$

The value of $G$ is $0$. To verify this, consider $\boldsymbol{x}_0 = \boldsymbol{y}_0 = \begin{bmatrix} \frac{1}{3} & \frac{1}{3} & \frac{1}{3} & 0 & 0 \end{bmatrix}$. Then we have for $\max_{\boldsymbol{y} \in \Delta_5} \boldsymbol{x}_0^\top G \boldsymbol{y} = \min_{\boldsymbol{x} \in \Delta_5} \boldsymbol{x}^\top G \boldsymbol{y}_0 = 0$. Direct calculation gives the following set of Nash equilibria.

$$
\mathcal{X}^* = \{\boldsymbol{x}_0\},
$$
$$
\mathcal{Y}^* = \left\{ \boldsymbol{y} \in \Delta_5 : y_1 = y_2 = y_3; \ \frac{1}{2} y_5 \leq y_4 \leq 2y_5 \right\}.
$$

Figure 6 shows the point-wise convergence result. $\Pi_{\mathcal{Z}^*}(z_t)$ is the projection of $z_t$ on the set of Nash qquilibria. One can observe from the plots that both OGDA and OMWU achieve linear convergence rate in this example. We thus conjecture that the uniqueness assumption for Theorem 3 can be further relaxed.

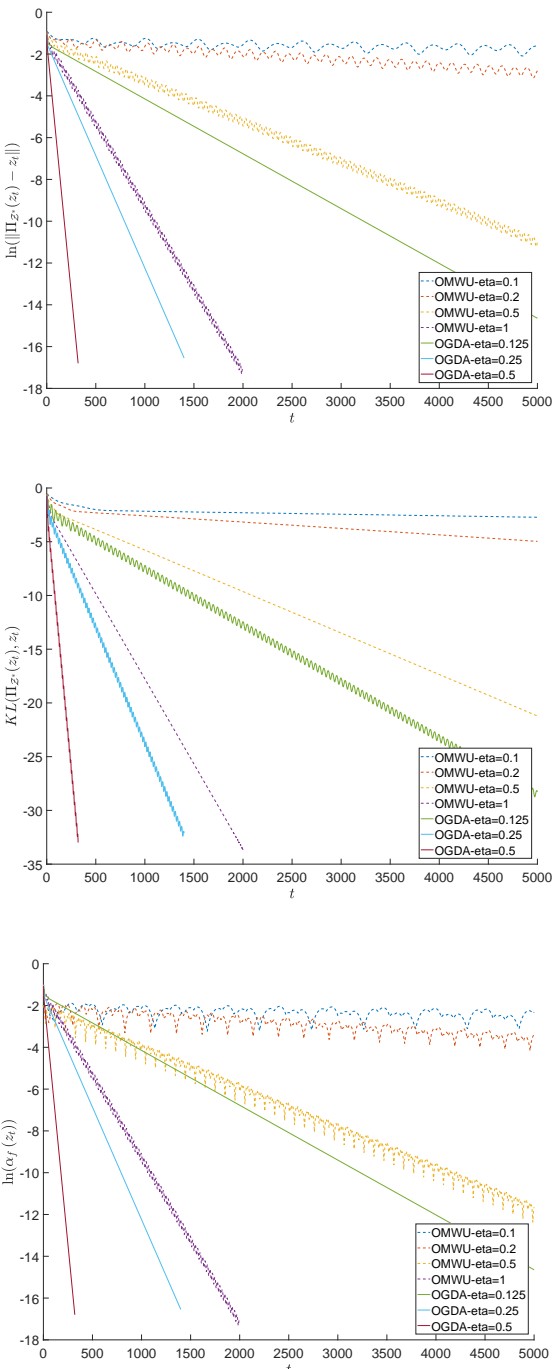

Figure 6: Experiments of OGDA and OMWU with different learning rates on a matrix game with multiple Nash equilibria. "OGDA/OMWU-eta=$\eta$" represents the curve of OGDA/OMWU with learning rate $\eta$. We observe from these plots that both OGDA and OMWU enjoy a linear convergence rate, even though we are only able to show the linear convergence of OMWU under the uniqueness assumption.

## B   LEMMAS FOR OPTIMISTIC MIRROR DESCENT

We prove Lemma 1 in this section. To do so, we use the following two lemmas.

**Lemma 10.** *Let $\mathcal{A}$ be a convex set and $\boldsymbol{u}' = \operatorname{argmin}_{\boldsymbol{u}' \in \mathcal{A}} \{\langle \boldsymbol{u}', \boldsymbol{g} \rangle + D_\psi(\boldsymbol{u}', \boldsymbol{u})\}$. Then for any $\boldsymbol{u}^* \in \mathcal{A}$,*

$$\langle \boldsymbol{u}' - \boldsymbol{u}^*, \boldsymbol{g} \rangle \leq D_\psi(\boldsymbol{u}^*, \boldsymbol{u}) - D_\psi(\boldsymbol{u}^*, \boldsymbol{u}') - D_\psi(\boldsymbol{u}', \boldsymbol{u}). \tag{10}$$

*Proof.* Since $D_\psi(\boldsymbol{u}', \boldsymbol{u}) = \psi(\boldsymbol{u}') - \psi(\boldsymbol{u}) - \langle \nabla \psi(\boldsymbol{u}), \boldsymbol{u}' - \boldsymbol{u} \rangle$, by the first-order optimality condition of $\boldsymbol{u}'$, we have

$$(\boldsymbol{g} + \nabla \psi(\boldsymbol{u}') - \nabla \psi(\boldsymbol{u}))^\top (\boldsymbol{u}^* - \boldsymbol{u}') \geq 0.$$

On the other hand, notice that the right-hand side of Eq. (10) is

$$\psi(\boldsymbol{u}^*) - \psi(\boldsymbol{u}) - \langle \nabla \psi(\boldsymbol{u}), \boldsymbol{u}^* - \boldsymbol{u} \rangle$$
$$- \psi(\boldsymbol{u}^*) + \psi(\boldsymbol{u}') + \langle \nabla \psi(\boldsymbol{u}'), \boldsymbol{u}^* - \boldsymbol{u}' \rangle$$
$$- \psi(\boldsymbol{u}') + \psi(\boldsymbol{u}) + \langle \nabla \psi(\boldsymbol{u}), \boldsymbol{u}' - \boldsymbol{u} \rangle$$
$$= \langle \nabla \psi(\boldsymbol{u}') - \nabla \psi(\boldsymbol{u}), \boldsymbol{u}^* - \boldsymbol{u}' \rangle.$$

Therefore, Eq. (10) is equivalent to $\langle \boldsymbol{g} + \nabla \psi(\boldsymbol{u}') - \nabla \psi(\boldsymbol{u}), \boldsymbol{u}^* - \boldsymbol{u}' \rangle \geq 0$, which we have already shown above. □

**Lemma 11.** *Suppose that $\psi$ satisfies $D_\psi(\boldsymbol{x}, \boldsymbol{x}') \geq \frac{1}{2}\|\boldsymbol{x} - \boldsymbol{x}'\|_p^2$ for some $p \geq 1$, and let $\boldsymbol{u}, \boldsymbol{u}_1, \boldsymbol{u}_2 \in \mathcal{A}$ (a convex set) be related by the following:*

$$\boldsymbol{u}_1 = \operatorname*{argmin}_{\boldsymbol{u}' \in \mathcal{A}} \{\langle \boldsymbol{u}', \boldsymbol{g}_1 \rangle + D_\psi(\boldsymbol{u}', \boldsymbol{u})\},$$
$$\boldsymbol{u}_2 = \operatorname*{argmin}_{\boldsymbol{u}' \in \mathcal{A}} \{\langle \boldsymbol{u}', \boldsymbol{g}_2 \rangle + D_\psi(\boldsymbol{u}', \boldsymbol{u})\}.$$

*Then we have*

$$\|\boldsymbol{u}_1 - \boldsymbol{u}_2\|_p \leq \|\boldsymbol{g}_1 - \boldsymbol{g}_2\|_q,$$

*where $q \geq 1$ and $\frac{1}{p} + \frac{1}{q} = 1$.*

*Proof.* By the first-order optimality conditions of $\boldsymbol{u}_1$ and $\boldsymbol{u}_2$, we have

$$\langle \nabla \psi(\boldsymbol{u}_1) - \nabla \psi(\boldsymbol{u}) + \boldsymbol{g}_1, \boldsymbol{u}_2 - \boldsymbol{u}_1 \rangle \geq 0,$$
$$\langle \nabla \psi(\boldsymbol{u}_2) - \nabla \psi(\boldsymbol{u}) + \boldsymbol{g}_2, \boldsymbol{u}_1 - \boldsymbol{u}_2 \rangle \geq 0.$$

Summing them up and rearranging the terms, we get

$$\langle \boldsymbol{u}_2 - \boldsymbol{u}_1, \boldsymbol{g}_1 - \boldsymbol{g}_2 \rangle \geq \langle \nabla \psi(\boldsymbol{u}_1) - \nabla \psi(\boldsymbol{u}_2), \boldsymbol{u}_1 - \boldsymbol{u}_2 \rangle. \tag{11}$$

By the condition on $\psi$, we have $\langle \nabla \psi(\boldsymbol{u}_1), \boldsymbol{u}_1 - \boldsymbol{u}_2 \rangle \geq \psi(\boldsymbol{u}_1) - \psi(\boldsymbol{u}_2) + \frac{1}{2}\|\boldsymbol{u}_1 - \boldsymbol{u}_2\|_p^2$ and $\langle \nabla \psi(\boldsymbol{u}_2), \boldsymbol{u}_2 - \boldsymbol{u}_1 \rangle \geq \psi(\boldsymbol{u}_2) - \psi(\boldsymbol{u}_1) + \frac{1}{2}\|\boldsymbol{u}_1 - \boldsymbol{u}_2\|_p^2$. Summing them up we get $\langle \nabla \psi(\boldsymbol{u}_1) - \nabla \psi(\boldsymbol{u}_2), \boldsymbol{u}_1 - \boldsymbol{u}_2 \rangle \geq \|\boldsymbol{u}_1 - \boldsymbol{u}_2\|_p^2$. Combining this with Eq. (11) we get

$$\langle \boldsymbol{u}_2 - \boldsymbol{u}_1, \boldsymbol{g}_1 - \boldsymbol{g}_2 \rangle \geq \|\boldsymbol{u}_1 - \boldsymbol{u}_2\|_p^2.$$

Since $\langle \boldsymbol{u}_2 - \boldsymbol{u}_1, \boldsymbol{g}_1 - \boldsymbol{g}_2 \rangle \leq \|\boldsymbol{u}_1 - \boldsymbol{u}_2\|_p \|\boldsymbol{g}_1 - \boldsymbol{g}_2\|_q$ by Hölder's inequality, we further get $\|\boldsymbol{u}_1 - \boldsymbol{u}_2\|_p \leq \|\boldsymbol{g}_1 - \boldsymbol{g}_2\|_q$. □

*Proof of Lemma 1.* Considering Eq. (2), and using Lemma 10 with $\boldsymbol{u} = \widehat{\boldsymbol{z}}_t$, $\boldsymbol{u}' = \widehat{\boldsymbol{z}}_{t+1}$, $\boldsymbol{u}^* = \boldsymbol{z}$, and $\boldsymbol{g} = \eta F(\boldsymbol{z}_t)$, we get

$$\eta F(\boldsymbol{z}_t)^\top (\widehat{\boldsymbol{z}}_{t+1} - \boldsymbol{z}) \leq D_\psi(\boldsymbol{z}, \widehat{\boldsymbol{z}}_t) - D_\psi(\boldsymbol{z}, \widehat{\boldsymbol{z}}_{t+1}) - D_\psi(\widehat{\boldsymbol{z}}_{t+1}, \widehat{\boldsymbol{z}}_t).$$

Considering Eq. (1), and using Lemma 10 with $\boldsymbol{u} = \widehat{\boldsymbol{z}}_t$, $\boldsymbol{u}' = \boldsymbol{z}_t$, $\boldsymbol{u}^* = \widehat{\boldsymbol{z}}_{t+1}$, and $\boldsymbol{g} = \eta F(\boldsymbol{z}_{t-1})$, we get

$$\eta F(\boldsymbol{z}_{t-1})^\top (\boldsymbol{z}_t - \widehat{\boldsymbol{z}}_{t+1}) \leq D_\psi(\widehat{\boldsymbol{z}}_{t+1}, \widehat{\boldsymbol{z}}_t) - D_\psi(\widehat{\boldsymbol{z}}_{t+1}, \boldsymbol{z}_t) - D_\psi(\boldsymbol{z}_t, \widehat{\boldsymbol{z}}_t).$$

Summing up the two inequalities above, and adding $\eta\left(F(z_t) - F(z_{t-1})\right)^\top (z_t - \widehat{z}_{t+1})$ to both sides, we get

$$\eta F(z_t)^\top (z_t - z)$$
$$\leq D_\psi(z, \widehat{z}_t) - D_\psi(z, \widehat{z}_{t+1}) - D_\psi(\widehat{z}_{t+1}, z_t) - D_\psi(z_t, \widehat{z}_t) + \eta\left(F(z_t) - F(z_{t-1})\right)^\top (z_t - \widehat{z}_{t+1}).$$
(12)

Using Lemma 11 with $u = \widehat{x}_t$, $u_1 = x_t$, $u_2 = \widehat{x}_{t+1}$, $g_1 = \eta\nabla_x f(z_{t-1})$ and $g_2 = \eta\nabla_x f(z_t)$, we get $\|x_t - \widehat{x}_{t+1}\|_p \leq \eta\|\nabla_x f(z_{t-1}) - \nabla_x f(z_t)\|_q$. Similarly, we have $\|y_t - \widehat{y}_{t+1}\|_p \leq \eta\|\nabla_y f(z_t) - \nabla_y f(z_{t-1})\|_q$. Therefore, by Hölder's inequality, we have

$$\eta\left(F(z_t) - F(z_{t-1})\right)^\top (z_t - \widehat{z}_{t+1})$$
$$\leq \eta\|x_t - \widehat{x}_{t+1}\|_p\|\nabla_x f(z_{t-1}) - \nabla_x f(z_t)\|_q + \eta\|y_t - \widehat{y}_{t+1}\|_p\|\nabla_y f(z_{t-1}) - \nabla_y f(z_t)\|_q$$
$$\leq \eta^2\|\nabla_x f(z_{t-1}) - \nabla_x f(z_t)\|_q^2 + \eta^2\|\nabla_y f(z_{t-1}) - \nabla_y f(z_t)\|_q^2$$
$$= \eta^2 \text{dist}_q^2(F(z_t), F(z_{t-1}))$$
$$\leq \eta^2 L^2 \text{dist}_p^2(z_t, z_{t-1}) \qquad\qquad\qquad\qquad\qquad \text{(by assumption)}$$
$$\leq \frac{1}{64}\text{dist}_p^2(z_t, z_{t-1}). \qquad\qquad\qquad\qquad\qquad \text{(by our choice of } \eta)$$

Continuing from Eq. (12), we then have

$$\eta F(z_t)^\top (z_t - z)$$
$$\leq D_\psi(z, \widehat{z}_t) - D_\psi(z, \widehat{z}_{t+1}) - D_\psi(\widehat{z}_{t+1}, z_t) - D_\psi(z_t, \widehat{z}_t) + \frac{1}{64}\text{dist}_p^2(z_t, z_{t-1})$$
$$\leq D_\psi(z, \widehat{z}_t) - D_\psi(z, \widehat{z}_{t+1}) - D_\psi(\widehat{z}_{t+1}, z_t) - D_\psi(z_t, \widehat{z}_t) + \frac{1}{32}\text{dist}_p^2(z_t, \widehat{z}_t) + \frac{1}{32}\text{dist}_p^2(\widehat{z}_t, z_{t-1})$$
$$\qquad\qquad\qquad\qquad\qquad (\|u + v\|_p^2 \leq (\|u\|_p + \|v\|_p)^2 \leq 2\|u\|_p^2 + 2\|v\|_p^2)$$
$$\leq D_\psi(z, \widehat{z}_t) - D_\psi(z, \widehat{z}_{t+1}) - D_\psi(\widehat{z}_{t+1}, z_t) - D_\psi(z_t, \widehat{z}_t) + \frac{1}{16}D_\psi(z_t, \widehat{z}_t) + \frac{1}{16}D_\psi(\widehat{z}_t, z_{t-1})$$
$$\qquad\qquad\qquad\qquad\qquad\qquad\qquad \text{(by the assumption on } \psi)$$
$$= D_\psi(z, \widehat{z}_t) - D_\psi(z, \widehat{z}_{t+1}) - D_\psi(\widehat{z}_{t+1}, z_t) - \frac{15}{16}D_\psi(z_t, \widehat{z}_t) + \frac{1}{16}D_\psi(\widehat{z}_t, z_{t-1}).$$

This concludes the proof. □

## C  AN AUXILIARY LEMMA ON RECURSIVE FORMULAS

Here, we provide an auxiliary lemma that gives an explicit bound based on a particular recursive formula. This will be useful later for deriving the convergence rate.

**Lemma 12.** *Consider a non-negative sequence $\{B_t\}_{t=1,2,\dots}$ that satisfies for some $p > 0$ and $q > 0$,*

- $B_{t+1} \leq B_t - qB_{t+1}^{p+1}, \quad \forall t \geq 1$

- $q(1+p)B_1^p \leq 1.$

*Then $B_t \leq ct^{-\frac{1}{p}}$, where $c = \max\left\{B_1, \left(\frac{2}{qp}\right)^{\frac{1}{p}}\right\}$.*

*Proof.* We first prove that $B_{t+1} \leq B_t - \frac{q}{2}B_t^{p+1}$. Notice that since $B_t$ are all non-negative, by the first condition, we have $B_{t+1} \leq B_t \leq \dots \leq B_1$. Using the fundamental theorem of calculus, we have

$$B_t^{p+1} - B_{t+1}^{p+1} = \int_{B_{t+1}}^{B_t} \left(\frac{\mathrm{d}}{\mathrm{d}x}x^{p+1}\right)\mathrm{d}x = (p+1)\int_{B_{t+1}}^{B_t} x^p \mathrm{d}x \leq (p+1)(B_t - B_{t+1})B_t^p$$

and thus

$$B_{t+1} \leq B_t - qB_{t+1}^{p+1} \leq B_t - qB_t^{p+1} + q(p+1)(B_t - B_{t+1})B_t^p.$$

By rearranging, we get

$$B_{t+1} \le \left(1 - \frac{qB_t^p}{1 + q(1+p)B_t^p}\right) B_t \le \left(1 - \frac{qB_t^p}{2}\right) B_t = B_t - \frac{q}{2}B_t^{p+1},$$

where the last inequality is because $q(1+p)B_t^p \le q(1+p)B_1^p \le 1$.

Below we use induction to prove $B_t \le ct^{-\frac{1}{p}}$, where $c = \max\left\{B_1, \left(\frac{2}{qp}\right)^{\frac{1}{p}}\right\}$. This clearly holds

for $t = 1$. Suppose that it holds for $1, \ldots, t$. Note that the function $f(B_t) = \left(1 - \frac{q}{2}B_t^p\right) B_t$ is increasing in $B_t$ as $f'(B_t) = 1 - \frac{q(p+1)}{2}B_t^p \ge 1 - \frac{q(p+1)}{2}B_1^p \ge 0$. Therefore, we apply the induction hypothesis and get

$$\begin{aligned}
B_{t+1} &\le \left(1 - \frac{q}{2}B_t^p\right) B_t \le \left(1 - \frac{q}{2}c^p t^{-1}\right) ct^{-\frac{1}{p}} \\
&= ct^{-\frac{1}{p}} - \frac{q}{2}c^{p+1}t^{-1-\frac{1}{p}} \le ct^{-\frac{1}{p}} - \frac{c}{p}t^{-1-\frac{1}{p}} \qquad (\frac{c}{p} \le \frac{q}{2}c^{p+1} \text{ by the definition of } c) \\
&\le c(t+1)^{-\frac{1}{p}},
\end{aligned}$$

where the last inequality is by the fundamental theorem of calculus:

$$\begin{aligned}
t^{-\frac{1}{p}} - (1+t)^{-\frac{1}{p}} &= \int_{1+t}^{t} \left(\frac{\mathrm{d}}{\mathrm{d}x}x^{-\frac{1}{p}}\right) \mathrm{d}x = \int_{1+t}^{t} \left(-\frac{1}{p}\right) x^{-1-\frac{1}{p}}\mathrm{d}x \\
&= \int_{t}^{t+1} \frac{1}{p}x^{-1-\frac{1}{p}}\mathrm{d}x \le \frac{1}{p}t^{-1-\frac{1}{p}}.
\end{aligned}$$

This completes the induction. $\square$

## D  PROOFS OF LEMMA 2 AND THEOREM 3

In this section, we consider $f(\boldsymbol{x}, \boldsymbol{y}) = \boldsymbol{x}^\top \boldsymbol{G}\boldsymbol{y}$ with $\mathcal{X} = \Delta_M$ and $\mathcal{Y} = \Delta_N$ being simplex and $\boldsymbol{G} \in [-1,1]^{M \times N}$. We assume that $\boldsymbol{G}$ has a unique Nash equilibrium $\boldsymbol{z}^* = (\boldsymbol{x}^*, \boldsymbol{y}^*)$. The value of the game is denoted as $\rho = \min_{\boldsymbol{x} \in \mathcal{X}} \max_{\boldsymbol{y} \in \mathcal{Y}} \boldsymbol{x}^\top \boldsymbol{G}\boldsymbol{y} = \max_{\boldsymbol{y} \in \mathcal{Y}} \min_{\boldsymbol{x} \in \mathcal{X}} \boldsymbol{x}^\top \boldsymbol{G}\boldsymbol{y} = \boldsymbol{x}^{*\top} \boldsymbol{G}\boldsymbol{y}^*$.

Before proving Lemma 2 and Theorem 3, in Section D.1, we define some constants for later analysis; in Section D.2, we state more auxiliary lemmas, which are useful when proving Lemma 2 and Theorem 3 in Section D.3.

### D.1  SOME PROBLEM-DEPENDENT CONSTANTS

First, we define a constant $\xi$ that is determined by $\boldsymbol{G}$.

**Definition 2.**

$$\xi \triangleq \min \left\{ \min_{i \notin supp(\boldsymbol{x}^*)} (\boldsymbol{G}\boldsymbol{y}^*)_i - \rho, \quad \rho - \max_{i \notin supp(\boldsymbol{y}^*)} (\boldsymbol{G}^\top \boldsymbol{x}^*)_i \right\} \in (0, 1].$$

The fact $\xi \le 1$ can be shown by:

$$\xi \le \frac{\min_{i \notin supp(\boldsymbol{x}^*)}(\boldsymbol{G}\boldsymbol{y}^*)_i - \rho + \rho - \max_{i \notin supp(\boldsymbol{y}^*)}(\boldsymbol{G}^\top \boldsymbol{x}^*)_i}{2} \le \frac{\|\boldsymbol{G}\boldsymbol{y}^*\|_\infty + \|\boldsymbol{G}^\top \boldsymbol{x}^*\|_\infty}{2} \le 1,$$

while the fact $\xi > 0$ is a direct consequence of Lemma C.3 of Mertikopoulos et al. (2018), stated below.

**Lemma 13** (Lemma C.3 of Mertikopoulos et al. (2018)). *Let $\boldsymbol{G} \in \mathbb{R}^{M \times N}$ be a game matrix for a two-player zero-sum game with value $\rho$. Then there exists a Nash equilibrium $(\boldsymbol{x}^*, \boldsymbol{y}^*)$ such that*

$$\begin{aligned}
(\boldsymbol{G}\boldsymbol{y}^*)_i &= \rho & \forall i \in supp(\boldsymbol{x}^*), \\
(\boldsymbol{G}\boldsymbol{y}^*)_i &> \rho & \forall i \notin supp(\boldsymbol{x}^*), \\
(\boldsymbol{G}^\top \boldsymbol{x}^*)_i &= \rho & \forall i \in supp(\boldsymbol{y}^*), \\
(\boldsymbol{G}^\top \boldsymbol{x}^*)_i &< \rho & \forall i \notin supp(\boldsymbol{y}^*).
\end{aligned}$$

Below, we define $\mathcal{V}^*(\mathcal{Z}) = \mathcal{V}^*(\mathcal{X}) \times \mathcal{V}^*(\mathcal{Y})$, where

$$\mathcal{V}^*(\mathcal{X}) \triangleq \{\boldsymbol{x} : \boldsymbol{x} \in \Delta_M, \ \mathrm{supp}(\boldsymbol{x}) \subseteq \mathrm{supp}(\boldsymbol{x}^*)\}$$

and

$$\mathcal{V}^*(\mathcal{Y}) \triangleq \{\boldsymbol{y} : \boldsymbol{y} \in \Delta_N, \ \mathrm{supp}(\boldsymbol{y}) \subseteq \mathrm{supp}(\boldsymbol{y}^*)\}.$$

**Definition 3.**

$$c_x \triangleq \min_{\boldsymbol{x} \in \Delta_M \setminus \{\boldsymbol{x}^*\}} \max_{\boldsymbol{y} \in \mathcal{V}^*(\mathcal{Y})} \frac{(\boldsymbol{x} - \boldsymbol{x}^*)^\top \boldsymbol{G} \boldsymbol{y}}{\|\boldsymbol{x} - \boldsymbol{x}^*\|_1}, \qquad c_y \triangleq \min_{\boldsymbol{y} \in \Delta_N \setminus \{\boldsymbol{y}^*\}} \max_{\boldsymbol{x} \in \mathcal{V}^*(\mathcal{X})} \frac{\boldsymbol{x}^\top \boldsymbol{G} (\boldsymbol{y}^* - \boldsymbol{y})}{\|\boldsymbol{y}^* - \boldsymbol{y}\|_1}.$$

Note that in the definition of $c_x$ and $c_y$, the outer minimization is over an open set, which may make the definition problematic as the optimal value may not be attained. However, the following lemma shows that $c_x$ and $c_y$ are well-defined.

**Lemma 14.** $c_x$ and $c_y$ are well-defined, and $0 < c_x, c_y \leq 1$.

*Proof.* We first show $c_x$ and $c_y$ are well-defined. To simplify the notations, we define $x_{\min}^* \triangleq \min_{i \in \mathrm{supp}(\boldsymbol{x}^*)} x_i^*$ and $\mathcal{X}' \triangleq \{\boldsymbol{x} : \boldsymbol{x} \in \Delta_M, \ \|\boldsymbol{x} - \boldsymbol{x}^*\|_1 \geq x_{\min}^*\}$, and define $y_{\min}^*$ and $\mathcal{Y}'$ similarly. We will show that

$$c_x = \min_{\boldsymbol{x} \in \mathcal{X}'} \max_{\boldsymbol{y} \in \mathcal{V}^*(\mathcal{Y})} \frac{(\boldsymbol{x} - \boldsymbol{x}^*)^\top \boldsymbol{G} \boldsymbol{y}}{\|\boldsymbol{x} - \boldsymbol{x}^*\|_1}, \quad c_y = \min_{\boldsymbol{y} \in \mathcal{Y}'} \max_{\boldsymbol{x} \in \mathcal{V}^*(\mathcal{X})} \frac{\boldsymbol{x}^\top \boldsymbol{G} (\boldsymbol{y}^* - \boldsymbol{y})}{\|\boldsymbol{y}^* - \boldsymbol{y}\|_1},$$

which are well-defined as the outer minimization is now over a closed set. Consider $c_x$, it suffices to show that for any $\boldsymbol{x} \in \Delta_M$ such that $\boldsymbol{x} \neq \boldsymbol{x}^*$ and $\|\boldsymbol{x} - \boldsymbol{x}^*\|_1 < x_{\min}^*$, there exists $\boldsymbol{x}' \in \Delta_M$ such that $\|\boldsymbol{x}' - \boldsymbol{x}^*\|_1 = x_{\min}^*$ and

$$\frac{(\boldsymbol{x} - \boldsymbol{x}^*)^\top \boldsymbol{G} \boldsymbol{y}}{\|\boldsymbol{x} - \boldsymbol{x}^*\|_1} = \frac{(\boldsymbol{x}' - \boldsymbol{x}^*)^\top \boldsymbol{G} \boldsymbol{y}}{\|\boldsymbol{x}' - \boldsymbol{x}^*\|_1}, \ \forall \boldsymbol{y}. \tag{13}$$

In fact, we can simply choose $\boldsymbol{x}' = \boldsymbol{x}^* + (\boldsymbol{x} - \boldsymbol{x}^*) \cdot \frac{x_{\min}^*}{\|\boldsymbol{x} - \boldsymbol{x}^*\|_1}$. We first argue that $\boldsymbol{x}'$ is still in $\Delta_M$. For each $j \in [K]$, if $x_j - x_j^* \geq 0$, we surely have $x_j' \geq x_j^* + 0 \geq 0$; otherwise, $x_j^* > x_j \geq 0$ and thus $j \in \mathrm{supp}(\boldsymbol{x}^*)$ and $x_j^* \geq x_{\min}^*$, which implies $x_j' \geq x_j^* - |x_j - x_j^*| \cdot \frac{x_{\min}^*}{\|\boldsymbol{x} - \boldsymbol{x}^*\|_1} \geq x_j^* - x_{\min}^* \geq 0$. In addition, $\sum_j x_j' = \sum_j x_j^* = 1$. Combining these facts, we have $\boldsymbol{x}' \in \Delta_M$.

Moreover, according to the definition of $\boldsymbol{x}'$, $\|\boldsymbol{x}' - \boldsymbol{x}^*\|_1 = x_{\min}^*$ holds. Also, since $\boldsymbol{x}^* - \boldsymbol{x}$ and $\boldsymbol{x}^* - \boldsymbol{x}'$ are parallel vectors, Eq. (13) is satisfied. The arguments above show that the $c_x$ in Definition 3 is a well-defined real number. The case of $c_y$ is similar.

Now we show $0 < c_x, c_y \leq 1$. The fact that $c_x, c_y \leq 1$ is a direct consequence of $\boldsymbol{G}$ being in $[-1, 1]^{M \times N}$. Below, we use contradiction to prove that $c_y > 0$. First, if $c_y < 0$, then there exists $\boldsymbol{y} \neq \boldsymbol{y}^*$ such that $\boldsymbol{x}^{*\top} \boldsymbol{G} \boldsymbol{y}^* < \boldsymbol{x}^{*\top} \boldsymbol{G} \boldsymbol{y}$. This contradicts with the fact that $(\boldsymbol{x}^*, \boldsymbol{y}^*)$ is the equilibrium.

On the other hand, if $c_y = 0$, then there is some $\boldsymbol{y} \neq \boldsymbol{y}^*$ such that

$$\max_{\boldsymbol{x} \in \mathcal{V}^*(\mathcal{X})} \boldsymbol{x}^\top \boldsymbol{G} (\boldsymbol{y}^* - \boldsymbol{y}) = 0. \tag{14}$$

Consider the point $\boldsymbol{y}' = \boldsymbol{y}^* + \frac{\xi}{2}(\boldsymbol{y} - \boldsymbol{y}^*)$ (recall the definition of $\xi$ in Definition 2 and that $0 < \xi \leq 1$), which lies on the line segment between $\boldsymbol{y}^*$ and $\boldsymbol{y}$. Then, for any $\boldsymbol{x} \in \mathcal{X}$,

$$
\begin{aligned}
\boldsymbol{x}^\top \boldsymbol{G} \boldsymbol{y}' &= \sum_{i \notin \text{supp}(\boldsymbol{x}^*)} x_i (\boldsymbol{G} \boldsymbol{y}')_i + \sum_{i \in \text{supp}(\boldsymbol{x}^*)} x_i (\boldsymbol{G} \boldsymbol{y}')_i \\
&\geq \sum_{i \notin \text{supp}(\boldsymbol{x}^*)} \left( x_i (\boldsymbol{G} \boldsymbol{y}^*)_i - x_i \|\boldsymbol{y}' - \boldsymbol{y}^*\|_1 \right) + \sum_{i \in \text{supp}(\boldsymbol{x}^*)} \left( \frac{\xi}{2} \cdot x_i (\boldsymbol{G}(\boldsymbol{y} - \boldsymbol{y}^*))_i + x_i (\boldsymbol{G} \boldsymbol{y}^*)_i \right) \\
&\qquad \text{(using } G_{ij} \in [-1, -1] \text{ for the first part and } \boldsymbol{y}' = \boldsymbol{y}^* + \frac{\xi}{2}(\boldsymbol{y} - \boldsymbol{y}^*) \text{ for the second)} \\
&\geq \sum_{i \notin \text{supp}(\boldsymbol{x}^*)} \left( x_i (\boldsymbol{G} \boldsymbol{y}^*)_i - x_i \|\boldsymbol{y}' - \boldsymbol{y}^*\|_1 \right) + \sum_{i \in \text{supp}(\boldsymbol{x}^*)} x_i \rho \\
&\qquad \text{(using Eq. (14) and } (\boldsymbol{G} \boldsymbol{y}^*)_i = \rho \text{ for all } i \in \text{supp}(\boldsymbol{x}^*)) \\
&\geq \sum_{i \notin \text{supp}(\boldsymbol{x}^*)} \left( x_i \left( (\boldsymbol{G} \boldsymbol{y}^*)_i - \xi \right) \right) + \sum_{i \in \text{supp}(\boldsymbol{x}^*)} x_i \rho \\
&\qquad \text{(using } \boldsymbol{y}' - \boldsymbol{y}^* = \frac{\xi}{2}(\boldsymbol{y} - \boldsymbol{y}^*) \text{ and } \|\boldsymbol{y} - \boldsymbol{y}^*\|_1 \leq 2) \\
&\geq \sum_{i \notin \text{supp}(\boldsymbol{x}^*)} x_i \rho + \sum_{i \in \text{supp}(\boldsymbol{x}^*)} x_i \rho \qquad \text{(by the definition of } \xi) \\
&= \rho.
\end{aligned}
$$

This shows that $\min_{\boldsymbol{x} \in \mathcal{X}} \boldsymbol{x}^\top \boldsymbol{G} \boldsymbol{y}' \geq \rho$, that is, $\boldsymbol{y}' \neq \boldsymbol{y}^*$ is also a maximin point, contradicting that $\boldsymbol{z}^*$ is unique. Therefore, $c_y > 0$ has to hold, and so does $c_x > 0$ by the same argument. $\qquad \square$

Finally, we define the following constant that depends on $\boldsymbol{G}$:

**Definition 4.**

$$
\epsilon \triangleq \min_{j \in \text{supp}(\boldsymbol{z}^*)} \exp\left( -\frac{\ln(MN)}{z_j^*} \right).
$$

### D.2   AUXILIARY LEMMAS

All lemmas stated in this section is for the case $f(\boldsymbol{x}, \boldsymbol{y}) = \boldsymbol{x}^\top \boldsymbol{G} \boldsymbol{y}$ with $\mathcal{Z} = \Delta_M \times \Delta_N$ and a unique Nash equilibrium $\boldsymbol{z}^* = (\boldsymbol{x}^*, \boldsymbol{y}^*)$.

**Lemma 15.** *For any $\boldsymbol{z} \in \mathcal{Z}$, we have*

$$
\max_{\boldsymbol{z}' \in \mathcal{V}^*(\mathcal{Z})} F(\boldsymbol{z})^\top (\boldsymbol{z} - \boldsymbol{z}') \geq C \|\boldsymbol{z}^* - \boldsymbol{z}\|_1
$$

*for $C = \min\{c_x, c_y\} \in (0, 1]$.*

*Proof.* Recall that $\rho = \boldsymbol{x}^{*\top} \boldsymbol{G} \boldsymbol{y}^*$ is the game value and note that

$$
\begin{aligned}
\max_{\boldsymbol{z}' \in \mathcal{V}^*(\mathcal{Z})} F(\boldsymbol{z})^\top (\boldsymbol{z} - \boldsymbol{z}') &= \max_{\boldsymbol{z}' \in \mathcal{V}^*(\mathcal{Z})} (\boldsymbol{x} - \boldsymbol{x}')^\top \boldsymbol{G} \boldsymbol{y} + \boldsymbol{x}^\top \boldsymbol{G} (\boldsymbol{y}' - \boldsymbol{y}) = \max_{\boldsymbol{z}' \in \mathcal{V}^*(\mathcal{Z})} -\boldsymbol{x}'^\top \boldsymbol{G} \boldsymbol{y} + \boldsymbol{x}^\top \boldsymbol{G} \boldsymbol{y}' \\
&= \max_{\boldsymbol{x}' \in \mathcal{V}^*(\mathcal{X})} \left( \rho - \boldsymbol{x}'^\top \boldsymbol{G} \boldsymbol{y} \right) + \max_{\boldsymbol{y}' \in \mathcal{V}^*(\mathcal{Y})} \left( \boldsymbol{x}^\top \boldsymbol{G} \boldsymbol{y}' - \rho \right) \\
&= \max_{\boldsymbol{x}' \in \mathcal{V}^*(\mathcal{X})} \boldsymbol{x}'^\top \boldsymbol{G} (\boldsymbol{y}^* - \boldsymbol{y}) + \max_{\boldsymbol{y}' \in \mathcal{V}^*(\mathcal{Y})} (\boldsymbol{x} - \boldsymbol{x}^*)^\top \boldsymbol{G} \boldsymbol{y}' \qquad \text{(Lemma 13)} \\
&\geq c_y \|\boldsymbol{y}^* - \boldsymbol{y}\|_1 + c_x \|\boldsymbol{x}^* - \boldsymbol{x}\|_1 \qquad \text{(by Definition 3)} \\
&\geq \min\{c_x, c_y\} \|\boldsymbol{z}^* - \boldsymbol{z}\|_1,
\end{aligned}
$$

which completes the proof. $\qquad \square$

**Lemma 16.** *For any $\boldsymbol{z} \in \mathcal{Z}$, we have*

$$
\text{KL}(\boldsymbol{z}^*, \boldsymbol{z}) \leq \sum_{i \in \text{supp}(\boldsymbol{z}^*)} \frac{(z_i^* - z_i)^2}{z_i} + \sum_{i \notin \text{supp}(\boldsymbol{z}^*)} z_i \leq \frac{1}{\min_{i \in \text{supp}(\boldsymbol{z}^*)} z_i} \|\boldsymbol{z}^* - \boldsymbol{z}\|_1.
$$

*Proof.* Using the definition of the Kullback-Leibler divergence, we have

$$\mathrm{KL}(\boldsymbol{x}^*, \boldsymbol{x}) = \sum_i x_i^* \ln\left(\frac{x_i^*}{x_i}\right) \leq \ln\left(\sum_i \frac{x_i^{*2}}{x_i}\right) = \ln\left(1 + \sum_i \frac{(x_i^* - x_i)^2}{x_i}\right) \leq \sum_i \frac{(x_i^* - x_i)^2}{x_i},$$

where the first inequality is by the concavity of the $\ln(\cdot)$ function, and the second inequality is because $\ln(1+u) \leq u$. Considering $i \in \mathrm{supp}(\boldsymbol{x}^*)$ and $i \notin \mathrm{supp}(\boldsymbol{x}^*)$ separately in the last summation, we have

$$\sum_i \frac{(x_i^* - x_i)^2}{x_i} = \sum_{i \in \mathrm{supp}(\boldsymbol{x}^*)} \frac{(x_i^* - x_i)^2}{x_i} + \sum_{i \notin \mathrm{supp}(\boldsymbol{x}^*)} \frac{(x_i)^2}{x_i} = \sum_{i \in \mathrm{supp}(\boldsymbol{x}^*)} \frac{(x_i^* - x_i)^2}{x_i} + \sum_{i \notin \mathrm{supp}(\boldsymbol{x}^*)} x_i.$$

The case for $\mathrm{KL}(\boldsymbol{y}^*, \boldsymbol{y})$ is similar. Combining both cases finishes the proof of the first inequality (recall that $\mathrm{KL}(\boldsymbol{z}^*, \boldsymbol{z})$ is defined as $\mathrm{KL}(\boldsymbol{x}^*, \boldsymbol{x}) + \mathrm{KL}(\boldsymbol{y}^*, \boldsymbol{y})$). The second inequality is straightforward:

$$\sum_{i \in \mathrm{supp}(\boldsymbol{z}^*)} \frac{(z_i^* - z_i)^2}{z_i} + \sum_{i \notin \mathrm{supp}(\boldsymbol{z}^*)} z_i \leq \frac{1}{\min_{i \in \mathrm{supp}(\boldsymbol{z}^*)} z_i}\left(\sum_{i \in \mathrm{supp}(\boldsymbol{z}^*)} |z_i^* - z_i| + \sum_{i \notin \mathrm{supp}(\boldsymbol{z}^*)} |z_i|\right)$$

$$= \frac{1}{\min_{i \in \mathrm{supp}(\boldsymbol{z}^*)} z_i} \|\boldsymbol{z}^* - \boldsymbol{z}\|_1.$$

$\square$

**Lemma 17.** *For $\eta \leq \frac{1}{8}$, OMWU guarantees $\frac{3}{4}\widehat{z}_{t,i} \leq z_{t,i} \leq \frac{4}{3}\widehat{z}_{t,i}$ and $\frac{3}{4}\widehat{z}_{t,i} \leq \widehat{z}_{t+1,i} \leq \frac{4}{3}\widehat{z}_{t,i}$.*

*Proof.* This is shown directly by the update of $\widehat{\boldsymbol{x}}_t$:

$$\frac{\widehat{x}_{t,i}\exp(-\eta)}{\exp(\eta)} \leq \widehat{x}_{t+1,i} = \frac{\widehat{x}_{t,i}\exp(-\eta \cdot (\boldsymbol{G}\boldsymbol{y}_t)_i)}{\sum_j \widehat{x}_{t,j}\exp(-\eta \cdot (\boldsymbol{G}\boldsymbol{y}_t)_j)} \leq \frac{\widehat{x}_{t,i}\exp(\eta)}{\exp(-\eta)}.$$

So by the condition on $\eta$, we have $\frac{3}{4}\widehat{x}_{t,i} \leq \exp(-2\eta) \cdot \widehat{x}_{t,i} \leq \widehat{x}_{t+1,i} \leq \exp(2\eta) \cdot \widehat{x}_{t,i} \leq \frac{4}{3}\widehat{x}_{t,i}$. The cases for $\boldsymbol{x}_t, \widehat{\boldsymbol{y}}_t$ and $\boldsymbol{y}_t$ are similar. $\square$

**Lemma 18.** *For any two probability vectors $\boldsymbol{u}, \boldsymbol{v}$, if for every entry $i$, $\frac{1}{2}u_i \leq v_i \leq \frac{3}{2}u_i$, then $\frac{1}{3}\sum_i \frac{(v_i - u_i)^2}{u_i} \leq \mathrm{KL}(\boldsymbol{u}, \boldsymbol{v}) \leq \sum_i \frac{(v_i - u_i)^2}{u_i} \leq \frac{1}{4}$.*

*Proof.* Using the definition of the Kullback-Leibler divergence, we have

$$\mathrm{KL}(\boldsymbol{u}, \boldsymbol{v}) = -\sum_i u_i \ln\frac{v_i}{u_i} \geq -\sum_i u_i\left(\frac{v_i - u_i}{u_i} - \frac{1}{3}\frac{(v_i - u_i)^2}{u_i^2}\right) = \frac{1}{3}\sum_i \frac{(v_i - u_i)^2}{u_i},$$

$$\mathrm{KL}(\boldsymbol{u}, \boldsymbol{v}) = -\sum_i u_i \ln\frac{v_i}{u_i} \leq -\sum_i u_i\left(\frac{v_i - u_i}{u_i} - \frac{(v_i - u_i)^2}{u_i^2}\right) = \sum_i \frac{(v_i - u_i)^2}{u_i} \leq \frac{1}{4},$$

where the first inequality is because $\ln(1+a) \leq a - \frac{1}{3}a^2$ for $-\frac{1}{2} \leq a \leq \frac{1}{2}$, and the second inequality is because $\ln(1+a) \geq a - a^2$ for $-\frac{1}{2} \leq a \leq \frac{1}{2}$. The third inequality is by using the condition $|u_i - v_i| \leq \frac{1}{2}u_i$. $\square$

**Lemma 19.** *For all $i \in \mathrm{supp}(\boldsymbol{z}^*)$ and $t$, OMWU guarantees $\widehat{z}_{t,i} \geq \epsilon$ ($\epsilon$ is defined in [Definition 4](#)).*

*Proof.* Using [Eq. (3)](#), we have

$$\mathrm{KL}(\boldsymbol{z}^*, \widehat{\boldsymbol{z}}_t) \leq \Theta_t \leq \cdots \leq \Theta_1 = \frac{1}{16}\mathrm{KL}(\widehat{\boldsymbol{z}}_1, \boldsymbol{z}_0) + \mathrm{KL}(\boldsymbol{z}^*, \widehat{\boldsymbol{z}}_1) = \mathrm{KL}(\boldsymbol{z}^*, \widehat{\boldsymbol{z}}_1), \qquad (15)$$

where the last equality is because $\widehat{\boldsymbol{z}}_1 = \boldsymbol{z}_0 = (\frac{\boldsymbol{1}_M}{M}, \frac{\boldsymbol{1}_N}{N})$.

Then, for any $i \in \operatorname{supp}(\boldsymbol{z}^*)$, we have

$$z_i^* \ln \frac{1}{\widehat{z}_{t,i}} \le \sum_j z_j^* \ln \frac{1}{\widehat{z}_{t,j}} = \mathrm{KL}(\boldsymbol{z}^*, \widehat{\boldsymbol{z}}_t) - \sum_j z_j^* \ln z_j^* \le \mathrm{KL}(\boldsymbol{z}^*, \widehat{\boldsymbol{z}}_1) - \sum_j z_j^* \ln z_j^*$$

$$= \sum_j z_j^* \ln \frac{1}{\widehat{z}_{1,j}} = \ln(MN).$$

Therefore, we conclude for all $t$ and $i \in \operatorname{supp}(\boldsymbol{z}^*)$, $\widehat{z}_{t,i}$ satisfies

$$\widehat{z}_{t,i} \ge \exp\left(-\frac{\ln(MN)}{z_i^*}\right) \ge \min_{j \in \operatorname{supp}(\boldsymbol{z}^*)} \exp\left(-\frac{\ln(MN)}{z_j^*}\right) = \epsilon.$$

$\square$

### D.3 PROOFS OF LEMMA 2 AND THEOREM 3

*Proof of Lemma 2.* Below we consider any $\boldsymbol{z}' \in \mathcal{Z}$ such that $\operatorname{supp}(\boldsymbol{z}') \subseteq \operatorname{supp}(\boldsymbol{z}^*)$, that is, $\boldsymbol{z}' \in \mathcal{V}^*(\mathcal{Z})$. Considering Eq. (1), and using the first-order optimality condition of $\widehat{\boldsymbol{z}}_{t+1}$, we have

$$(\nabla \psi(\widehat{\boldsymbol{z}}_{t+1}) - \nabla \psi(\widehat{\boldsymbol{z}}_t) + \eta F(\boldsymbol{z}_t))^\top (\boldsymbol{z}' - \widehat{\boldsymbol{z}}_{t+1}) \ge 0,$$

where $\psi(\boldsymbol{z}) = \sum_i z_i \ln z_i$. Rearranging the terms and we get

$$\eta F(\boldsymbol{z}_t)^\top (\widehat{\boldsymbol{z}}_{t+1} - \boldsymbol{z}') \le (\nabla \psi(\widehat{\boldsymbol{z}}_{t+1}) - \nabla \psi(\widehat{\boldsymbol{z}}_t))^\top (\boldsymbol{z}' - \widehat{\boldsymbol{z}}_{t+1}) = \sum_i (z_i' - \widehat{z}_{t+1,i}) \ln \frac{\widehat{z}_{t+1,i}}{\widehat{z}_{t,i}}.$$

$$(16)$$

The left hand side of Eq. (16) is lower bounded as

$$\eta F(\boldsymbol{z}_t)^\top (\widehat{\boldsymbol{z}}_{t+1} - \boldsymbol{z}') = \eta F(\widehat{\boldsymbol{z}}_{t+1})^\top (\widehat{\boldsymbol{z}}_{t+1} - \boldsymbol{z}') + \eta (F(\boldsymbol{z}_t) - F(\widehat{\boldsymbol{z}}_{t+1}))^\top (\widehat{\boldsymbol{z}}_{t+1} - \boldsymbol{z}')$$

$$\ge \eta F(\widehat{\boldsymbol{z}}_{t+1})^\top (\widehat{\boldsymbol{z}}_{t+1} - \boldsymbol{z}') - \eta \|F(\boldsymbol{z}_t) - F(\widehat{\boldsymbol{z}}_{t+1})\|_\infty \|\widehat{\boldsymbol{z}}_{t+1} - \boldsymbol{z}'\|_1$$

$$\ge \eta F(\widehat{\boldsymbol{z}}_{t+1})^\top (\widehat{\boldsymbol{z}}_{t+1} - \boldsymbol{z}') - 4\eta \|\boldsymbol{z}_t - \widehat{\boldsymbol{z}}_{t+1}\|_1$$
$$(\|F(\boldsymbol{z}_t) - F(\widehat{\boldsymbol{z}}_{t+1})\|_\infty \le \|\boldsymbol{z}_t - \widehat{\boldsymbol{z}}_{t+1}\|_1 \le 4)$$

$$\ge \eta F(\widehat{\boldsymbol{z}}_{t+1})^\top (\widehat{\boldsymbol{z}}_{t+1} - \boldsymbol{z}') - \frac{1}{2} \|\boldsymbol{z}_t - \widehat{\boldsymbol{z}}_{t+1}\|_1; \qquad (\eta \le 1/8)$$

on the other hand, the right hand side of Eq. (16) is upper bounded by

$$\sum_i (z_i' - \widehat{z}_{t+1,i}) \ln \frac{\widehat{z}_{t+1,i}}{\widehat{z}_{t,i}} = \sum_{i \in \operatorname{supp}(\boldsymbol{z}^*)} z_i' \ln \frac{\widehat{z}_{t+1,i}}{\widehat{z}_{t,i}} - \mathrm{KL}(\widehat{\boldsymbol{z}}_{t+1}, \widehat{\boldsymbol{z}}_t) \qquad (\operatorname{supp}(\boldsymbol{z}') \subseteq \operatorname{supp}(\boldsymbol{z}^*))$$

$$\le \sum_{i \in \operatorname{supp}(\boldsymbol{z}^*)} \left| \ln \frac{\widehat{z}_{t+1,i}}{\widehat{z}_{t,i}} \right|$$

$$= \sum_{i \in \operatorname{supp}(\boldsymbol{z}^*)} \max\left\{ \ln\left(1 + \frac{\widehat{z}_{t+1,i} - \widehat{z}_{t,i}}{\widehat{z}_{t,i}}\right), \ln\left(1 + \frac{\widehat{z}_{t,i} - \widehat{z}_{t+1,i}}{\widehat{z}_{t+1,i}}\right) \right\}$$

$$\le \sum_{i \in \operatorname{supp}(\boldsymbol{z}^*)} \ln\left(1 + \frac{|\widehat{z}_{t+1,i} - \widehat{z}_{t,i}|}{\min\{\widehat{z}_{t+1,i}, \widehat{z}_{t,i}\}}\right)$$

$$\le \frac{4}{3} \sum_{i \in \operatorname{supp}(\boldsymbol{z}^*)} \frac{|\widehat{z}_{t+1,i} - \widehat{z}_{t,i}|}{\widehat{z}_{t,i}}. \qquad (\ln(1+a) \le a \text{ and Lemma 17})$$

Combining the bounds on the two sides of Eq. (16), we get

$$\eta F(\widehat{\boldsymbol{z}}_{t+1})^\top (\widehat{\boldsymbol{z}}_{t+1} - \boldsymbol{z}') \le \frac{4}{3} \sum_{i \in \operatorname{supp}(\boldsymbol{z}^*)} \frac{|\widehat{z}_{t+1,i} - \widehat{z}_{t,i}|}{\widehat{z}_{t,i}} + \frac{1}{2} \|\boldsymbol{z}_t - \widehat{\boldsymbol{z}}_{t+1}\|_1.$$

Since $z'$ can be chosen as any point in $\mathcal{V}^*(\mathcal{Z})$, we further lower bound the left-hand side above using Lemma 15 and get

$$\eta C \|z^* - \widehat{z}_{t+1}\|_1 \leq \frac{4}{3} \sum_{i \in \mathrm{supp}(z^*)} \frac{|\widehat{z}_{t+1,i} - \widehat{z}_{t,i}|}{\widehat{z}_{t,i}} + \frac{1}{2} \|z_t - \widehat{z}_{t+1}\|_1$$

$$\leq \frac{4}{3\epsilon} \|\widehat{z}_{t+1} - \widehat{z}_t\|_1 + \frac{1}{2} \|z_t - \widehat{z}_{t+1}\|_1, \qquad \text{(Lemma 19)}$$

$$\leq \frac{4}{3\epsilon} \left( \|\widehat{z}_{t+1} - \widehat{z}_t\|_1 + \|z_t - \widehat{z}_{t+1}\|_1 \right) \qquad (17)$$

where the last inequality uses $\epsilon \leq 1$. With the help of Eq. (17), below we prove the desired inequalities.

**Case 1. General case.**

$$\mathrm{KL}(\widehat{z}_{t+1}, z_t) + \mathrm{KL}(z_t, \widehat{z}_t)$$

$$\geq \frac{1}{2} \|\widehat{x}_{t+1} - x_t\|_1^2 + \frac{1}{2} \|\widehat{y}_{t+1} - y_t\|_1^2 + \frac{1}{2} \|x_t - \widehat{x}_t\|_1^2 + \frac{1}{2} \|y_t - \widehat{y}_t\|_1^2 \quad \text{(Pinsker's inequality)}$$

$$\geq \frac{1}{4} \|\widehat{z}_{t+1} - z_t\|_1^2 + \frac{1}{4} \|z_t - \widehat{z}_t\|_1^2 \qquad (a^2 + b^2 \geq \tfrac{1}{2}(a+b)^2)$$

$$\geq \frac{1}{16} \|\widehat{z}_{t+1} - z_t\|_1^2 + \frac{1}{8} \left( \|\widehat{z}_{t+1} - z_t\|_1^2 + \|z_t - \widehat{z}_t\|_1^2 \right)$$

$$\geq \frac{1}{16} \|\widehat{z}_{t+1} - z_t\|_1^2 + \frac{1}{16} \|\widehat{z}_{t+1} - \widehat{z}_t\|_1^2 \qquad (a^2 + b^2 \geq \tfrac{1}{2}(a+b)^2 \text{ and triangle inequality})$$

$$\geq \frac{1}{32} \left( \|\widehat{z}_{t+1} - z_t\|_1 + \|\widehat{z}_{t+1} - \widehat{z}_t\|_1 \right)^2 \qquad (a^2 + b^2 \geq \tfrac{1}{2}(a+b)^2)$$

$$\geq \frac{1}{32} \left( \frac{3\epsilon\eta C}{4} \right)^2 \|z^* - \widehat{z}_{t+1}\|_1^2 \qquad \text{(Eq. (17))}$$

$$\geq \frac{\epsilon^2 \eta^2 C^2}{64} \times \epsilon^2 \mathrm{KL}(z^*, \widehat{z}_{t+1})^2 = \frac{\epsilon^4 \eta^2 C^2}{64} \mathrm{KL}(z^*, \widehat{z}_{t+1})^2. \qquad \text{(Lemma 16 and Lemma 19)}$$

This proves the first part of the lemma with $C_1 = \epsilon^4 C^2 / 64$.

**Case 2. The case when** $\max\{\|z^* - \widehat{z}_t\|_1, \|z^* - z_t\|_1\} \leq \frac{\eta\xi}{10}$.

$$\mathrm{KL}(\widehat{z}_{t+1}, z_t) + \mathrm{KL}(z_t, \widehat{z}_t)$$

$$\geq \frac{1}{3} \sum_i \left( \frac{(\widehat{z}_{t+1,i} - z_{t,i})^2}{\widehat{z}_{t+1,i}} + \frac{(z_{t,i} - \widehat{z}_{t,i})^2}{z_{t,i}} \right) \qquad \text{(Lemma 17 and Lemma 18)}$$

$$\geq \frac{1}{4} \sum_{i \notin \mathrm{supp}(z^*)} \left( \frac{(\widehat{z}_{t+1,i} - z_{t,i})^2}{\widehat{z}_{t,i}} + \frac{(z_{t,i} - \widehat{z}_{t,i})^2}{\widehat{z}_{t,i}} \right) \qquad \text{(Lemma 17)}$$

$$\geq \frac{1}{8} \sum_{i \notin \mathrm{supp}(z^*)} \frac{(\widehat{z}_{t+1,i} - \widehat{z}_{t,i})^2}{\widehat{z}_{t,i}}. \qquad (18)$$

Below we continue to bound $\sum_{i \notin \mathrm{supp}(z^*)} \frac{(\widehat{z}_{t+1,i} - \widehat{z}_{t,i})^2}{\widehat{z}_{t,i}}$.

By the assumption, we have $\|y_t - y^*\|_1 \leq \frac{\eta\xi}{10}$, which by Lemma 13 and Definition 2 implies

$$\forall i \in \mathrm{supp}(x^*), \qquad (Gy_t)_i \leq (Gy^*)_i + \frac{\eta\xi}{10} = \rho + \frac{\eta\xi}{10} \leq \rho + \frac{\xi}{10},$$

$$\forall i \notin \mathrm{supp}(x^*), \qquad (Gy_t)_i \geq (Gy^*)_i - \frac{\eta\xi}{10} \geq \rho + \xi - \frac{\eta\xi}{10} \geq \rho + \frac{9\xi}{10}.$$

We also have $\|\widehat{\boldsymbol{x}}_t - \boldsymbol{x}^*\|_1 \leq \frac{\eta\xi}{10}$, so $\sum_{j \notin \text{supp}(\boldsymbol{x}^*)} \widehat{x}_{t,j} \leq \frac{\eta\xi}{10}$. Then, for $i \notin \text{supp}(\boldsymbol{x}^*)$, we have

$$
\begin{aligned}
\widehat{x}_{t+1,i} &= \frac{\widehat{x}_{t,i} \exp(-\eta(\boldsymbol{G}\boldsymbol{y}_t)_i)}{\sum_j \widehat{x}_{t,j} \exp(-\eta(\boldsymbol{G}\boldsymbol{y}_t)_j)} \\
&\leq \frac{\widehat{x}_{t,i} \exp(-\eta(\boldsymbol{G}\boldsymbol{y}_t)_i)}{\sum_{j \in \text{supp}(\boldsymbol{x}^*)} \widehat{x}_{t,j} \exp(-\eta(\boldsymbol{G}\boldsymbol{y}_t)_j)} \\
&\leq \frac{\widehat{x}_{t,i} \exp(-\eta(\rho + \frac{9\xi}{10}))}{\sum_{j \in \text{supp}(\boldsymbol{x}^*)} \widehat{x}_{t,j} \exp(-\eta(\rho + \frac{\xi}{10}))} \\
&= \frac{\widehat{x}_{t,i} \exp\left(-\frac{8}{10}\eta\xi\right)}{\left(1 - \sum_{j \notin \text{supp}(\boldsymbol{x}^*)} \widehat{x}_{t,j}\right)} \\
&\leq \frac{\widehat{x}_{t,i} \exp\left(-\frac{8}{10}\eta\xi\right)}{\left(1 - \frac{\eta\xi}{10}\right)} \leq \widehat{x}_{t,i}\left(1 - \frac{1}{2}\eta\xi\right),
\end{aligned}
$$

where the last inequality is because $\frac{\exp(-0.8u)}{1-0.1u} \leq 1 - 0.5u$ for $u \in [0,1]$. Rearranging gives

$$
\frac{|\widehat{x}_{t+1,i} - \widehat{x}_{t,i}|^2}{\widehat{x}_{t,i}} \geq \frac{\eta^2\xi^2}{4}\widehat{x}_{t,i} \geq \frac{\eta^2\xi^2}{8}\widehat{x}_{t+1,i},
$$

where the last step uses [Lemma 17](). The case for $\widehat{\boldsymbol{y}}_t$ is similar, so we have

$$
\frac{|\widehat{z}_{t+1,i} - \widehat{z}_{t,i}|^2}{\widehat{z}_{t,i}} \geq \frac{\eta^2\xi^2}{8}\widehat{z}_{t+1,i}.
$$

Combining this with [Eq. (18)](), we get

$$
\text{KL}(\widehat{\boldsymbol{z}}_{t+1}, \boldsymbol{z}_t) + \text{KL}(\boldsymbol{z}_t, \widehat{\boldsymbol{z}}_t) \geq \frac{\eta^2\xi^2}{64} \sum_{i \notin \text{supp}(\boldsymbol{z}^*)} \widehat{z}_{t+1,i}. \tag{19}
$$

Now we combine two lower bounds of $\text{KL}(\widehat{\boldsymbol{z}}_{t+1}, \boldsymbol{z}_t) + \text{KL}(\boldsymbol{z}_t, \widehat{\boldsymbol{z}}_t)$. Using an intermediate step in **Case 1**, and [Eq. (19)](), we get

$$
\begin{aligned}
\text{KL}(\widehat{\boldsymbol{z}}_{t+1}, \boldsymbol{z}_t) + \text{KL}(\boldsymbol{z}_t, \widehat{\boldsymbol{z}}_t) &= \frac{1}{2}\left(\text{KL}(\widehat{\boldsymbol{z}}_{t+1}, \boldsymbol{z}_t) + \text{KL}(\boldsymbol{z}_t, \widehat{\boldsymbol{z}}_t)\right) + \frac{1}{2}\left(\text{KL}(\widehat{\boldsymbol{z}}_{t+1}, \boldsymbol{z}_t) + \text{KL}(\boldsymbol{z}_t, \widehat{\boldsymbol{z}}_t)\right) \\
&\geq \frac{\epsilon^2\eta^2 C^2}{128}\|\boldsymbol{z}^* - \widehat{\boldsymbol{z}}_{t+1}\|_1^2 + \frac{\eta^2\xi^2}{128} \sum_{i \notin \text{supp}(\boldsymbol{z}^*)} \widehat{z}_{t+1,i} \\
&= \frac{\epsilon^3\eta^2 C^2\xi^2}{128}\left(\frac{1}{\xi^2\epsilon}\|\widehat{\boldsymbol{z}}_{t+1} - \boldsymbol{z}^*\|_1^2 + \frac{1}{\epsilon^3 C^2} \sum_{i \notin \text{supp}(\boldsymbol{z}^*)} \widehat{z}_{t+1,i}\right) \\
&\geq \frac{\epsilon^3\eta^2 C^2\xi^2}{128}\left(\frac{1}{\epsilon}\|\widehat{\boldsymbol{z}}_{t+1} - \boldsymbol{z}^*\|_1^2 + \sum_{i \notin \text{supp}(\boldsymbol{z}^*)} \widehat{z}_{t+1,i}\right) \\
&\qquad\qquad\qquad\qquad\qquad\qquad\qquad\qquad (\xi \leq 1, C \leq 1, \text{ and } \epsilon \leq 1) \\
&\geq \frac{\epsilon^3\eta^2 C^2\xi^2}{128}\text{KL}(\boldsymbol{z}^*, \widehat{\boldsymbol{z}}_{t+1}). \qquad\qquad \text{([Lemma 16]() and [Lemma 19]())}
\end{aligned}
$$

This proves the second part of the lemma with $C_2 = \epsilon^3 C^2\xi^2/128$. $\qquad\square$

Now we are ready to prove [Theorem 3]().

*Proof of [Theorem 3]().* As argued in [Section 4](), with $\Theta_t = \text{KL}(\boldsymbol{z}^*, \widehat{\boldsymbol{z}}_t) + \frac{1}{16}\text{KL}(\widehat{\boldsymbol{z}}_t, \boldsymbol{z}_{t-1})$ and $\zeta_t = \text{KL}(\widehat{\boldsymbol{z}}_{t+1}, \boldsymbol{z}_t) + \text{KL}(\boldsymbol{z}_t, \widehat{\boldsymbol{z}}_t)$, we have (see [Eq. (3)]())

$$
\Theta_{t+1} \leq \Theta_t - \tfrac{15}{16}\zeta_t.
$$

We the proceed as,

$$\zeta_t \geq \frac{1}{2}\text{KL}(\widehat{z}_{t+1}, z_t) + \frac{1}{2}\zeta_t$$

$$\geq \frac{1}{2}\text{KL}(\widehat{z}_{t+1}, z_t) + \frac{\eta^2 C_1}{2}\text{KL}(z^*, \widehat{z}_{t+1})^2 \qquad \text{(Lemma 2)}$$

$$\geq 2\text{KL}(\widehat{z}_{t+1}, z_t)^2 + \frac{\eta^2 C_1}{2}\text{KL}(z^*, \widehat{z}_{t+1})^2 \qquad \text{(by Lemma 17 and Lemma 18)}$$

$$\geq \frac{\eta^2 C_1}{2}\left(\text{KL}(\widehat{z}_{t+1}, z_t)^2 + \text{KL}(z^*, \widehat{z}_{t+1})^2\right)$$

$$(C_1 = \epsilon^4 C^2/64 \leq 1/64 \text{ as shown in the proof of Lemma 2})$$

$$\geq \frac{\eta^2 C_1}{4}\left(\text{KL}(\widehat{z}_{t+1}, z_t) + \text{KL}(z^*, \widehat{z}_{t+1})\right)^2$$

$$\geq \frac{\eta^2 C_1}{4}\Theta_{t+1}^2.$$

Therefore, $\Theta_{t+1} \leq \Theta_t - \frac{15\eta^2 C_1}{64}\Theta_{t+1}^2 \leq \Theta_t - \frac{15\eta^2 C_1}{64+\ln MN}\Theta_{t+1}^2$. Also, recall $\widehat{z}_1 = z_0 = \left(\frac{\mathbf{1}_M}{M}, \frac{\mathbf{1}_N}{N}\right)$ and thus $\Theta_1 = \text{KL}(z^*, \widehat{z}_1) \leq \ln(MN)$. Therefore, the conditions of Lemma 12 are satisfied with $p = 1$ and $q = \frac{15\eta^2 C_1}{64+\ln(MN)}$, and we conclude that

$$\Theta_t \leq \frac{C'}{t},$$

where $C' = \max\left\{\ln(MN), \frac{128+2\ln(MN)}{15\eta^2 C_1}\right\} = \frac{128+2\ln(MN)}{15\eta^2 C_1}$.

Next we prove the main result. Set $T_0 = \frac{12800C'}{\eta^2\xi^2}$. For $t \geq T_0$, we have using Pinsker's inequality,

$$\|z^* - \widehat{z}_t\|_1^2 \leq 2\|x^* - \widehat{x}_t\|_1^2 + 2\|y^* - \widehat{y}_t\|_1^2 \leq 4\text{KL}(z^*, \widehat{z}_t) \leq \frac{4C'}{T_0} \leq \frac{\eta^2\xi^2}{100},$$

$$\|z^* - z_t\|_1^2 \leq 2\|z^* - \widehat{z}_{t+1}\|_1^2 + 2\|\widehat{z}_{t+1} - z_t\|_1^2$$

$$\leq 4\|x^* - \widehat{x}_{t+1}\|_1^2 + 4\|\widehat{x}_{t+1} - x_t\|_1^2 + 4\|y^* - \widehat{y}_{t+1}\|_1^2 + 4\|\widehat{y}_{t+1} - y_t\|_1^2$$

$$\leq 8\text{KL}(z^*, \widehat{z}_{t+1}) + 8\text{KL}(\widehat{z}_{t+1}, z_t)$$

$$\leq 128\Theta_{t+1} \leq \frac{128C'}{T_0} \leq \frac{\eta^2\xi^2}{100}.$$

Therefore, when $t \geq T_0$, the condition of the second part of Lemma 2 is satisfied, and we have

$$\zeta_t \geq \frac{1}{2}\text{KL}(\widehat{z}_{t+1}, z_t) + \frac{1}{2}\zeta_t$$

$$\geq \frac{1}{2}\text{KL}(\widehat{z}_{t+1}, z_t) + \frac{\eta^2 C_2}{2}\text{KL}(z^*, \widehat{z}_{t+1}) \qquad \text{(by Lemma 2)}$$

$$\geq \frac{\eta^2 C_2}{2}\Theta_{t+1}. \qquad (C_2 = \epsilon^3 C^2\xi^2/128 \leq 1/128 \text{ as shown in the proof of Lemma 2})$$

Therefore, when $t \geq T_0$, $\Theta_{t+1} \leq \Theta_t - \frac{15\eta^2 C_2}{32}\Theta_{t+1}$, which further leads to

$$\Theta_t \leq \Theta_{T_0} \cdot \left(1 + \frac{15\eta^2 C_2}{32}\right)^{T_0 - t} \leq \Theta_1 \cdot \left(1 + \frac{15\eta^2 C_2}{32}\right)^{T_0 - t} \leq \ln(MN)\left(1 + \frac{15\eta^2 C_2}{32}\right)^{T_0 - t}.$$

where the second inequality uses Eq. (15). The inequality trivially holds for $t < T_0$ as well, so it holds for all $t$.

We finish the proof by relating $\text{KL}(z^*, z_t)$ and $\Theta_{t+1}$. Note that by Lemma 16, Lemma 17, and Lemma 19, we have

$$\text{KL}(z^*, z_t)^2 \leq \frac{\|z^* - z_t\|^2}{\min_{i\in\text{supp}(z^*)} z_{t,i}^2} \leq \frac{16\|z^* - z_t\|^2}{9\epsilon^2} \leq 4\left(\frac{\|z^* - \widehat{z}_{t+1}\|^2 + \|\widehat{z}_{t+1} - z_t\|^2}{\epsilon^2}\right).$$

We continue to bound the last term as

$$4\left(\frac{\|z^* - \widehat{z}_{t+1}\|^2 + \|\widehat{z}_{t+1} - z_t\|^2}{\epsilon^2}\right)$$

$$= 4\left(\frac{\|x^* - \widehat{x}_{t+1}\|^2 + \|y^* - \widehat{y}_{t+1}\|^2 + \|\widehat{x}_{t+1} - x_t\|^2 + \|\widehat{y}_{t+1} - y_t\|^2}{\epsilon^2}\right)$$

$$= 4\left(\frac{\|x^* - \widehat{x}_{t+1}\|_1^2 + \|y^* - \widehat{y}_{t+1}\|_1^2 + \|\widehat{x}_{t+1} - x_t\|_1^2 + \|\widehat{y}_{t+1} - y_t\|_1^2}{\epsilon^2}\right) \quad (\|x\|_2 \le \|x\|_1)$$

$$\le \frac{128}{\epsilon^2}\left(\frac{\mathrm{KL}(z^*, \widehat{z}_{t+1})}{16} + \frac{\mathrm{KL}(\widehat{z}_{t+1}, z_t)}{16}\right) \qquad \text{(Pinsker's inequality)}$$

$$\le \frac{128}{\epsilon^2}\Theta_{t+1}.$$

Combining everything, we get

$$\mathrm{KL}(z^*, z_t) \le \frac{\sqrt{128}}{\epsilon}\sqrt{\Theta_{t+1}} \le \frac{\sqrt{128\ln(MN)}}{\epsilon}\left(1 + \frac{15\eta^2 C_2}{32}\right)^{\frac{T_0 - t - 1}{2}},$$

which completes the proof. $\qquad\square$

## E  PROOFS OF LEMMA 4 AND THE SUM-OF-DUALITY-GAP BOUND

*Proof of Lemma 4.* Below we consider any $z' \ne \widehat{z}_{t+1} \in \mathcal{Z}$. Considering Eq. (1) with $D_\psi(u, v) = \frac{1}{2}\|u - v\|^2$, and using the first-order optimality condition of $\widehat{z}_{t+1}$, we have

$$(\widehat{z}_{t+1} - \widehat{z}_t + \eta F(z_t))^\top (z' - \widehat{z}_{t+1}) \ge 0,$$
$$(z_{t+1} - \widehat{z}_{t+1} + \eta F(z_t))^\top (z' - z_{t+1}) \ge 0.$$

Rearranging the terms and we get

$$(\widehat{z}_{t+1} - \widehat{z}_t)^\top (z' - \widehat{z}_{t+1}) \ge \eta F(z_t)^\top (\widehat{z}_{t+1} - z')$$
$$= \eta F(\widehat{z}_{t+1})^\top (\widehat{z}_{t+1} - z') + \eta (F(z_t) - F(\widehat{z}_{t+1}))^\top (\widehat{z}_{t+1} - z')$$
$$\ge \eta F(\widehat{z}_{t+1})^\top (\widehat{z}_{t+1} - z') - \eta L\|z_t - \widehat{z}_{t+1}\|\|\widehat{z}_{t+1} - z'\|$$
$$\ge \eta F(\widehat{z}_{t+1})^\top (\widehat{z}_{t+1} - z') - \frac{1}{8}\|z_t - \widehat{z}_{t+1}\|\|\widehat{z}_{t+1} - z'\|,$$

and

$$(z_{t+1} - \widehat{z}_{t+1})^\top (z' - z_{t+1}) \ge \eta F(z_t)^\top (z_{t+1} - z')$$
$$= \eta F(z_{t+1})^\top (z_{t+1} - z') + \eta (F(z_t) - F(z_{t+1}))^\top (z_{t+1} - z')$$
$$\ge \eta F(z_{t+1})^\top (z_{t+1} - z') - \eta L\|z_t - z_{t+1}\|\|z_{t+1} - z'\|$$
$$\ge \eta F(z_{t+1})^\top (z_{t+1} - z') - \frac{1}{8}\|z_t - z_{t+1}\|\|z_{t+1} - z'\|.$$

Here, for both block, the third step uses Hölder's inequality and the smoothness condition Assumption 1, and the last step uses the condition $\eta \le 1/(8L)$. Upper bounding the left-hand side of the two inequalities by $\|\widehat{z}_{t+1} - \widehat{z}_t\|\|\widehat{z}_{t+1} - z'\|$ and $\|z_{t+1} - \widehat{z}_{t+1}\|\|z_{t+1} - z'\|$ respectively and then rearranging, we get

$$\|\widehat{z}_{t+1} - z'\|\left(\|\widehat{z}_{t+1} - \widehat{z}_t\| + \frac{1}{8}\|z_t - \widehat{z}_{t+1}\|\right) \ge \eta F(\widehat{z}_{t+1})^\top (\widehat{z}_{t+1} - z'),$$

$$\|z_{t+1} - z'\|\left(\|z_{t+1} - \widehat{z}_{t+1}\| + \frac{1}{8}\|z_t - z_{t+1}\|\right) \ge \eta F(z_{t+1})^\top (z_{t+1} - z').$$

Therefore, we have

$$\left(\|\widehat{z}_{t+1} - \widehat{z}_t\| + \frac{1}{8}\|z_t - \widehat{z}_{t+1}\|\right)^2 \ge \frac{\eta^2 [F(\widehat{z}_{t+1})^\top (\widehat{z}_{t+1} - z')]_+^2}{\|\widehat{z}_{t+1} - z'\|^2},$$

$$\left(\|z_{t+1} - \widehat{z}_{t+1}\| + \frac{1}{8}\|z_t - z_{t+1}\|\right)^2 \ge \frac{\eta^2 [F(z_{t+1})^\top (z_{t+1} - z')]_+^2}{\|z_{t+1} - z'\|^2}.$$

Finally, by the triangle inequality and the fact $(a + b)^2 \leq 2a^2 + 2b^2$, we have

$$
\begin{aligned}
\left( \|\widehat{z}_{t+1} - \widehat{z}_t\| + \frac{1}{8}\|z_t - \widehat{z}_{t+1}\| \right)^2 &\leq \left( \|z_t - \widehat{z}_t\| + \frac{9}{8}\|z_t - \widehat{z}_{t+1}\| \right)^2 \\
&\leq \left( \frac{9}{8}\|z_t - \widehat{z}_t\| + \frac{9}{8}\|z_t - \widehat{z}_{t+1}\| \right)^2 \\
&\leq \frac{81}{32} \left( \|z_t - \widehat{z}_t\|^2 + \|z_t - \widehat{z}_{t+1}\|^2 \right),
\end{aligned}
$$

$$
\begin{aligned}
\left( \|\widehat{z}_{t+1} - z_{t+1}\| + \frac{1}{8}\|z_t - z_{t+1}\| \right)^2 &\leq \left( \frac{9}{8}\|z_{t+1} - \widehat{z}_{t+1}\| + \|z_t - \widehat{z}_{t+1}\| \right)^2 \\
&\leq \left( \frac{9}{8}\|z_{t+1} - \widehat{z}_{t+1}\| + \frac{9}{8}\|z_t - \widehat{z}_{t+1}\| \right)^2 \\
&\leq \frac{81}{32} \left( \|z_{t+1} - \widehat{z}_{t+1}\|^2 + \|z_t - \widehat{z}_{t+1}\|^2 \right),
\end{aligned}
$$

which finishes the proof. $\qquad \square$

Next, we use Eq. (4) and Eq. (6) to derive a result on the convergence of "average duality gap" across time. First, we use the following lemma to relate the right-hand side of Eq. (6) to the duality gap of $z_t$.

**Lemma 20.** *Let $\mathcal{Z}$ be closed and bounded. Then for any $z \in \mathcal{Z}$, we have $\alpha_f(z) \leq \max_{z' \in \mathcal{Z}} F(z)^\top (z - z')$.*

*Proof.* This is a direct consequence of the convexity of $f(\cdot, y)$ and the concavity of $f(x, \cdot)$:

$$
\begin{aligned}
\alpha_f(z) &= \max_{(x', y') \in \mathcal{X} \times \mathcal{Y}} \left( f(x, y') - f(x, y) + f(x, y) - f(x', y) \right) \\
&\leq \max_{(x', y') \in \mathcal{X} \times \mathcal{Y}} \left( \nabla_y f(x, y)^\top (y' - y) + \nabla_x f(x, y)^\top (x - x') \right) = \max_{z' \in \mathcal{Z}} F(z)^\top (z - z').
\end{aligned}
$$

$\qquad \square$

With Lemma 20, the following theorem can be proven straightforwardly.

**Theorem 21.** *Let $\mathcal{Z}$ be closed and bounded. Then OGDA with $\eta \leq \frac{1}{8L}$ ensures $\frac{1}{T}\sum_{t=1}^T \alpha_f(z_t) = O\left( \frac{D}{\eta\sqrt{T}} \right)$ for any $T$, where $D \triangleq \sup_{z, z' \in \mathcal{Z}} \|z - z'\|$.*

*Proof.* We first bound the sum of squared duality gap as (recall $\zeta_t = \|\widehat{z}_{t+1} - z_t\|^2 + \|z_t - \widehat{z}_t\|^2$):

$$
\begin{aligned}
\sum_{t=1}^T \alpha_f(z_t)^2 &\leq \sum_{t=1}^T \left( \max_{z' \in \mathcal{Z}} F(z_t)^\top (z_t - z') \right)^2 && \text{(Lemma 20)} \\
&\leq \frac{81}{32\eta^2} \sum_{t=1}^T (\zeta_{t-1} + \zeta_t)\|z_t - z'\|^2 && \text{(Lemma 4)} \\
&\leq \mathcal{O}\left( \frac{D^2}{\eta^2} \sum_{t=2}^T (\Theta_{t-1} - \Theta_t + \Theta_t - \Theta_{t+1}) \right) && \text{(Eq. (4))} \\
&= \mathcal{O}\left( \frac{D^2}{\eta^2} \right). && \text{(telescoping)}
\end{aligned}
$$

Finally, by Cauchy-Schwarz inequality, we get $\frac{1}{T}\sum_{t=1}^T \alpha_f(z_t) \leq \frac{1}{T}\sqrt{T \sum_{t=1}^T \alpha_f(z_t)^2} = \mathcal{O}\left( \frac{D}{\eta\sqrt{T}} \right)$. $\qquad \square$

This theorem indicates that $\alpha_f(z_t)$ is converging to zero. A rate of $\alpha_f(z_t) = \mathcal{O}(\frac{D}{\eta\sqrt{t}})$ would be compatible with the theorem, but is not directly implied by it. In a recent work, Golowich et al. (2020b) consider the unconstrained setting and show that the extra-gradient algorithm obtains the rate $\alpha_f(z_t) = \mathcal{O}(\frac{D}{\eta\sqrt{t}})$, under an extra assumption that the Hessian of $f$ is also Lipschitz (since Golowich et al. (2020b) study the unconstrained setting, their duality gap $\alpha_f$ is defined only with respect to the best responses that lie within a ball of radius $D$ centered around the equilibrium). Note that the extra-gradient algorithm requires more cooperation between the two players compared to OGDA and is less suitable for a repeated game setting.

## F  THE EQUIVALENCE BETWEEN SP-MS AND METRIC SUBREGULARITY

In this section, we formally that show our SP-MS condition with $\beta = 0$ is equivalent to metric subregularity. Before introducing the main theorem, we introduce several definitions. We let $\mathcal{Z}^* \subseteq \mathcal{Z} \subseteq \mathbb{R}^K$ ($\mathcal{Z}^*$ and $\mathcal{Z}$ follow the same definitions as in our main text). First, we define the element-to-set distance function $d$:

**Definition 5.** *The element-to-set distance function $d$: $\mathbb{R}^K \times 2^{\mathbb{R}^K} \to \mathbb{R}$ is defined as $d(z, \mathcal{S}) = \inf_{z' \in \mathcal{S}} \|z - z'\|$.*

The definition of metric subregularity involves a set-valued operator $\mathcal{T} : \mathcal{Z} \to 2^{\mathbb{R}^K}$, which maps an element of $\mathcal{Z}$ to a set in $\mathbb{R}^K$.

**Definition 6.** *A set-valued operator $\mathcal{T}$ is called* metric subregular *at $(\bar{z}, v)$ for $v \in \mathcal{T}(\bar{z})$ if there exists $\kappa > 0$ and a neighborhood $\Omega$ of $\bar{z}$ such that*

$$d(v, \mathcal{T}(z)) \geq \kappa d(z, \mathcal{T}^{-1}(v))$$

*for all $z \in \Omega$, where $x \in \mathcal{T}^{-1}(v) \Leftrightarrow v \in \mathcal{T}(x)$. If $\Omega = \mathcal{Z}$, we call $\mathcal{T}$ globally metric subregular.*

The following definition of *normal cone* is also required in the analysis:

**Definition 7.** *The normal cone of $\mathcal{Z}$ at point $z$ is $\mathcal{N}(z) = \{g \mid g^\top(z' - z) \leq 0, \ \forall z' \in \mathcal{Z}\}$ (we omit its dependence on $\mathcal{Z}$ for simplicity). Equivalently, $\mathcal{N}(z)$ is the polar cone of the convex set $\mathcal{Z} - z$ (a property that we will use in the proof).*

Now we are ready to show that our SP-MS condition with $\beta = 0$ is equivalent to metric subregularity of the operator $\mathcal{N} + F$, defined via: $(\mathcal{N} + F)(z) = \{g + F(z) \mid g \in \mathcal{N}(z)\}$.

**Theorem 22.** *Let $z^* \in \mathcal{Z}^*$. Then the following two statements are equivalent:*

- *$(\mathcal{N} + F)$ is globally metric subregular at $(z^*, \mathbf{0})$ with $\kappa > 0$;*

- *For all $z \in \mathcal{Z}\backslash\mathcal{Z}^*$, $\max_{z' \in \mathcal{Z}} F(z)^\top \frac{(z-z')}{\|z-z'\|} \geq \kappa d(z, \mathcal{Z}^*)$.*

*Proof.* Let $\mathcal{T} = \mathcal{N} + F$. Notice that

$$z \in \mathcal{Z}^* \ \Leftrightarrow \ F(z)^\top(z' - z) \geq 0 \ \Leftrightarrow \ -F(z) \in \mathcal{N}(z) \ \Leftrightarrow \ \mathbf{0} \in (\mathcal{N} + F)(z).$$

Therefore, $\mathbf{0} \in \mathcal{T}(z^*)$ indeed holds, and we have $\mathcal{T}^{-1}(\mathbf{0}) = \mathcal{Z}^*$. This means that the first statement in the theorem is equivalent to

$$d(\mathbf{0}, \mathcal{T}(z)) \geq \kappa d(z, \mathcal{T}^{-1}(\mathbf{0})) \ \Leftrightarrow \ d(\mathbf{0}, \mathcal{N}(z) + F(z)) \geq \kappa d(z, \mathcal{Z}^*).$$

This inequality holds trivially when $z \in \mathcal{Z}^*$. Thus, to complete the proof, it suffices to prove that $d(\mathbf{0}, \mathcal{N}(z) + F(z)) = \max_{z' \in \mathcal{Z}} F(z)^\top \frac{(z-z')}{\|z-z'\|}$ for $z \in \mathcal{Z}\backslash\mathcal{Z}^*$. To do so, note that

$$
\begin{aligned}
&d(\mathbf{0}, \mathcal{N}(z) + F(z)) \\
&= d(-F(z), \mathcal{N}(z)) \\
&= \| - F(z) - \Pi_{\mathcal{N}(z)}(-F(z))\| \\
&= \|\Pi_{\mathcal{N}^\circ(z)}(-F(z))\|
\end{aligned}
$$

where $\mathcal{N}^\circ(\boldsymbol{z}) = \{\boldsymbol{g} \mid \boldsymbol{g}^\top \boldsymbol{n} \leq 0, \ \forall \boldsymbol{n} \in \mathcal{N}(z)\}$ is the polar cone of $\mathcal{N}(\boldsymbol{z})$ and the last step is by Moreau's theorem. Now consider the projection of $-F(\boldsymbol{z})$ onto the polar cone $\mathcal{N}^\circ(\boldsymbol{z})$:

$$
\begin{aligned}
\Pi_{\mathcal{N}^\circ(\boldsymbol{z})}(-F(\boldsymbol{z})) &= \operatorname*{argmin}_{\boldsymbol{y} \in \mathcal{N}^\circ(\boldsymbol{z})} \| -F(\boldsymbol{z}) - \boldsymbol{y}\|^2 \\
&= \operatorname*{argmin}_{\boldsymbol{y} \in \mathcal{N}^\circ(\boldsymbol{z})} \left\{ 2F(\boldsymbol{z})^\top \boldsymbol{y} + \|\boldsymbol{y}\|^2 \right\} \\
&= \operatorname*{argmin}_{\boldsymbol{y} \in \mathcal{N}^\circ(\boldsymbol{z})} \left\{ 2F(\boldsymbol{z})^\top \frac{\boldsymbol{y}}{\|\boldsymbol{y}\|} \cdot \|\boldsymbol{y}\| + \|\boldsymbol{y}\|^2 \right\} \\
&= \operatorname*{argmin}_{\lambda \geq 0, \ \bar{\boldsymbol{z}} \in \mathcal{N}^\circ(\boldsymbol{z}), \ \|\bar{\boldsymbol{z}}\|=1} \left\{ 2\lambda F(\boldsymbol{z})^\top \bar{\boldsymbol{z}} + \lambda^2 \right\},
\end{aligned}
$$

where the last equality is because $\mathcal{N}^\circ(\boldsymbol{z})$ is a cone. Next, we find the $\bar{\boldsymbol{z}}^*$ and $\lambda^*$ that realize the last argmin operator: notice that the objective is increasing in $F(\boldsymbol{z})^\top \bar{\boldsymbol{z}}$, so $\bar{\boldsymbol{z}}^* = \operatorname{argmin}_{\bar{\boldsymbol{z}} \in \mathcal{N}^\circ(\boldsymbol{z}): \|\bar{\boldsymbol{z}}\|=1} \left\{ F(\boldsymbol{z})^\top \bar{\boldsymbol{z}} \right\}$, and thus $\lambda^* = -F(\boldsymbol{z})^\top \bar{\boldsymbol{z}}^*$ when $F(\boldsymbol{z})^\top \bar{\boldsymbol{z}}^* \leq 0$ and $\lambda^* = 0$ otherwise. Therefore,

$$
\|\Pi_{\mathcal{N}^\circ(\boldsymbol{z})}(-F(\boldsymbol{z}))\| = \lambda^* = \max\left\{ 0, \max_{\bar{\boldsymbol{z}} \in \mathcal{N}^\circ(\boldsymbol{z}), \|\bar{\boldsymbol{z}}\|=1} -F(\boldsymbol{z})^\top \bar{\boldsymbol{z}} \right\}.
$$

Note that $\mathcal{N}(z)$ is the polar cone of the conic hull of $\mathcal{Z} - \boldsymbol{z}$. Therefore, $\mathcal{N}^\circ(z) = (\mathbf{ConicHull}(\mathcal{Z} - \boldsymbol{z}))^{\circ\circ} = \mathbf{ConicHull}(\mathcal{Z} - \boldsymbol{z})$ and

$$
\max\left\{ 0, \max_{\bar{\boldsymbol{z}} \in \mathcal{N}^\circ(\boldsymbol{z}), \|\bar{\boldsymbol{z}}\|=1} -F(\boldsymbol{z})^\top \bar{\boldsymbol{z}} \right\} = \max\left\{ 0, \max_{\boldsymbol{z}' \in \mathcal{Z}} F(\boldsymbol{z})^\top \frac{(\boldsymbol{z} - \boldsymbol{z}')}{\|\boldsymbol{z}' - \boldsymbol{z}\|} \right\}.
$$

Finally, note that when $\boldsymbol{z} \in \mathcal{Z} \backslash \mathcal{Z}^*$, we have $\max_{\boldsymbol{z}' \in \mathcal{Z}} F(\boldsymbol{z})^\top (\boldsymbol{z} - \boldsymbol{z}') > 0$. Combining all the facts above, we have shown $d(\boldsymbol{0}, \mathcal{N}(\boldsymbol{z}) + F(\boldsymbol{z})) = \max_{\boldsymbol{z}' \in \mathcal{Z}} F(\boldsymbol{z})^\top \frac{(\boldsymbol{z}-\boldsymbol{z}')}{\|\boldsymbol{z}-\boldsymbol{z}'\|}$. $\qquad\square$

## G  PROOF OF THEOREM 5

*Proof of Theorem 5.* Let $\rho = \min_{\boldsymbol{x} \in \mathcal{X}} \max_{\boldsymbol{y} \in \mathcal{Y}} \boldsymbol{x}^\top \boldsymbol{G} \boldsymbol{y} = \max_{\boldsymbol{y} \in \mathcal{Y}} \min_{\boldsymbol{x} \in \mathcal{X}} \boldsymbol{x}^\top \boldsymbol{G} \boldsymbol{y}$ be the game value. In this proof, we prove that there exists some $c > 0$ such that

$$
\max_{\boldsymbol{y}' \in \mathcal{Y}} \boldsymbol{x}^\top \boldsymbol{G} \boldsymbol{y}' - \rho \geq c\|\boldsymbol{x} - \Pi_{\mathcal{X}^*}(\boldsymbol{x})\| \tag{20}
$$

for all $\boldsymbol{x} \in \mathcal{X}$. Similarly we prove

$$
\max_{\boldsymbol{x}' \in \mathcal{X}} \rho - \boldsymbol{x}'^\top \boldsymbol{G} \boldsymbol{y} \geq c\|\boldsymbol{y} - \Pi_{\mathcal{Y}^*}(\boldsymbol{y})\|
$$

for all $\boldsymbol{y} \in \mathcal{Y}$. Assume that the diameter of the polytope is $D < \infty$. Then combining the two proves

$$
\begin{aligned}
\max_{\boldsymbol{z}'} \frac{F(\boldsymbol{z})^\top (\boldsymbol{z} - \boldsymbol{z}')}{\|\boldsymbol{z} - \boldsymbol{z}'\|} &\geq \frac{1}{D} \max_{\boldsymbol{z}'} F(\boldsymbol{z})^\top (\boldsymbol{z} - \boldsymbol{z}') = \frac{1}{D} \left( \max_{\boldsymbol{y}'} \boldsymbol{x}^\top \boldsymbol{G} \boldsymbol{y}' - \min_{\boldsymbol{x}'} \boldsymbol{x}'^\top \boldsymbol{G} \boldsymbol{y} \right) \\
&\geq \frac{c}{D} \left( \|\boldsymbol{y} - \Pi_{\mathcal{Y}^*}(\boldsymbol{y})\| + \|\boldsymbol{x} - \Pi_{\mathcal{X}^*}(\boldsymbol{x})\| \right) \geq \frac{c}{D} \|\boldsymbol{z} - \Pi_{\mathcal{Z}^*}(\boldsymbol{z})\|,
\end{aligned}
$$

meaning that SP-MS holds with $\beta = 0$. We break the proof into following several claims.

**Claim 1.**  If $\mathcal{X}, \mathcal{Y}$ are polytopes, then $\mathcal{X}^*$ and $\mathcal{Y}^*$ are also polytopes.

*Proof of Claim 1.* Note that $\mathcal{X}^* = \left\{ \boldsymbol{x} \in \mathcal{X} : \max_{\boldsymbol{y} \in \mathcal{Y}} \boldsymbol{x}^\top \boldsymbol{G} \boldsymbol{y} \leq \rho \right\}$. Since $\mathcal{Y}$ is a polytope, the maximum is attained at vertices of $\mathcal{Y}$. Therefore, $\mathcal{X}^*$ can be equivalently written as $\left\{ \boldsymbol{x} \in \mathcal{X} : \max_{\boldsymbol{y} \in \mathcal{V}(\mathcal{Y})} \boldsymbol{x}^\top \boldsymbol{G} \boldsymbol{y} \leq \rho \right\}$, where $\mathcal{V}(\mathcal{Y})$ is the set of vertices of $\mathcal{Y}$. Since the constraints of $\mathcal{X}^*$ are all linear constraints, $\mathcal{X}^*$ is a polytope. $\qquad\square$

With Claim 1, we without loss of generality write $\mathcal{X}^*$ as

$$
\mathcal{X}^* = \left\{ \boldsymbol{x} \in \mathbb{R}^M : \quad \boldsymbol{a}_i^\top \boldsymbol{x} \leq b_i, \ \text{for } i = 1, \dots, L, \qquad \boldsymbol{c}_i^\top \boldsymbol{x} \leq d_i, \ \text{for } i = 1, \dots, K \right\},
$$

where the $a_i^\top x \le b_i$ constraints come from $x \in \mathcal{X}$ and the $c_i^\top x \le d_i$ constraints come from $\max_{y \in \mathcal{V}(\mathcal{Y})} x^\top G y \le \rho$. Below, we refer to $a_i^\top x \le b_i$ as the *feasibility constraints*, and $c_i^\top x \le d_i$ as the *optimality constraints*. In fact, one can identify the $i$-th optimality constraint as $c_i = G y^{(i)}$ and $d_i = \rho$, where $y^{(i)}$ is the $i$-th vertex of $\mathcal{Y}$. This is based on our construction of $\mathcal{X}^*$ in the proof of Claim 1. Therefore, $K = |\mathcal{V}(\mathcal{Y})|$.

Since Eq. (20) clearly holds for $x \in \mathcal{X}^*$, below, we focus on an $x \in \mathcal{X} \backslash \mathcal{X}^*$, and let $x^* \triangleq \Pi_{\mathcal{X}^*}(x)$.

We say a constraint is *tight* at $x^*$ if $a_i^\top x^* = b_i$ or $c_i^\top x^* = d_i$. Below we assume that there are $\ell$ tight feasibility constraints at and $k$ tight optimality constraints at $x^*$. Without loss of generality, we assume these tight constraints correspond to $i = 1, \ldots, \ell$ and $i = 1, \ldots, k$ respectively. That is,

$$a_i^\top x^* = b_i, \qquad \text{for } i = 1, \ldots, \ell,$$
$$c_i^\top x^* = d_i, \qquad \text{for } i = 1, \ldots, k.$$

**Claim 2.** $x$ violates at least one of the tight optimality constraint at $x^*$.

*Proof of Claim 2.* We prove this by contradiction. Suppose that $x$ satisfies all $k$ tight optimality constraints at $x^*$. Then $x$ must violates some of the remaining $K - k$ optimality constraints (otherwise $x \in \mathcal{X}^*$). Assume that it violates constraints $K - n + 1, \ldots, K$ for some $1 \le n \le K - k$. Thus, we have the following:

$$c_i^\top x \le d_i \quad \text{for } i = 1, \ldots K - n;$$
$$c_i^\top x > d_i \quad \text{for } i = K - n + 1, \ldots, K.$$

Recall that $c_i^\top x^* \le d_i$ for $i = 1, \ldots, K - n$ and $c_i^\top x^* < d_i$ for all $i = K - n + 1, \ldots, K$. Thus, there exists some $x'$ that lies strictly between $x$ and $x^*$ that makes all constraints hold (notice that $x$ and $x^*$ both satisfy all feasibility constraints), which contradicts with $\Pi_{\mathcal{X}^*}(x) = x^*$. $\qquad\square$

**Claim 3.** $\max_{y' \in \mathcal{Y}} \left(x^\top G y' - \rho\right) \ge \max_{i \in \{1, \ldots, k\}} c_i^\top (x - x^*)$.

*Proof of Claim 3.* Recall that we identify $c_i$ with $G y^{(i)}$ and $d_i = \rho$. Therefore,

$$\max_{y' \in \mathcal{Y}} \left(x^\top G y' - \rho\right) = \max_{i \in \{1, \ldots, |\mathcal{V}(\mathcal{Y})|\}} \left(c_i^\top x - d_i\right) \ge \max_{i \in \{1, \ldots, k\}} \left(c_i^\top x - d_i\right) = \max_{i \in \{1, \ldots, k\}} c_i^\top (x - x^*),$$

where the last equality is because $c_i^\top x^* = d_i$ for $i = 1, \ldots, k$. $\qquad\square$

Recall from linear programming literature Davis (2016a;b) that the *normal cone* of $\mathcal{X}^*$ at $x^*$ is expressed as follows:

$$\mathcal{N}_{x^*} = \left\{ x' - x^* : \ x' \in \mathbb{R}^M, \quad \Pi_{\mathcal{X}^*}(x') = x^* \right\} = \left\{ \sum_{i=1}^{\ell} p_i a_i + \sum_{i=1}^{k} q_i c_i : \quad p_i \ge 0, \quad q_i \ge 0 \right\}.$$

The normal cone of $\mathcal{X}^*$ at $x^*$ consists of all outgoing normal vectors of $\mathcal{X}^*$ originated from $x^*$. Clearly, $x - x^*$ belongs to $\mathcal{N}_{x^*}$. However, besides the fact that $x - x^*$ is a normal vector of $\mathcal{X}^*$, we also have the additional constraints that $x \in \mathcal{X}$. We claim that in our case, $x - x^*$ lies in the following smaller cone (which is a subset of $\mathcal{N}_{x^*}$):

**Claim 4.** $x - x^*$ belongs to

$$\mathcal{M}_{x^*} = \left\{ \sum_{i=1}^{\ell} p_i a_i + \sum_{i=1}^{k} q_i c_i : \ p_i \ge 0, \ q_i \ge 0, \ a_j^\top \left( \sum_{i=1}^{\ell} p_i a_i + \sum_{i=1}^{k} q_i c_i \right) \le 0, \ \forall j = 1, \ldots, \ell \right\}.$$

*Proof of Claim 4.* As argued above, $x - x^* \in \mathcal{N}_{x^*}$, and thus $x - x^*$ can be expressed as $\sum_{i=1}^{\ell} p_i a_i + \sum_{i=1}^{k} q_i c_i$ with $p_i \ge 0, q_i \ge 0$. To prove that $x - x^* \in \mathcal{M}_{x^*}$, we only need to prove that it satisfies the additional constraints, that is,

$$a_i^\top (x - x^*) \le 0, \ \forall i = 1, \ldots, \ell.$$

This is shown by noticing that for all $i = 1, \ldots, \ell$,

$$
\begin{aligned}
\boldsymbol{a}_i^\top (\boldsymbol{x} - \boldsymbol{x}^*) &= \left(\boldsymbol{a}_i^\top \boldsymbol{x}^* - b_i\right) + \boldsymbol{a}_i^\top (\boldsymbol{x} - \boldsymbol{x}^*) && \text{(the $i$-th constraint is tight at $\boldsymbol{x}^*$)} \\
&= \boldsymbol{a}_i^\top (\boldsymbol{x}^* + \boldsymbol{x} - \boldsymbol{x}^*) - b_i \\
&= \boldsymbol{a}_i^\top \boldsymbol{x} - b_i \leq 0. && (\boldsymbol{x} \in \mathcal{X})
\end{aligned}
$$

$\square$

**Claim 5.** $\boldsymbol{x} - \boldsymbol{x}^*$ can be written as $\sum_{i=1}^{\ell} p_i \boldsymbol{a}_i + \sum_{i=1}^{k} q_i \boldsymbol{c}_i$ with $0 \leq p_i, q_i \leq C' \|\boldsymbol{x} - \boldsymbol{x}^*\|$ for all $i$ and some problem-dependent constant $C' < \infty$.

*Proof of Claim 5.* Notice that $\frac{\boldsymbol{x} - \boldsymbol{x}^*}{\|\boldsymbol{x} - \boldsymbol{x}^*\|} \in \mathcal{M}_{\boldsymbol{x}^*}$ (because $\boldsymbol{0} \neq \boldsymbol{x} - \boldsymbol{x}^* \in \mathcal{M}_{\boldsymbol{x}^*}$ and $\mathcal{M}_{\boldsymbol{x}^*}$ is a cone). Furthermore, $\frac{\boldsymbol{x} - \boldsymbol{x}^*}{\|\boldsymbol{x} - \boldsymbol{x}^*\|} \in \{\boldsymbol{v} \in \mathbb{R}^M : \|\boldsymbol{v}\|_\infty \leq 1\}$. Therefore, $\frac{\boldsymbol{x} - \boldsymbol{x}^*}{\|\boldsymbol{x} - \boldsymbol{x}^*\|} \in \mathcal{M}_{\boldsymbol{x}^*} \cap \{\boldsymbol{v} \in \mathbb{R}^M : \|\boldsymbol{v}\|_\infty \leq 1\}$, which is a bounded subset of the cone $\mathcal{M}_{\boldsymbol{x}^*}$.

Below we argue that there exists a large enough $C' > 0$ such that

$$
\left\{ \sum_{i=1}^{\ell} p_i \boldsymbol{a}_i + \sum_{i=1}^{k} q_i \boldsymbol{c}_i : 0 \leq p_i, q_i \leq C', \ \forall i \right\} \quad \supseteq \quad \mathcal{M}_{\boldsymbol{x}^*} \cap \{\boldsymbol{v} \in \mathbb{R}^M : \|\boldsymbol{v}\|_\infty \leq 1\} \quad \triangleq \quad \mathcal{P}.
$$

To see this, first note that $\mathcal{P}$ is a polytope. For every vertex $\widehat{\boldsymbol{v}}$ of $\mathcal{P}$, the smallest $C'$ such that $\widehat{\boldsymbol{v}}$ belongs to the left-hand side is the solution of the following linear programming:

$$
\min_{p_i, q_i, C'_{\widehat{\boldsymbol{v}}}} C'_{\widehat{\boldsymbol{v}}} \quad s.t. \quad \widehat{\boldsymbol{v}} = \sum_{i=1}^{\ell} p_i \boldsymbol{a}_i + \sum_{i=1}^{k} q_i \boldsymbol{c}_i, \quad 0 \leq p_i, q_i \leq C'_{\widehat{\boldsymbol{v}}}.
$$

Since $\widehat{v} \in \mathcal{M}_{\boldsymbol{x}^*}$, this linear programming is always feasible and admits a finite solution $C'_{\widehat{\boldsymbol{v}}} < \infty$. Now let $C' = \max_{\widehat{\boldsymbol{v}} \in \mathcal{V}(\mathcal{P})} C'_{\widehat{\boldsymbol{v}}}$, where $\mathcal{V}(\mathcal{P})$ is the set of all vertices of $\mathcal{P}$. Then since any $v \in \mathcal{P}$ can be expressed as a convex combination of points in $\mathcal{V}(\mathcal{P})$, $v$ can be also be expressed as $\sum_{i=1}^{\ell} p_i \boldsymbol{a}_i + \sum_{i=1}^{k} q_i \boldsymbol{c}_i$ with $0 \leq p_i, q_i \leq C'$.

To sum up, $\frac{\boldsymbol{x} - \boldsymbol{x}^*}{\|\boldsymbol{x} - \boldsymbol{x}^*\|}$ can be represented as $\sum_{i=1}^{\ell} p_i \boldsymbol{a}_i + \sum_{i=1}^{k} q_i \boldsymbol{c}_i$ with $0 \leq p_i, q_i \leq C'$. This further implies that $\boldsymbol{x} - \boldsymbol{x}^*$ can be represented as $\sum_{i=1}^{\ell} p_i \boldsymbol{a}_i + \sum_{i=1}^{k} q_i \boldsymbol{c}_i$ with $0 \leq p_i, q_i \leq C' \|\boldsymbol{x} - \boldsymbol{x}^*\|$. Notice that $C'$ only depends on the set of tight constraints at $\boldsymbol{x}^*$. $\square$

Finally, we are ready to combine all previous claims and prove the desired inequality.

Define $A_i \triangleq \boldsymbol{a}_i^\top (\boldsymbol{x} - \boldsymbol{x}^*)$ and $C_i \triangleq \boldsymbol{c}_i^\top (\boldsymbol{x} - \boldsymbol{x}^*)$. By Claim 5, we can write $\boldsymbol{x} - \boldsymbol{x}^*$ as $\sum_{i=1}^{\ell} p_i \boldsymbol{a}_i + \sum_{i=1}^{k} q_i \boldsymbol{c}_i$ with $0 \leq p_i, q_i \leq C' \|\boldsymbol{x} - \boldsymbol{x}^*\|$, and thus,

$$
\sum_{i=1}^{\ell} p_i A_i + \sum_{i=1}^{k} q_i C_i = \left( \sum_{i=1}^{\ell} p_i \boldsymbol{a}_i + \sum_{i=1}^{k} q_i \boldsymbol{c}_i \right)^\top (\boldsymbol{x} - \boldsymbol{x}^*) = \|\boldsymbol{x} - \boldsymbol{x}^*\|^2.
$$

On the other hand, since $\boldsymbol{x} - \boldsymbol{x}^* \in \mathcal{M}_{\boldsymbol{x}^*}$ by Claim 4, we have

$$
\sum_{i=1}^{\ell} p_i A_i = \sum_{i=1}^{\ell} p_i \boldsymbol{a}_i^\top (\boldsymbol{x} - \boldsymbol{x}^*) \leq 0
$$

and

$$
\sum_{i=1}^{k} q_i C_i \leq \left( \max_{i \in \{1, \ldots, k\}} C_i \right) \sum_{i=1}^{k} q_i \leq \left( \max_{i \in \{1, \ldots, k\}} C_i \right) k C' \|\boldsymbol{x} - \boldsymbol{x}^*\|,
$$

where in the first inequality we use the fact $p_i \geq 0$, and in the second inequality we use the fact $\max_{i \in \{1, \ldots, k\}} C_i > 0$ (by Claim 2) and $0 \leq q_i \leq C' \|\boldsymbol{x} - \boldsymbol{x}^*\|$.

Combining the three inequalities above, we get

$$\max_{i \in \{1,\dots,k\}} C_i \geq \frac{1}{kC'} \|\boldsymbol{x} - \boldsymbol{x}^*\|.$$

Then by Claim 3,

$$\max_{\boldsymbol{y}' \in \mathcal{Y}} \left( \boldsymbol{x}^\top \boldsymbol{G} \boldsymbol{y}' - \rho \right) \geq \max_{i \in \{1,\dots,k\}} C_i \geq \frac{1}{kC'} \|\boldsymbol{x} - \boldsymbol{x}^*\|.$$

Note that $k$ and $C'$ only depend on the set of tight constraints at the projection point $\boldsymbol{x}^*$, and there are only finitely many different sets of tight constraints. Therefore, we conclude that there exists a constant $c > 0$ such that $\max_{\boldsymbol{y}' \in \mathcal{Y}} \left( \boldsymbol{x}^\top \boldsymbol{G} \boldsymbol{y}' - \rho \right) \geq c\|\boldsymbol{x} - \boldsymbol{x}^*\|$ holds for all $\boldsymbol{x}$ and $\boldsymbol{x}^*$, which completes the proof. $\qquad \square$

## H   PROOF OF THEOREM 6 AND THEOREM 7

*Proof of Theorem 6.* Suppose that $f$ is $\gamma$-strongly-convex in $\boldsymbol{x}$ and $\gamma$-strongly-concave in $\boldsymbol{y}$, and let $(\boldsymbol{x}^*, \boldsymbol{y}^*) \in \mathcal{Z}^*$. Then for any $(\boldsymbol{x}, \boldsymbol{y})$ we have

$$f(\boldsymbol{x}, \boldsymbol{y}) - f(\boldsymbol{x}^*, \boldsymbol{y}) \leq \nabla_x f(\boldsymbol{x}, \boldsymbol{y})^\top (\boldsymbol{x} - \boldsymbol{x}^*) - \frac{\gamma}{2} \|\boldsymbol{x} - \boldsymbol{x}^*\|^2,$$

$$f(\boldsymbol{x}, \boldsymbol{y}^*) - f(\boldsymbol{x}, \boldsymbol{y}) \leq \nabla_y f(\boldsymbol{x}, \boldsymbol{y})^\top (\boldsymbol{y}^* - \boldsymbol{y}) - \frac{\gamma}{2} \|\boldsymbol{y} - \boldsymbol{y}^*\|^2.$$

Summing up the two inequalities, and noticing that $f(\boldsymbol{x}, \boldsymbol{y}^*) - f(\boldsymbol{x}^*, \boldsymbol{y}) \geq 0$ for any $(\boldsymbol{x}^*, \boldsymbol{y}^*) \in \mathcal{Z}^*$, we get

$$F(\boldsymbol{z})^\top (\boldsymbol{z} - \boldsymbol{z}^*) \geq \frac{\gamma}{2} \|\boldsymbol{z} - \boldsymbol{z}^*\|^2,$$

and therefore, for $\boldsymbol{z} \notin \mathcal{Z}^*$,

$$\frac{F(\boldsymbol{z})^\top (\boldsymbol{z} - \boldsymbol{z}^*)}{\|\boldsymbol{z} - \boldsymbol{z}^*\|} \geq \frac{\gamma}{2} \|\boldsymbol{z} - \boldsymbol{z}^*\|,$$

which implies SP-MS with $\beta = 0$ and $C = \gamma/2$. $\qquad \square$

*Proof of Theorem 7.* First, we show that $f$ has a unique Nash Equilibrium $\boldsymbol{z}^* = (\boldsymbol{x}^*, \boldsymbol{y}^*) = ((0,1),(0,1))$. As $f$ is a strictly monotone decreasing function with respect to $y_1$, we must have $y_1^* = 0$ and $y_2^* = 1$. In addition, if $\boldsymbol{x} = (0,1)$, $\max_{\boldsymbol{y} \in \mathcal{Y}} f(\boldsymbol{x}, \boldsymbol{y}) = -\min_{\boldsymbol{y} \in \mathcal{Y}} y_1^{2n} = 0$. If $\boldsymbol{x} \neq (0,1)$, then by choosing $\boldsymbol{y}^* = (0,1)$, $f(\boldsymbol{x}, \boldsymbol{y}^*) = x_1^{2n} > 0$. Therefore, we have $\boldsymbol{x}^* = (0,1)$, which proves that the unique Nash Equilibrium is $\boldsymbol{x}^* = (0,1), \boldsymbol{y}^* = (0,1)$.

Second, we show that $f$ satisfies SP-MS with $\beta = 2n - 2$. In fact, for any $\boldsymbol{z} = (\boldsymbol{x}, \boldsymbol{y}) \neq \boldsymbol{z}^*$, we have

$$
\begin{aligned}
F(\boldsymbol{z})^\top (\boldsymbol{z} - \boldsymbol{z}^*) &= \begin{bmatrix} 2nx_1^{2n-1} - y_1 \\ 0 \\ 2ny_1^{2n-1} + x_1 \\ 0 \end{bmatrix}^\top \begin{bmatrix} x_1 \\ x_2 - 1 \\ y_1 \\ y_2 - 1 \end{bmatrix} \\
&= 2n \left( x_1^{2n} + y_1^{2n} \right) \\
&\geq 4n \cdot \left( \frac{x_1^2 + y_1^2}{2} \right)^n \qquad \text{(Jensen's inequality)} \\
&= \frac{n}{2^{n-2}} \left( x_1^2 + y_1^2 \right)^n.
\end{aligned}
$$

Note that $\|\boldsymbol{z} - \boldsymbol{z}^*\| = \sqrt{x_1^2 + (1 - x_2)^2 + y_1^2 + (1 - y_2)^2} = \sqrt{2x_1^2 + 2y_1^2}$. Therefore, we have $\frac{F(\boldsymbol{z})^\top (\boldsymbol{z} - \boldsymbol{z}^*)}{\|\boldsymbol{z} - \boldsymbol{z}^*\|} \geq \frac{n}{2^{2n-2}} \|\boldsymbol{z} - \boldsymbol{z}^*\|^{2n-1}$. This shows that $f$ satisfies SP-MS with $\beta = 2n - 2$ and $C = \frac{n}{2^{2n-2}}$. $\qquad \square$

# I  PROOF OF THEOREM 8

*Proof of Theorem 8.* As argued in Section 5, with $\Theta_t = \|\widehat{z}_t - \Pi_{\mathcal{Z}^*}(\widehat{z}_t)\|^2 + \frac{1}{16}\|\widehat{z}_t - z_{t-1}\|^2$, $\zeta_t = \|\widehat{z}_{t+1} - z_t\|^2 + \|z_t - \widehat{z}_t\|^2$, we have (see Eq. (4))

$$\Theta_{t+1} \leq \Theta_t - \frac{15}{16}\zeta_t. \tag{21}$$

Below, we relate $\zeta_t$ to $\Theta_{t+1}$ using the SP-MS condition, and then apply Lemma 12 to show

$$\Theta_t \leq \begin{cases} 2\mathrm{dist}^2(\widehat{z}_1, \mathcal{Z}^*)(1+C_5)^{-t} & \text{if } \beta = 0, \\ \left[\left(1 + 4\left(\frac{4}{\beta}\right)^{\frac{1}{\beta}}\right)\mathrm{dist}^2(\widehat{z}_1, \mathcal{Z}^*) + 2\left(\frac{2}{C_5\beta}\right)^{\frac{1}{\beta}}\right]t^{-\frac{1}{\beta}} & \text{if } \beta > 0, \end{cases} \tag{22}$$

where $C_5 = \min\left\{\frac{16\eta^2 C^2}{81}, \frac{1}{2}\right\}$ as defined in the statement of the theorem. This is enough to prove the theorem since

$$\begin{aligned} \mathrm{dist}^2(z_t, \mathcal{Z}^*) &\leq \|z_t - \Pi_{\mathcal{Z}^*}(\widehat{z}_{t+1})\|^2 \\ &\leq 2\|\widehat{z}_{t+1} - \Pi_{\mathcal{Z}^*}(\widehat{z}_{t+1})\|^2 + 2\|\widehat{z}_{t+1} - z_t\|^2 \\ &\leq 32\Theta_{t+1} \leq 32\Theta_t. \end{aligned}$$

Next, we prove Eq. (22). We first show a simple fact by Eq. (21):

$$\|\widehat{z}_{t+1} - z_t\|^2 \leq \zeta_t \leq \frac{16}{15}\Theta_t \leq \cdots \leq \frac{16}{15}\Theta_1. \tag{23}$$

Notice that

$$\begin{aligned} \zeta_t &\geq \frac{1}{2}\|\widehat{z}_{t+1} - z_t\|^2 + \frac{1}{2}\left(\|\widehat{z}_{t+1} - z_t\|^2 + \|z_t - \widehat{z}_t\|^2\right) \\ &\geq \frac{1}{2}\|\widehat{z}_{t+1} - z_t\|^2 + \frac{16\eta^2}{81}\sup_{z'\in\mathcal{Z}}\frac{\left[F(\widehat{z}_{t+1})^\top(\widehat{z}_{t+1} - z')\right]_+^2}{\|\widehat{z}_{t+1} - z'\|^2} &&\text{(Lemma 4)} \\ &\geq \frac{1}{2}\|\widehat{z}_{t+1} - z_t\|^2 + \frac{16\eta^2 C^2}{81}\|\widehat{z}_{t+1} - \Pi_{\mathcal{Z}^*}(\widehat{z}_{t+1})\|^{2(\beta+1)} &&\text{(SP-MS condition)} \\ &\geq \min\left\{\frac{16\eta^2 C^2}{81}, \frac{1}{2}\left(\frac{15}{16\Theta_1}\right)^\beta\right\}\left(\|\widehat{z}_{t+1} - z_t\|^{2(\beta+1)} + \|\widehat{z}_{t+1} - \Pi_{\mathcal{Z}^*}(\widehat{z}_{t+1})\|^{2(\beta+1)}\right) \\ &&&\text{(by Eq. (23))} \\ &\geq \min\left\{\frac{16\eta^2 C^2}{2^\beta \cdot 81}, \frac{1}{2}\left(\frac{15}{32\Theta_1}\right)^\beta\right\}\left(\|\widehat{z}_{t+1} - z_t\|^2 + \|\widehat{z}_{t+1} - \Pi_{\mathcal{Z}^*}(\widehat{z}_{t+1})\|^2\right)^{\beta+1} \\ &&&\text{(by Hölder's inequality: } (a^{\beta+1} + b^{\beta+1})(1+1)^\beta \geq (a+b)^{\beta+1}) \\ &\geq \min\left\{\frac{C_5}{2^\beta}, \frac{1}{2}\left(\frac{1}{4\Theta_1}\right)^\beta\right\}\Theta_{t+1}^{\beta+1} &&\text{(recall that } C_5 = \min\{\frac{16\eta^2 C^2}{81}, \frac{1}{2}\}) \\ &= C'\Theta_{t+1}^{\beta+1}. &&\text{(define } C' = \min\left\{\frac{C_5}{2^\beta}, \frac{1}{2}\left(\frac{1}{4\Theta_1}\right)^\beta\right\}) \end{aligned}$$

Combining this with Eq. (21), we get

$$\Theta_{t+1} \leq \Theta_t - C'\Theta_{t+1}^{\beta+1} \tag{24}$$

When $\beta = 0$, Eq. (24) implies $\Theta_{t+1} \leq (1 + C_5)^{-1}\Theta_t$, which immediately implies $\Theta_t \leq (1 + C_5)^{-t+1}\Theta_1 \leq 2\Theta_1(1 + C_5)^{-t}$. When $\beta > 0$, Eq. (24) is of the form specified in Lemma 12 with $p = \beta$ and $q = C'$. Note that the second required condition is satisfied: $C'(\beta + 1)\Theta_1^\beta \leq \frac{\beta+1}{2\cdot 4^\beta} \leq 1$. Therefore, by the conclusion of Lemma 12,

$$\begin{aligned} \Theta_t &\leq \max\left\{\Theta_1, \left(\frac{2}{C'\beta}\right)^{\frac{1}{\beta}}\right\}t^{-\frac{1}{\beta}} = \max\left\{\Theta_1, \left(\frac{2\cdot 2^\beta}{C_5\beta}\right)^{\frac{1}{\beta}}, 4\Theta_1\left(\frac{4}{\beta}\right)^{\frac{1}{\beta}}\right\}t^{-\frac{1}{\beta}} \\ &\leq \left[\left(1 + 4\left(\frac{4}{\beta}\right)^{\frac{1}{\beta}}\right)\Theta_1 + 2\left(\frac{2}{C_5\beta}\right)^{\frac{1}{\beta}}\right]t^{-\frac{1}{\beta}}. \end{aligned}$$

Eq. (22) is then proven by noticing that $\Theta_1 = \mathrm{dist}^2(\widehat{z}_1, \mathcal{Z}^*)$. □

## J    PROOF OF THEOREM 9

*Proof of Theorem 9.* Consider the following $2 \times 2$ bilinear game with curved feasible sets:

$$f(\boldsymbol{x}, \boldsymbol{y}) = \boldsymbol{x}^\top \boldsymbol{G} \boldsymbol{y} = [x_1 \quad x_2] \begin{bmatrix} 0 & -1 \\ 1 & 0 \end{bmatrix} \begin{bmatrix} y_1 \\ y_2 \end{bmatrix},$$

$$\mathcal{X} = \left\{ \boldsymbol{x}: \quad 0 \leq x_1 \leq \frac{1}{2}, \quad 0 \leq x_2 \leq \frac{1}{4}, \quad x_2 \geq x_1^2 \right\},$$

$$\mathcal{Y} = \left\{ \boldsymbol{y}: \quad 0 \leq y_1 \leq \frac{1}{2}, \quad 0 \leq y_2 \leq \frac{1}{4}, \quad y_2 \geq y_1^2 \right\}.$$

Below, we use **Claim 1** - **Claim 5** to argue that if the two players start from $\boldsymbol{x}_0 = \boldsymbol{y}_0 = \widehat{\boldsymbol{x}}_0 = \widehat{\boldsymbol{y}}_0 = (\frac{1}{2}, \frac{1}{4})$, and use any constant learning rate $\eta \leq \frac{1}{64}$, then the convergence is sublinear in the sense that $\|\boldsymbol{z}_t - \boldsymbol{z}^*\| \geq \Omega(1/t)$. Then, in **Claim 6**, we show that in this example, SP-MS holds with $\beta = 3$.

**Claim 1.**    The unique equilibrium is $\boldsymbol{x}^* = \boldsymbol{0}, \boldsymbol{y}^* = \boldsymbol{0}$.

When $\boldsymbol{x} = \boldsymbol{0}$, clearly $\max_{\boldsymbol{y}' \in \mathcal{Y}} f(\boldsymbol{x}, \boldsymbol{y}') = 0$. When $\boldsymbol{x} \neq \boldsymbol{0}$, we prove $\max_{\boldsymbol{y}' \in \mathcal{Y}} f(\boldsymbol{x}, \boldsymbol{y}') > 0$ below. If $x_1 \neq 0$, we let $y_1' = \frac{1}{2}x_1$ and $y_2' = \frac{1}{4}x_1^2$ (which satisfies $\boldsymbol{y}' \in \mathcal{Y}$), and thus

$$f(\boldsymbol{x}, \boldsymbol{y}') = x_2 y_1' - x_1 y_2' = x_1^2 \cdot \frac{1}{2}x_1 - x_1 \cdot \frac{1}{4}x_1^2 = \frac{1}{4}x_1^3 > 0.$$

If $x_1 = 0$ but $x_2 \neq 0$, we let $y_1' = \frac{1}{2}, y_2' = \frac{1}{4}$, and thus

$$f(\boldsymbol{x}, \boldsymbol{y}') = x_2 y_1' - x_1 y_2' = \frac{1}{2}x_2 > 0.$$

Thus, $\max_{\boldsymbol{y}' \in \mathcal{Y}} f(\boldsymbol{x}, \boldsymbol{y}') > 0$ if $\boldsymbol{x} \neq \boldsymbol{0}$, and $\boldsymbol{x}^* = \boldsymbol{0}$ is the unique optimal solution for $\boldsymbol{x}$. By the symmetry between $\boldsymbol{x}$ and $\boldsymbol{y}$ (because $\boldsymbol{G} = -\boldsymbol{G}^\top$), we can also prove that the unique optimal solution for $\boldsymbol{y}$ is $\boldsymbol{y}^* = \boldsymbol{0}$.

**Claim 2.**    Suppose that $\boldsymbol{x}_0 = \boldsymbol{y}_0 = \widehat{\boldsymbol{x}}_0 = \widehat{\boldsymbol{y}}_0 = (\frac{1}{2}, \frac{1}{4})$. Then, at any step $t \in [T]$, we have $\boldsymbol{x}_t = \boldsymbol{y}_t$ and $\widehat{\boldsymbol{x}}_t = \widehat{\boldsymbol{y}}_t$, and all $\boldsymbol{x}_t, \boldsymbol{y}_t, \widehat{\boldsymbol{x}}_t, \widehat{\boldsymbol{y}}_t$ belong to $\{\boldsymbol{u} \in \mathbb{R}^2 : u_2 = u_1^2\}$.

We prove this by induction. The base case trivially holds. Suppose that for step $t$, we have $\boldsymbol{x}_t = \boldsymbol{y}_t$, $\widehat{\boldsymbol{x}}_t = \widehat{\boldsymbol{y}}_t$, and $\boldsymbol{x}_t, \boldsymbol{y}_t, \widehat{\boldsymbol{x}}_t, \widehat{\boldsymbol{y}}_t \in \{\boldsymbol{u} \in \mathbb{R}^2 : u_2 = u_1^2\}$. Then consider step $t + 1$. According to the dynamic of OGDA, we have

$$\widehat{\boldsymbol{x}}_{t+1} = \Pi_{\mathcal{X}} \left\{ \widehat{\boldsymbol{x}}_t - \eta \begin{bmatrix} -y_{t,2} \\ y_{t,1} \end{bmatrix} \right\} = \Pi_{\mathcal{X}} \left\{ \begin{bmatrix} \widehat{x}_{t,1} + \eta y_{t,2} \\ \widehat{x}_{t,2} - \eta y_{t,1} \end{bmatrix} \right\}, \tag{25}$$

$$\boldsymbol{x}_{t+1} = \Pi_{\mathcal{X}} \left\{ \widehat{\boldsymbol{x}}_{t+1} - \eta \begin{bmatrix} -y_{t,2} \\ y_{t,1} \end{bmatrix} \right\} = \Pi_{\mathcal{X}} \left\{ \begin{bmatrix} \widehat{x}_{t+1,1} + \eta y_{t,2}, \\ \widehat{x}_{t+1,2} - \eta y_{t,1} \end{bmatrix} \right\},$$

$$\widehat{\boldsymbol{y}}_{t+1} = \Pi_{\mathcal{Y}} \left\{ \widehat{\boldsymbol{y}}_t + \eta \begin{bmatrix} x_{t,2} \\ -x_{t,1} \end{bmatrix} \right\} = \Pi_{\mathcal{Y}} \left\{ \begin{bmatrix} \widehat{y}_{t,1} + \eta x_{t,2} \\ \widehat{y}_{t,2} - \eta x_{t,1} \end{bmatrix} \right\},$$

$$\boldsymbol{y}_{t+1} = \Pi_{\mathcal{Y}} \left\{ \widehat{\boldsymbol{y}}_{t+1} + \eta \begin{bmatrix} x_{t,2} \\ -x_{t,1} \end{bmatrix} \right\} = \Pi_{\mathcal{Y}} \left\{ \begin{bmatrix} \widehat{y}_{t+1,1} + \eta x_{t,2} \\ \widehat{y}_{t+1,2} - \eta x_{t,1} \end{bmatrix} \right\}.$$

According to induction hypothesis, we have $\widehat{\boldsymbol{x}}_{t+1} = \widehat{\boldsymbol{y}}_{t+1}$, which further leads to $\boldsymbol{x}_{t+1} = \boldsymbol{y}_{t+1}$.

Now we prove that for any $\begin{bmatrix} x_1 \\ x_2 \end{bmatrix}$ such that $x_1 \geq 0$, $x_2 \leq \frac{1}{4}$ and $x_2 < x_1^2$, $\begin{bmatrix} \overline{x}_1 \\ \overline{x}_2 \end{bmatrix} = \Pi_{\mathcal{X}} \left\{ \begin{bmatrix} x_1 \\ x_2 \end{bmatrix} \right\}$ satisfies that $\overline{x}_1^2 = \overline{x}_2$. Otherwise, suppose that $\overline{x}_1^2 < \overline{x}_2$. Then according to the intermediate value theorem, there exists $\begin{bmatrix} \widetilde{x}_1 \\ \widetilde{x}_2 \end{bmatrix}$ that lies in the line segment of $\begin{bmatrix} x_1 \\ x_2 \end{bmatrix}$ and $\begin{bmatrix} \overline{x}_1 \\ \overline{x}_2 \end{bmatrix}$ such that $\widetilde{x}_1^2 = \widetilde{x}_2$. Moreover, as $x_1 \geq 0$, $\widetilde{x}_1 \geq 0$, $x_2 \leq \frac{1}{4}$, $\widetilde{x}_2 \leq \frac{1}{4}$, we know that $\begin{bmatrix} \widetilde{x}_1 \\ \widetilde{x}_2 \end{bmatrix} \in \mathcal{X}$. Therefore, we have $\|\widetilde{\boldsymbol{x}} - \boldsymbol{x}\| < \|\overline{\boldsymbol{x}} - \boldsymbol{x}\|$, which leads to contradiction.

Now consider $\widehat{x}_{t+1}$. According to induction hypothesis, we have $(\widehat{x}_{t,1} + \eta y_{t,2})^2 \geq \widehat{x}_{t,1}^2 = \widehat{x}_{t,2} \geq \widehat{x}_{t,2} - \eta y_{t,1}$. If equalities hold, trivially we have $\widehat{x}_{t+1,1}^2 = \widehat{x}_{t,1}^2 = \widehat{x}_{t,2} = \widehat{x}_{t+1,2}$ according to Eq. (25). Otherwise, as $\widehat{x}_{t,1} + \eta y_{t,2} \geq 0$, $\widehat{x}_{t,2} - \eta y_{t,1} \leq \frac{1}{4}$, according to the analysis above, we also have $\widehat{x}_{t+1,1}^2 = \widehat{x}_{t+1,2}$. Applying similar analysis to $\widehat{y}_{t+1}$, $x_{t+1}$ and $y_{t+1}$ finishes the induction proof.

**Claim 3.** With $\eta \leq \frac{1}{64}$, the following holds for all $t \geq 1$,

$$x_{t,1} \in \left[\frac{1}{2}\widehat{x}_{t,1}, 2\widehat{x}_{t,1}\right], \tag{26}$$

$$\widehat{x}_{t,1} \in \left[\widehat{x}_{t-1,1} - 4\eta\widehat{x}_{t-1,1}^2, \widehat{x}_{t-1,1} + 4\eta\widehat{x}_{t-1,1}^2\right]. \tag{27}$$

We prove the claim by induction on $t$. The case $t = 1$ trivially holds. Suppose that Eq. (26) and Eq. (27) hold at step $t$. Now consider step $t + 1$.

**Induction to get Eq. (27).** According to Claim 2, we have

$$\widehat{x}_{t+1} = \Pi_{\mathcal{X}}\left\{\widehat{x}_t - \eta \begin{bmatrix} -y_{t,2} \\ y_{t,1} \end{bmatrix}\right\} = \Pi_{\mathcal{X}}\left\{\begin{bmatrix} \widehat{x}_{t,1} + \eta x_{t,1}^2 \\ \widehat{x}_{t,1}^2 - \eta x_{t,1} \end{bmatrix}\right\},$$

and $\widehat{x}_{t+1} = (u, u^2)$ for some $u \in [0, 1/2]$. Using the definition of the projection function, we have

$$\widehat{x}_{t+1,1} = \operatorname*{argmin}_{u\in[0,\frac{1}{2}]}\left\{\left(\widehat{x}_{t,1} + \eta x_{t,1}^2 - u\right)^2 + \left(\widehat{x}_{t,1}^2 - \eta x_{t,1} - u^2\right)^2\right\} \triangleq \operatorname*{argmin}_{u\in[0,\frac{1}{2}]} g(u).$$

Now we show that $\operatorname{argmin}_{u\in[0,\frac{1}{2}]} g(u) = \operatorname{argmin}_{u\in\mathbb{R}} g(u)$. Note that

$$\nabla g(u) = 2(u - \widehat{x}_{t,1} - \eta x_{t,1}^2) + 4u\left(u^2 + \eta x_{t,1} - \widehat{x}_{t,1}^2\right), \tag{28}$$

Therefore, when $u > \frac{1}{2}$, using $x_{t,1} \leq \frac{1}{2}$, we have

$$\nabla g(u) > -2\eta x_{t,1}^2 + 2\eta x_{t,1} \geq 0, \tag{29}$$

which means $g(u) > g(\frac{1}{2})$. On the other hand, when $u < 0$, using $\widehat{x}_{t,1} \leq \frac{1}{2}$, we have

$$\nabla g(u) < 2u - 4u\widehat{x}_{t,1}^2 \leq u < 0, \tag{30}$$

which means $g(u) > g(0)$. Combining Eq. (29) and Eq. (30), we know that $\operatorname{argmin}_{u\in[0,\frac{1}{2}]} g(u) = \operatorname{argmin}_{u\in\mathbb{R}} g(u)$. Therefore, $\widehat{x}_{t+1,1}$ is the unconstrained minimizer of convex function $g(u)$, which means $\nabla g(\widehat{x}_{t+1,1}) = 0$. Below we use contradiction to prove that $\widehat{x}_{t+1,1} \geq \widehat{x}_{t,1} - 4\eta\widehat{x}_{t,1}^2$. If $\widehat{x}_{t+1,1} < \widehat{x}_{t,1} - 4\eta\widehat{x}_{t,1}^2$, we use Eq. (28) and get

$$\begin{aligned}
\nabla g(\widehat{x}_{t+1,1}) &= 2(\widehat{x}_{t+1,1} - \widehat{x}_{t,1} - \eta x_{t,1}^2) + 4\widehat{x}_{t+1,1}\left(\widehat{x}_{t+1,1}^2 + \eta x_{t,1} - \widehat{x}_{t,1}^2\right) \\
&< 2(-4\eta\widehat{x}_{t,1}^2 - \eta x_{t,1}^2) + 4\widehat{x}_{t+1,1}\left(\eta x_{t,1} - 8\eta\widehat{x}_{t,1}^3 + 16\eta^2\widehat{x}_{t,1}^4\right) \\
&\leq -\frac{17}{2}\eta\widehat{x}_{t,1}^2 + 4\widehat{x}_{t+1,1}\left(2\eta\widehat{x}_{t,1} - 8\eta\widehat{x}_{t,1}^3 + 16\eta^2\widehat{x}_{t,1}^4\right) && \text{(Eq. (26))} \\
&\leq -\frac{17}{2}\eta\widehat{x}_{t,1}^2 + 4\widehat{x}_{t+1,1}\left(2\eta\widehat{x}_{t,1} + 16\eta^2\widehat{x}_{t,1}^4\right) \\
&\leq -\frac{17}{2}\eta\widehat{x}_{t,1}^2 + 4(\widehat{x}_{t,1} - 4\eta\widehat{x}_{t,1}^2)\left(2\eta\widehat{x}_{t,1} + 16\eta^2\widehat{x}_{t,1}^4\right) && (\widehat{x}_{t+1,1} < \widehat{x}_{t,1} - 4\eta\widehat{x}_{t,1}^2) \\
&= -\frac{1}{2}\eta\widehat{x}_{t,1}^2 + 64\eta^2\widehat{x}_{t,1}^5 - 32\eta^2\widehat{x}_{t,1}^3 - 256\eta^3\widehat{x}_{t,1}^6 \\
&\leq -\frac{1}{2}\eta\widehat{x}_{t,1}^2 - 16\eta^2\widehat{x}_{t,1}^3 - 256\eta^3\widehat{x}_{t,1}^6 && (\widehat{x}_{t,1} \leq \frac{1}{2}) \\
&\leq 0,
\end{aligned}$$

which leads to contradiction. Similarly, if $\widehat{x}_{t+1,1} > \widehat{x}_{t,1} + 4\eta\widehat{x}_{t,1}^2$, we have

$$
\begin{aligned}
\nabla g(\widehat{x}_{t+1,1}) &= 2(\widehat{x}_{t+1,1} - \widehat{x}_{t,1} - \eta x_{t,1}^2) + 4\widehat{x}_{t+1,1}\left(\widehat{x}_{t+1,1}^2 + \eta x_{t,1} - \widehat{x}_{t,1}^2\right) \\
&> 2(4\eta\widehat{x}_{t,1}^2 - \eta x_{t,1}^2) + 4\widehat{x}_{t+1,1}\left(\eta x_{t,1} + 8\eta\widehat{x}_{t,1}^3 + 16\eta^2\widehat{x}_{t,1}^4\right) \\
&\geq 0. && \text{(Eq. (26))}
\end{aligned}
$$

The calculations above conclude that

$$
\widehat{x}_{t+1,1} \in \left[\widehat{x}_{t,1} - 4\eta\widehat{x}_{t,1}^2, \widehat{x}_{t,1} + 4\eta\widehat{x}_{t,1}^2\right]. \tag{31}
$$

**Induction to get Eq. (26).** Similarly, we have

$$
x_{t+1,1} = \operatorname*{argmin}_{u\in[0,\frac{1}{2}]}\left\{\left(\widehat{x}_{t+1,1} + \eta x_{t,1}^2 - u\right)^2 + \left(\widehat{x}_{t+1,1}^2 - \eta x_{t,1} - u^2\right)^2\right\} \triangleq \operatorname*{argmin}_{u\in[0,\frac{1}{2}]} h(u),
$$

$$
\nabla h(u) = 2(u - \widehat{x}_{t+1,1} - \eta x_{t,1}^2) + 4u(u^2 + \eta x_{t,1} - \widehat{x}_{t+1,1}^2),
$$

and $\nabla h(x_{t+1,1}) = 0$. If $x_{t+1,1} < \frac{1}{2}\widehat{x}_{t+1,1}$, we have

$$
\begin{aligned}
\nabla h(x_{t+1,1}) &= 2(x_{t+1,1} - \widehat{x}_{t+1,1} - \eta x_{t,1}^2) + 4x_{t+1,1}\left(x_{t+1,1}^2 + \eta x_{t,1} - \widehat{x}_{t+1,1}^2\right) \\
&< -\widehat{x}_{t+1,1} - 2\eta x_{t,1}^2 - 3x_{t+1,1}\widehat{x}_{t+1,1}^2 + 2\eta\widehat{x}_{t+1,1}x_{t,1} && (x_{t+1,1} < \tfrac{1}{2}\widehat{x}_{t+1,1}) \\
&\leq 0. && (\eta \leq \tfrac{1}{64}, x_{t,1} \leq \tfrac{1}{2})
\end{aligned}
$$

If $x_{t+1,1} > 2\widehat{x}_{t+1,1}$, we also have

$$
\begin{aligned}
\nabla h(x_{t+1,1}) &= 2(x_{t+1,1} - \widehat{x}_{t+1,1} - \eta x_{t,1}^2) + 4x_{t+1,1}\left(x_{t+1,1}^2 + \eta x_{t,1} - \widehat{x}_{t+1,1}^2\right) \\
&> 2\widehat{x}_{t+1,1} - 2\eta x_{t,1}^2 + 24\widehat{x}_{t+1,1}^3 + 8\eta\widehat{x}_{t+1,1}x_{t,1} && (x_{t+1,1} > 2\widehat{x}_{t+1,1}) \\
&\geq 2\widehat{x}_{t+1,1} - 2\eta x_{t,1}^2 + 24\widehat{x}_{t+1,1}^3 + 8\eta(\widehat{x}_{t,1} - 4\eta\widehat{x}_{t,1}^2)x_{t,1} && \text{(Eq. (31))} \\
&\geq 2\widehat{x}_{t+1,1} - 2\eta x_{t,1}^2 + 24\widehat{x}_{t+1,1}^3 + 8\eta(\tfrac{1}{2}x_{t,1} - 4\eta\widehat{x}_{t,1}^2)x_{t,1} && \text{(Eq. (26))} \\
&= 2\widehat{x}_{t+1,1} + 2\eta x_{t,1}^2 + 24\widehat{x}_{t+1,1}^3 - 32\eta^2\widehat{x}_{t,1}^2 x_{t,1} \\
&\geq 2\widehat{x}_{t+1,1} + \frac{1}{4}\eta\widehat{x}_{t,1}^2 + 24\widehat{x}_{t+1,1}^3 - 32\eta^2\widehat{x}_{t,1}^2 x_{t,1} && \text{(Eq. (26))} \\
&\geq 0. && (\eta \leq \tfrac{1}{64}, x_{t,1} \leq \tfrac{1}{2})
\end{aligned}
$$

Both lead to contradiction. Therefore, we conclude that $x_{t+1} \in [\frac{1}{2}\widehat{x}_{t+1,1}, 2\widehat{x}_{t+1,1}]$, which finishes the induction proof.

**Claim 4.** $x_{t,1} \geq \widehat{x}_{t,1} - 4\eta\widehat{x}_{t,1}^2$, for all $t \geq 1$.

The case $t = 1$ holds trivially. For $t \geq 2$, we prove this by contradiction. Using the definition of the projection function, we have:

$$
x_{t+1,1} = \operatorname*{argmin}_{u\in\left[0,\frac{1}{2}\right]}\left\{\left(\widehat{x}_{t+1,1} + \eta x_{t,1}^2 - u\right)^2 + \left(\widehat{x}_{t+1,1}^2 - \eta x_{t,1} - u^2\right)^2\right\} \triangleq \operatorname*{argmin}_{u\in\left[0,\frac{1}{2}\right]} h(u).
$$

Similar to the analysis in **Claim 3**, we have $\operatorname{argmin}_{u\in\left[0,\frac{1}{2}\right]} h(u) = \operatorname{argmin}_{u\in\mathbb{R}} h(u)$, which means that $\nabla h(x_{t+1,1}) = 0$. Note that $\eta \leq \frac{1}{64}$ and $0 \leq \widehat{x}_{t,1} \leq \frac{1}{2}$, according to Eq. (26) and Eq. (27), we have

$$
\widehat{x}_{t+1,1} \in \left[\widehat{x}_{t,1} - 4\eta\widehat{x}_{t,1}^2, \widehat{x}_{t,1} + 4\eta\widehat{x}_{t,1}^2\right] \subseteq \left[\frac{31}{32}\widehat{x}_{t,1}, \frac{33}{32}\widehat{x}_{t,1}\right],
$$

which means that

$$
x_{t,1} \in \left[\frac{1}{2}\widehat{x}_{t,1}, 2\widehat{x}_{t,1}\right] \subseteq \left[\frac{16}{33}\widehat{x}_{t+1,1}, \frac{64}{31}\widehat{x}_{t+1,1}\right]. \tag{32}
$$

If $x_{t+1,1} < \widehat{x}_{t+1,1} - 4\eta\widehat{x}_{t+1,1}^2$, we show that $\nabla h(x_{t+1,1}) < 0$. In fact,

$$
\begin{aligned}
\nabla h(x_{t+1,1}) &= 2(x_{t+1,1} - \widehat{x}_{t+1,1} - \eta x_{t,1}^2) + 4x_{t+1,1}\left(x_{t+1,1}^2 + \eta x_{t,1} - \widehat{x}_{t+1,1}^2\right) \\
&< 2(-4\eta\widehat{x}_{t+1,1}^2 - \eta x_{t,1}^2) + 4x_{t+1,1}\left(\eta x_{t,1} - 8\eta\widehat{x}_{t+1,1}^3 + 16\eta^2\widehat{x}_{t+1,1}^4\right) \\
&\le -\frac{42}{5}\eta\widehat{x}_{t+1,1}^2 + 4x_{t+1,1}\left(\frac{64}{31}\eta\widehat{x}_{t+1,1} - 8\eta\widehat{x}_{t+1,1}^3 + 16\eta^2\widehat{x}_{t+1,1}^4\right) && \text{(Eq. (32))} \\
&\le -\frac{42}{5}\eta\widehat{x}_{t+1,1}^2 + 4x_{t+1,1}\left(\frac{64}{31}\eta\widehat{x}_{t+1,1} + 16\eta^2\widehat{x}_{t+1,1}^4\right) \\
&< -\frac{42}{5}\eta\widehat{x}_{t+1,1}^2 + 4(\widehat{x}_{t+1,1} - 4\eta\widehat{x}_{t+1,1}^2)\left(\frac{64}{31}\eta\widehat{x}_{t+1,1} + 16\eta^2\widehat{x}_{t+1,1}^4\right) \\
&\le 64\eta^2\widehat{x}_{t+1,1}^5 - 32\eta^2\widehat{x}_{t+1,1}^3 - 256\eta^3\widehat{x}_{t+1,1}^6 \\
&\le -16\eta^2\widehat{x}_{t,1}^3 - 256\eta^3\widehat{x}_{t,1}^6 && (\widehat{x}_{t,1} \le \tfrac{1}{2}) \\
&\le 0,
\end{aligned}
$$

which leads to contradiction. Therefore, we show that $x_{t,1} \ge \widehat{x}_{t,1} - 4\eta\widehat{x}_{t,1}^2$ for all $t \ge 1$.

**Claim 5.** If $\eta \le \frac{1}{64}$, we have $\|z_t - z^*\| \ge \Omega(1/t)$.

Now we are ready to prove $\|z_t - z^*\| \ge \Omega(1/t)$. First we show $\widehat{x}_{t,1} \ge \frac{1}{2t}$ for all $t \ge 1$ by induction. The case $t = 1$ trivially holds. Suppose that it holds at step $t$. Considering step $t + 1$, we have

$$
\begin{aligned}
\widehat{x}_{t+1,1} &\ge \widehat{x}_{t,1} - 4\eta\widehat{x}_{t,1}^2 && \textbf{(Claim 3)} \\
&\ge \widehat{x}_{t,1} - \frac{1}{16}\widehat{x}_{t,1}^2 && (\eta \le \tfrac{1}{64}) \\
&\ge \frac{1}{2t} - \frac{1}{64t^2} && (\tfrac{1}{2t} \le \widehat{x}_{t,1} \le \tfrac{1}{2}, \text{ and } x - \tfrac{1}{16}x^2 \text{ is increasing when } x \le 8) \\
&\ge \frac{1}{2(t+1)}. && (t \ge 1)
\end{aligned}
$$

Therefore, $\widehat{x}_{t,1} \ge \frac{1}{2t}$, $\forall t \ge 1$. This, by **Claim 4** and the analysis above, shows that

$$
x_{t,1} \ge \widehat{x}_{t,1} - 4\eta\widehat{x}_{t,1}^2 \ge \frac{1}{2(t+1)}.
$$

Note that according to **Claim 1**, $x^* = 0$. Therefore, we have $\|z_t - z^*\| \ge x_{t,1} \ge \frac{1}{2(t+1)}$, which finishes the proof.

**Claim 6.** In this example, SP-MS holds with $\beta = 3$. This can be seen by the following:

$$
\begin{aligned}
\max_{z' \in \mathcal{Z}} \frac{F(z)^\top(z - z')}{\|z - z'\|} &\ge \max_{z' \in \mathcal{Z}} F(z)^\top(z - z') \\
&= \max_{x' \in \mathcal{X}, y' \in \mathcal{Y}} \left\{x^\top G y' - x'^\top G y\right\} \\
&= \max_{x' \in \mathcal{X}, y' \in \mathcal{Y}} \left\{-x_1 y_2' + x_2 y_1' + x_1' y_2 - x_2' y_1\right\} \\
&\ge -x_1 x_2^2 + x_2^2 + y_2^2 - y_2^2 y_1 \\
&&& \text{(picking } y_1' = x_2, y_2' = x_2^2, x_1' = y_2, x_2' = y_2^2) \\
&\ge \frac{1}{2}x_2^2 + \frac{1}{2}y_2^2 && (x_1, y_1 \le \tfrac{1}{2}) \\
&\ge \frac{1}{4}\left(x_1^4 + x_2^4 + y_1^4 + y_2^4\right) && (x_2 \ge \{x_1^2, x_2^2\}, y_2 \ge \{y_1^2, y_2^2\}) \\
&\ge \frac{1}{16}\left(x_1^2 + x_2^2 + y_1^2 + y_2^2\right)^2 && \text{(Cauchy-Schwarz)} \\
&= \frac{1}{16}\|z - z^*\|^4, && (z^* = (0, 0, 0, 0))
\end{aligned}
$$

which implies $\beta = 3$.                                                                                    $\square$

