# OpenReview forum: "Linear Last-iterate Convergence in Constrained Saddle-point Optimization"
_ICLR.cc/2021/Conference — ICLR 2021 Poster_

### Official Review · AnonReviewer4 · 2020-10-19

**Rating:** 6
**Confidence:** 4

**Review:**

In this paper, the authors consider the convergence of optimistic gradient for constrained saddle-point optimization. Saddle-point problems are very popular in the ML community and lead to non-trivial extensions of the usual (projected) gradient methods. Typically, first-order methods for saddle-point problems are based on the Extra-Gradient algorithm; however, this methods implies two oracle calls per iterations which is undesirable in game theoretical contexts. Fortunately, single call variants have been introduced in the literature to overcome this drawback. Although they all resort to the same algorithm/principle when there are no constraints; single call variants are different in the constrained case. Here, the authors consider the Optimistic Gradient variant (sometimes also called Past-Extra Gradient) and show last-iterate convergence rates under more general constrained case than the previous literature.

The paper offers a strict improvement over known results of the literature. It can definitively interesting for specialists and may lead to interesting developments. More precisely, they replace the "locally strongly convex + sufficiently small stepsize" condition of e.g. [Hsieh2019, Th. 2] to "introduced SP-RSI +  near-usual stepsize" and still manage to show linear convergence. While this is interesting, in the present version, the presentation of the paper fails to meet the expectations for a paper that tackles a very specific technical point by not being pedagogical and clear enough. I am rather convinced of the interest of the approach regarding the linear convergence with multiple minimizers but the point 2 and 3 page 8 (global vs local and stepsize) are not completely conving since they may be related to the implicit unit diameter of X.

All in all, while the paper have merits, the lower quality of presentation and discussion of the results on this very technical matter makes me lean towards rejection.

Concerns:
* In the introduction (typically the second paragraph), I feel that the "oracle call difference" between (EG) and (OGDA) is not clear and may confuse an inexperienced reader.
* Since the authors base their analysis on a template inequality (Lemma 1); what would happen for other single call variants such as the reflected gradient (see Chambolle and Pock " A first-order primal-dual algorithm for convex problems withapplications to imaging") or "optimistic" (Daskalakis et al. "Training GANs with optimism")?
* A drawback of the paper is the lack of a conducting thread along the paper. After the first  3 sections, we go from "Matrix games with OMWU" to "general case + RSI w/ OGDA" then back to matrix games  w/ OGDA. If the authors want to point out the condition RSI and their analysis then this should go first and maybe a section for matrix games w/ two subsections 1) OGDA 2) OMWU would be interesting. In addition, would it be possible to derive a similar RSI w/ KL divergences to get similar results for OMWU?
* [SP-RSI 1] seems taylored for matrix games, I have trouble to see when else this could be applied (or when beta can be > 0). Theorem 9 should be more "explained" I guess.
* [SP-RSI 2] looks like a generalization of strong convexity. What bothers me is that i) the fact that F is monotonous should be recalled here otherwise this might be confusing; ii) since we are in the smooth case, this only works if \|z-z*\|<=1 i.e. assumption 1 (note here that the assumptions are defined but not actually mentioned in the results). Would the constants change is \|z-z'\|>1 ?
* Is there a direct link between these conditions and Kurdyka/Lojasiewicz-type rule. This is especially striking since Th.8 and more precisely appendix C share similarities with the Theorem 5 of "On the convergence of the proximal algorithm for nonsmooth functions involving analytic features" by Attouch and Bolte.

Minor comments/typos:
* From what I get, the main result concerns: Saddle-point + Constraints + Non-strongly convex/usual stepsize; this should be more explicit in the introduction/Sec. 2, maybe with a table recalling related results (eg. the unconstrained case).
* (OGDA) and (OMWU) are basically the same algorithm with two different metrics (as stated in Sec. 3), it can thus be troubling to see them opposed in the first two sections without this precision.
* I am not sure that the 1/sqrt(T) average duality gap rate should be put forward in the intro since it may blur the whole message.
* Average-iterate cv: another reason not to do averaging is the lack of guarantee for non-convex losses.
* I find the notation "dist" for "\|x-Pi(x)\|^2" troubling due to the square. Using dist for the same quantity without the square (or renaming dist^2) would be less confusing for me (typically in Lemma 1).
* In "Other notations", in supp(u), I think u is supposed to be in the positive orthant.
* OGDA seems actually referred to as "Past extra-gradient" in [Hsieh2019], "single call" is just the class of variants of EG w/ one call per step.
* In the matrix game at the bottom of p4: is it standard to take entries in [-1,1] or is it just to have L=1? The equilibrium may not be unique without conditions here.
* The top of page 5 is nice to read and explains well the general reasoning without having to specifically look into the tedious proof in appendix. However, in Theorem 3 the result is not so explicit: C_4 could be made explicit since it is not so complicated as I see from the appendix.
* I found the proofs rather hard to follow e.g. the proof of Theorem 5 is broken down several interdependent claims but the lack of text make them hard to digest.

---

> ### Author Response · Authors · 2020-11-15
> **Response to AnnoReviewer4 (Part 2/2)**
>
> **5. Is there a direct link between these conditions and the Kurdyka/Lojasiewicz-type rule. This is especially striking since Th.8 and more precisely appendix C share similarities with Theorem 5 of "On the convergence of the proximal algorithm for nonsmooth functions involving analytic features" by Attouch and Bolte.**
>
> In short, both the Kurdyka-Lojasiewicz (KL) rule and SP-RSI (with $\beta=0$) are conditions that guarantee linear convergence. However, the KL condition is for minimization problems with proximal gradient methods, while SP-RSI is for saddle-point (min-max) problems with the OGDA method.
>
> You might also find the paper [Karimi et al. 2016] relevant. It provides a thorough overview on the relations between several conditions that drives linear convergence in optimization, including the KL condition you mentioned, and the Restricted Secant Inequality (RSI) condition --- where the name of our SP-RSI comes from. Although our SP-RSI condition is for saddle-point (min-max) problems, it shares some similarity with the RSI condition for minimization problems.
>
> (It is not surprising that you find Appendix C familiar because many proofs for linear convergence go through those lines of analysis)
>
> [Karimi et al. 2016] Linear Convergence of Gradient and Proximal-Gradient Methods Under the Polyak-Lojasiewicz Condition
>
>
> **6. Is it standard to take entries in [-1,1] or is it just to have $L=1$? The equilibrium may not be unique without conditions here.**
>
> For matrix games, it is quite standard to assume that all entries are in [-1, 1] (e.g., [Syrgkanis et al. 2015], [Rakhlin and Sridharan, 2013]), since we can always scale the game matrix to ensure this.
> Such scaling clearly does not affect the uniqueness of the equilibrium.
>
>
> [Syrgkanis et al. 2015] Fast Convergence of Regularized Learning in Games
>
> [Rakhlin and Sridharan, 2013] Optimization, Learning, and Games with Predictable Sequences
>
>
> **7. The top of page 5 is nice to read and explains well the general reasoning without having to specifically look into the tedious proof in appendix. However, in Theorem 3 the result is not so explicit: $C_4$ could be made explicit since it is not so complicated as I see from the appendix.**
>
> We use this constant for the conciseness of presentation. $C_4$ is $15\eta^2\epsilon^3 C^2\xi^2/32$, where $\epsilon, C, \xi$ are all problem-dependent constants defined in Definition 2, Lemma 15, and Definition 4, respectively. Due to the page limit, we did not introduce these problem-dependent constants in the main text and had to use a single constant $C_4$ for conciseness.
>
> **Thank you for all other suggestions related to writing clarity and paper organization. They are very helpful for us to improve readability. We will try our best to include them in the final version.**

---

> > ### Comment · AnonReviewer4 · 2020-11-20
> > **Thank you for the response**
> >
> > I have now a better view of the positioning of the authors results with respect to the literature.
> > I think the paper's clarity can be quite straightforwardly improved based on the reviews and authors' response. Hence, I will upgrade my score during the discussion phase.

---

> ### Author Response · Authors · 2020-11-15
> **Response to AnnoReviewer4 (Part 1/2)**
>
> We thank the reviewer for the valuable comments and suggestions. We will answer your individual questions below.
>
> **1. Since the authors base their analysis on a template inequality (Lemma 1); what would happen for other single call variants such as the reflected gradient or "optimistic" (Daskalakis et al. "Training GANs with optimism")?**
>
> Regarding counterparts of Lemma 1 for other single-call extra-gradient algorithms, please refer to [Hsieh et al. 2019, Section B.2] (see their Eq.(B.4), (B.18), (B.26)). One can observe that they are of very similar form. Therefore, we believe that we can get similar results by applying our conditions to these methods (we still need to verify it though). Note that even though the counterparts of Lemma 1 are known, there was no linear convergence result derived before.
>
> (The setting Daskalakis et al. consider is the unconstrained setting, where every single call variant is equivalent to each other.)
>
>
> **2. A drawback of the paper is the lack of a conducting thread along the paper. If the authors want to point out the condition RSI and their analysis then this should go first and maybe a section for matrix games w/ two subsections 1) OGDA 2) OMWU would be interesting. In addition, would it be possible to derive a similar RSI w/ KL divergences to get similar results for OMWU?**
>
> We thank the reviewer for the valuable suggestions on presenting the results. In short, one may derive a similar RSI w/ KL divergences for OMWU, but the results are restricted compared to the results for OGDA since it requires more conditions and assumptions (explained below). Therefore, it is  also difficult to present the RSI conditions first and state the results for OGDA and OMWU afterward. The current flow of the paper starts from the less general OMWU results for matrix games, and then goes on to the more general OGDA results under more general conditions.
>
> Below we briefly explain why a similar SP-RSI condition w/ KL divergences might be more restricted. In our current analysis for OMWU, we need stronger conditions to obtain similar KL divergence decrease. One condition is stated in Lemma 15. It is similar to SP-RSI-1, but it is actually stronger and requires the stronger assumption of unique equilibrium. Moreover, we need to further relate the $\|z^*-z\|_1$ term in Lemma 15 to KL$(z^*,z)$. This requires more efforts developed in Lemma 16-19 and Lemma 2 that are quite specific to matrix games. Our approach also forces us to use the "two-stage" argument to argue linear convergence.
>
> In summary, we did not state these conditions for general functions since the generalization is much less clean compared to the OGDA counterparts. But making the OMWU result more general is indeed one of the key future directions.
>
> **3. [SP-RSI 1] seems taylored for matrix games, I have trouble to see when else this could be applied (or when beta can be $>$ 0). Theorem 9 should be more "explained" I guess.**
>
> Sorry for the confusion. Theorem 9 actually gives an example with $\beta>0$. In that example, we still consider a bilinear game, but the feasible set boundary is quadratically-shaped (so it is not a polytope). In fact, we can prove $\beta=3$ in that example. On the other hand, this example can not be captured by SP-RSI-2 with any $\beta$, showing that SP-RSI-1 is indeed useful. (More generally, piecewise linear functions with curved boundaries like this one should also be able to be captured by SP-RSI-1 with $\beta>0$).
>
> An example of SP-RSI-1 with $\beta=0$ other than matrix games is when $f(x,y)$ is a piecewise linear function for both $x$ and $y$, and when the feasible sets are polytopes. To show this, we can largely reuse the proof of Theorem 5 because the set of equilibrium is also a polytope in this case.
>
> We admit that Theorem 9 was not fully explained because of the space limit. We will provide more discussion in the revised version.
>
> **4. [SP-RSI 2] looks like a generalization of strong convexity. What bothers me is that i) the fact that F is monotonous should be recalled here otherwise this might be confusing; ii) since we are in the smooth case, this only works given assumption 1. Would the constants change is $|z-z'|>1$ ?**
>
> We thank the reviewer for the comment in i) and will mention that in our updated version. For ii), we explained how to handle the case when $|z-z^*|>1$ in Footnote 1 on page 6. Basically, if the diameter (i.e., $\sup_{z_1, z_2\in\mathcal{Z}}\|z_1-z_2\|$) is $D$, then all dependence on $L$ in our bounds becomes $LD$.
>
> We hope this also addresses another earlier comment "...point 2 and 3 on page 8 (global vs local and stepsize) are not completely convincing since they may be related to the implicit unit diameter of X''.

---

### Official Review · AnonReviewer2 · 2020-10-27
**Recommendation to Accept**

**Rating:** 7
**Confidence:** 3

**Review:**

This paper studies Optimistic Gradient Descent Ascent (OGDA) and Optimistic Multiplicative Weights Update (OMWU) for solving minimax problem. For OMWU, it shows that if the equilibrium is unique and the objective is x^Ty, then a constant linear stepsize results in a linear convergence for the last iterate. For OGDA, it shows that, with constant stepsize, the average duality gap converges with slow rate. Moreover, it shows that under an extra condition, the last iterate converges linearly as well.

***
Strength: The paper is solid in theorem and the proof seems correct as far as I checked. It is in a good writing and highly readable. The convergence is interesting and novel in the sense that it shows explicit linear rate with constant stepsize.

***
Concern: My main concern is about the SP-RSI. Considering it has not beed discussed in other papers, it would be better to provide a more general nonlinear objective class (besides bilinear games on polytopes) that satisfies this condition. Moreover, the role of constraint set in this paper is not clear to me. Is the constraint sets X and Y essential to the problem? What's the difficulty in constrained minimax over unconstrained minimax?

***
For future improvement, I suggest to discuss SP-RSI more deeply, since this condition is added to overcome the intermediate difficulty in technical proof. Also the authors can be more clear about the role of constraint sets. If noting special, it'd better not emphasize the setting is constrained.

***
During rebuttal: I thank the toy examples provided by the authors.

---

> ### Author Response · Authors · 2020-11-15
> **Response to AnnoReviewer2 (Part 1/1)**
>
> We thank the reviewer for the valuable comments and suggestions. We will answer your individual questions below.
>
> **1. The role of constraint set in this paper is not clear to me. Is the constraint sets X and Y essential to the problem? What's the difficulty in constrained minimax over unconstrained minimax?**
>
> In the literature, many unconstrained versions of our problems (bilinear games in particular) are already solved. Please see, for example, [Mokhtari et al., 2019] and [Liang and Stokes, 2019].  In their analysis, they explicitly write down the recursive relationship between the iterates of consecutive rounds and solve the recursive relationship. However, in the constrained setting, with the existence of projection, it is not direct to do the same, which forces us to use different analysis. In this sense, the constraint setting is even more challenging.
>
> **2. Considering it has not been discussed in other papers, it would be better to provide a more general nonlinear objective class (besides bilinear games on polytopes) that satisfies this condition. For future improvement, I suggest to discuss SP-RSI more deeply, since this condition is added to overcome the intermediate difficulty in technical proof.**
>
> We thank the reviewer for the valuable suggestion. We provided some examples of nonlinear objectives in Theorems 6,7,and 9. We will add more discussions on the SP-RSI conditions in our final version. Please also refer to our response to AnnoReviewer 4's Question 3 for more details.
>
> [Mokhtari et al., 2019]: A Unified Analysis of Extra-gradient and Optimistic Gradient Methods for Saddle Point Problems: Proximal Point Approach
>
> [Liang and Stokes, 2018]: Interaction Matters: A Note on Non-asymptotic Local Convergence of Generative Adversarial Networks

---

### Official Review · AnonReviewer1 · 2020-10-30
**Good paper, improves our understanding of OMWU, OGDA in zero-sum games**

**Rating:** 7
**Confidence:** 4

**Review:**

This paper studies the performance of optimistic multiplicative weights update (OMWU) and optimistic gradient descent (OGDA) in constrained zero-sum settings and provide linear convergence rate guarantees. For OMWU in bilinear games over the simplex, they show that when the equilibrium is unique, linear last-iterate convergence is achievable with a constant learning rate. In the case of projected OGDA algorithm, they introduce a sufficient condition under which it convergence fast with a constant learning learning rate. They show that bilinear games over any polytope satisfy this condition and OGDA converges exponentially fast even without the unique equilibrium assumption.

This is overall a nice paper that extends and improves our understanding about optimistic versions of OMWU and OGDA especially in constrained bilinear zero-sum games. The paper does a good job at explaining technical improvements over prior results in the area and particularly the works by Daskalakis and Panageas and Hsieh et al.

The experimental section could be slightly improved. For example in the case of OMWU it is seems hard to detect whether the error curve is best fit by an exponential even after the initial slower phase. It would be very interesting to see numerical estimation of the base of these exponents and see how close they match their theoretical bounds. Also the question about OMWU with a continuum of equilibria could be explored experimentally as well. Do experiments support fast convergence in this case?

Overall, this is a nice paper and I recommend acceptance.

Related references:
In terms of fast convergence in bilinear zero-sum games with fixed learning rates
[1] Proves that even with large fixed learning rates the average duality gap of alternating GDA in unconstrained bilinear zero-sum games converges to zero at a rate of O(1/t). [2] proves O(1/sqr{t}) convergence under arbitrarily large learning rates for a variant of GDA (Follow the regularized leader with Euclidean regularizer) in small constrained bilinear zero-sum games, despite divergence of the day-to-day behavior to the boundary.

[3, 4] OMWU is shown to stabilize fast in bilinear constrained zero-sum games in a different sense by arguing exponentially fast shrinking of the volume of sets of initial conditions in the dual/payoff space.


[1] Bailey et al. Finite Regret and Cycles with Fixed StepSize via Alternating Gradient Descent-Ascent. COLT 2020.
[2]  Bailey, Piliouras. Fast and Furious learning in zero-sum games: vanishing regret with non-vanishing step sizes. Advances in Neural Information Processing Systems. 2019.
[3] Cheung, Piliouras. Chaos, Extremism and Optimism: Volume Analysis of Learning in Games. arXiv preprint arXiv:2005.13996 (2020).

---

> ### Author Response · Authors · 2020-11-16
> **AnnoReviewer1 (Part 1/1)**
>
> We thank the reviewer for the valuable comments. We will add the related works you mentioned in our final version. We first answer your question below.
>
> **1. It would be very interesting to see numerical estimation of the base of these exponents and see how close they match their theoretical bounds. Also the question about OMWU with a continuum of equilibria could be explored experimentally as well. Do experiments support fast convergence in this case?**
>
> We thank the reviewer for the valuable suggestions on experiments. Generally speaking, we believe that the uniqueness is only a technical assumption for the proof. We also believe that the constants in our linear convergence rates can be improved.
>
> Regarding the tightness of our theoretical bounds, we use rock paper scissors as an example. In the experiment, OGDA with $\eta=\frac{1}{8}$ and random initialization gives a linear convergence rate $\approx 1.0519^{-t}$ in terms of dist$(z_t,z^*)$. On the other hand, $1/\sqrt{2}$ is a constant  $C$ for rock paper scissors game to satisfy the SP-RSI-1 condition and this leads to a $96\cdot 1.00145^{-t}$ convergence guarantee provided by Theorem 8. We admit that there is a gap between theory and experiments, and it is an interesting future direction to improve the theoretical bounds.
>
> As for the performance of OMWU when there are multiple Nash equilibria. We also did a few experiments in this case. It showed that OMWU with a continuum of equilibria also has linear convergence in these instances. Therefore, it is an interesting open problem to remove the uniqueness assumption for OMWU.
>
> We will do more experiments and try to incorporate the results in our final versions.

---

> > ### Comment · AnonReviewer1 · 2020-11-21
> > **Feedback**
> >
> > Thank you for your thorough response.

---

### Official Review · AnonReviewer3 · 2020-11-06
**Interesting results, however key comparisons with related works are missing, making it difficult to gauge the significance**

**Rating:** 7
**Confidence:** 4

**Review:**

This paper studies optimistic gradient descent ascent (OGDA) and optimistic multiplicative weights update (OMWU) in the constrained convex-concave min-max optimization setting. For OMWU, under the assumption of unique minimum, the authors show linear rate of convergence of bilinear minmax problems over simplices. For OGDA in constrained setting, the authors show linear convergence under some error bound conditions which the authors name as saddle-point restricted secant inequality (SPRCI). Moreover, the authors prove $1/\sqrt{T}$ rate for the average of duality gap for OGDA.

Even though the results of the paper can be significant, the authors fail to mention important related works, therefore it is not clear how the contributions of the paper adds onto what is already known in these related works. Below I will list my concerns:

OMWU:
- For OMWU, the authors state that even though the analysis might look similar to Daskalakis&Panaegas, the new analysis is *very* different. Here, I would like to see more explanations. What are the different tools that the authors use to improve Daskalakis&Panaegas? I do not find it convincing when I see subjective adjectives such as "very different" analysis: I would like to see what exactly the contribution of the analysis on top of Daskalakis&Panaegas. It does not need to be very low-level details, but I would like to see some high level discussion of the novelty of the techniques here.

Looking at the analysis of Thm 3, I see that the authors use some local arguments depending on $T_0$ to get a term $\alpha^{T_0 - t}$ for some $\alpha$. Then the authors make $\alpha^{T_0}$ to the constant. Here, my concern is that a constant exponential in $T_0$ might be too large. For example, how does this rate compare to standard sublinear rate of OMWU? If the constants of the linear rate are very pessimistic, then both in theory and in practice, sublinear rate might be better.

OGDA:
- I think the main reference the authors are missing is FORB by [1]. FORB is known to be equivalent to OGDA in the unconstrained setting. Therefore, it can be seen as a specific version of OGDA in constrained case. How does the constrained OGDA the authors propose in this paper differ from FORB which is already given in constrained setting?

For instance, FORB gets $1/T$ rate [2] for the average of duality gap, whereas this paper gets $1/\sqrt{T}$. Why does the rate degrade in this paper? It is well known for VIs that the average of duality gap has $1/T$ rate as in [2], then one can use convexity to convert this rate to a rate on the averaged iterate. It is worth noting that the rate referred in this paper due to Golowich et al., is on the last iterate, therefore not comparable to this paper. It is known in Golowich et al., and earlier due to [7] that $1/\sqrt{T}$ is essentially tight for last iterate. However, for the averaged duality gap this paper considers this is not the case, and $1/T$ is known to be obtained with averaging ([2] and many others).

- There exist a big literature on error bound conditions [6], which I suspect to be related to SPRSI proposed in this paper. I think the authors need to add the related work on metric subregularity (MS) and compare their results with the algorithms utilizing metric subregularity for linear convergence [3, 4, 5]. Moreover, I think it is necessary to see the relation of SPRSI with metric subregularity. For example, it is well known that MS holds for piecewise linear quadratic functions which include the setting of bilinear games over polytopes that the authors consider. Moreover, MS also holds for strongly convex strongly concave games. I suspect using MS in FORB analysis can directly yield similar linear convergence rates to this paper. Then, what is the advantage of SPRSI and the new constrained OGDA compared to FORB?

Moreover, since the contribution of the paper is on linear convergence, how tight is the rate, and how good the condition SPRSI for detecting structure? For example, what happens when the problem is strongly convex-concave, how is the rate derived in this paper, compare to linear rate of [1] and others on similar algorithms? Similarly, can the authors compute the linear rate explicitly for some toy problems and then compare with the performance in practice to see tightness?

========= after discussion with authors ========
During the discussion phase, the authors addressed my concerns and improved their results. Therefore I increase my score to reflect this.

[1] Malitsky, Yura, and Matthew K. Tam. "A forward-backward splitting method for monotone inclusions without cocoercivity." SIAM Journal on Optimization 30.2 (2020): 1451-1472.

[2] Böhm, Axel, et al. "Two steps at a time--taking GAN training in stride with Tseng's method." arXiv preprint arXiv:2006.09033 (2020).

[3] Latafat, Puya, Nikolaos M. Freris, and Panagiotis Patrinos. "A New Randomized Block-Coordinate Primal-Dual Proximal Algorithm for Distributed Optimization." arXiv preprint arXiv:1706.02882 (2017).

[4] Liang, Jingwei, Jalal Fadili, and Gabriel Peyré. "Convergence rates with inexact non-expansive operators." Mathematical Programming 159.1-2 (2016): 403-434.

[5] Alacaoglu, Ahmet, Olivier Fercoq, and Volkan Cevher. "On the convergence of stochastic primal-dual hybrid gradient." arXiv preprint arXiv:1911.00799 (2019).

[6] Rockafellar, R. Tyrrell, and Roger J-B. Wets. Variational analysis. Vol. 317. Springer Science & Business Media, 2009.

[7] Davis, Damek, and Wotao Yin. "Convergence rate analysis of several splitting schemes." Splitting methods in communication, imaging, science, and engineering. Springer, Cham, 2016. 115-163.

---

> ### Author Response · Authors · 2020-11-13
> **Response to AnnoReviewer3 (Part 3/3)**
>
> We hope our responses in Part 1 and Part 2 address most of your concerns.
>
> To even further answer how the contributions of the paper adds onto what is already known in these related works'', below we provide a more systematic literature summary and clearly point out our contributions again.
>
> **OGDA**
>
> The OGDA algorithm we study dates back to [Popov 1980]. It belongs to a class of algorithms called “single-call extragradient algorithms”, as summarized in [Hsieh et al. 2019]. Hsieh et. al. categorize this class of algorithms into three categories. Specifically, the FORB algorithm you mentioned belongs to “RG”, while the OGDA we study belongs to “PEG”. Although they are different, they are closely related (and become the same in the unconstrained setting), and actually have similar theoretical guarantees. This class of algorithms is important because they can be readily implemented as “no-regret” algorithms, and each player can run the algorithm independently without cooperating with the other player (though there is some caveat in applying RG as a no-regret algorithm, because the point $2x_t-x_{t-1}$ may fall outside the feasible set).
>
> Below, we summarize the known results about single-call extragradient algorithms in two-player zero-sum games, based our best knowledge:
>
> - $O(1/T)$ convergence rate for “average-iterate“ is established by [Rakhlin et al. 2013]
> - Linear convergence rate for “last iterates”:
>     * For the unconstrained setting, linear convergence has been established for bilinear games (e.g. [Liang and Stokes, 2018]) and strongly-convex-strongly-concave functions (e.g. [Mokhtari et al., 2019]).
>     * For the constrained setting, linear convergence has been established for strongly-convex-strongly-concave functions [Malitsky and Tim, 2018],
>
>
> For the constrained setting, linear last-iterate convergence has not been shown for bilinear games on any single-call extragradient algorithm (including FORB). Our work makes significant progress in this specific setting, which is considered extensively in game theory and online learning problems.
>
> We not only show that OGDA exhibits linear last-iterate convergence in bilinear games with polytope feasible sets, but also that it provably does not achieve linear last-iterate convergence if the feasible set is not polytope (surprisingly). This also indicates that the constrained setting is not a direct result from the unconstrained settings, and there are new aspects to look into. Our analysis easily recovers the results for strongly-convex strongly-concave settings. To generalize our results, we identify the more general SPRSI conditions which include both bilinear and strongly-convex-strongly-concave functions as special cases.
>
> **OMWU**
>
> OMWU algorithms are more natural for constrained settings, in particular when the constraints are the probability simplex. We are only aware of the following results for OMWU (all on probability simplex):
>
> - Rakhlin and Sridharan (2013), Syrgkanis et al. (2015) show that OMWU has $O(1/T)$ average-iterate convergence in bilinear games
> - Daskalakis and Panaegas (2019) shows that OMWU has last-iterate convergence in bilinear games.  However,  their analysis 1) requires exponentially small (in some problem-dependent quantity) learning rate 2) does not give an explicit convergence rate, and 3) requires a unique equilibrium.
> - Lei et al. (2020) studies OMWU with general convex-concave functions, but with more assumptions
>
> Our analysis investigates the same setting as [Dakalakis and Panaegas 2019], but resolves their first two issues: our analysis holds for large learning rates, and we show linear convergence with an explicit convergence rate, when the equilibrium is unique.
>
> [Rakhlin et al. 2013] Optimization, Learning, and Games with Predictable Sequences
> [Popov 1980] A modification of the Arrow–Hurwicz method for search of saddle points
> [Dakalakis and Panaegas 2019] Last-Iterate Convergence: Zero-Sum Games and Constrained Min-Max Optimization
> [Hsieh et al. 2019] On the Convergence of Single-Call Stochastic Extra-Gradient Methods
> [Mokhtari et al., 2019] A Unified Analysis of Extra-gradient and Optimistic Gradient Methods for Saddle Point Problems: Proximal Point Approach
> [Lei et al., 2020] Last iterate convergence in no-regret learning: constrained min-max optimization for convex-concave landscapes
> [Liang and Stokes, 2018] Interaction Matters: A Note on Non-asymptotic Local Convergence of Generative Adversarial Networks
> [Malitsky and Tim, 2018] A Forward-Backward Splitting Method for Monotone Inclusions Without Cocoercivity
> [Syrgkanis et al. 2015] Fast Convergence of Regularized Learning in Games

---

> > ### Comment · AnonReviewer3 · 2020-11-18
> > **Discussion**
> >
> > I thank the authors for the detailed explanations.
> >
> > 1. 2. I suggest the authors to include these discussions in the updated version of the paper.
> >
> > 3. It is not clear to me how important the "caveat" mentioned by the authors to compare constrained (OGDA) and FORB algorithms.
> >
> > 4. The authors are right, my remark was not correct here. Indeed $Q_1$ is the max of average but $Q_2$ is the average of max and standard results do not bound $Q_2$. However, it is still not clear to me what the authors say about Golowich et al. I checked Theorem 10 of Golowich et al and they bound the average of the norm of monotone operator $F$ which is used to write the saddle point problem as a VI. I think the authors should clarify this point and show why their averaging of duality gaps is comparable to averaging of operator norms as in Golowich et al. Theorem 10.
> >
> > 5. From what I understand from remarks of the authors, Gilpin et al.'s condition is weaker than SP-RSI. I would also guess that metric subregularity is weaker than SP-RSI (These should be verified by the authors). If this is the case, then I would be more skeptical with SP-RSI and proper comparison and giving proper credit to previous works using metric subregularity is even more important.
> >
> > In particular, the idea of analysis using metric subregularity is to use the negative terms in the RHS of bounds (of the form $-\| x_{k+1} -x_k\|^2$) to obtain a term of the form $-\text{dist}(x_k, X^\star)$, which will give contraction. This also seems similar to the idea of this paper (discussion before Lemma 4) (to be verified or refuted by the authors).
> >
> > I think it would be quite easy (probably already done) to apply metric subregularity to standard extragradient algorithm to obtain linear rate.
> >
> > Another important point related here is that I do not understand the why the authors emphasize the concept of "single call extragradient" methods. It is true that these methods do one call to operator, however, it is well known that they need to use at least half of the step size of EG. For example, EG converges with step size $1/L$ and 2 operator calls whereas Popov or FORB uses step size $1/(2L)$ and one operator calls, therefore the comparison is unclear both in theory and practice. In fact, I know that EG works better in a lot of cases in practice, compared to single-call methods. Of course I am not aware of an extensive practical comparison and I think the authors should compare with standard EG in practice to show the importance of "single call extragradient methods". Can the authors clarify more about the importance of single call methods to address my argument in this paragraph?
> >
> > Moreover, the step size that the authors use is $1/(8L)$ which is 4 times worse than the step size given in Hsieh et al., for Popov's algorithm. I think this is an important point, especially in practice. I understand for OMWU, this step size is quite good compared to previous work, however for OGDA, I am not so sure as both Popov's algorithm and FORB normally work with $1/(2L)$. I know that metric subregularity-based analysis normally work without the need to decrease the step sizes (decrease as in going from 1/(2L) to 1/(8L) as in this paper). If this is a fundamental requirement of this papers' analysis, it might be restrictive.
> >
> > 6. I understand and this is a common issue in metric subregularity based analyses also.
> >
> > I want to emphasize that I recognize the contribution of the paper (especially for OMWU), but the above points are still unclear for me on the side of OGDA/Popov. Quality of the presentation is also concerning for me, similar to AnonReviewer4. I hope the authors can incorporate the changes proposed by me and the other reviewers in their manuscript so it would be easier for us to make a final decision.
> >
> > EDIT: Can the authors also comment on bounded domain assumption? Looking at Lemma 4, this assumption seems quite necessary to me for the analysis. If so, it rules out many important applications for OGDA. For example, metric subregularity (maybe also SP-RSI, though authors should verify this) holds with polyhedral sets, but the authors need to restrict to polytopes due to boundedness assumption. Does the authors think there can be any way to relax boundedness assumption?

---

> > > ### Author Response · Authors · 2020-11-24
> > > **Response to AnnoReviewer3 (Part 2/2)**
> > >
> > > **5. The importance of single call methods**
> > >
> > > EG may have some advantage over single-call methods like OGDA. However, OGDA can find the equilibrium by simply letting two players play the game repeatedly following the game protocol, while EG requires more “coordination” between the players. The decoupled nature of OGDA enables it to work under the case when the other player is arbitrary or adversarial, and still achieve the “no-regret” performance (please see [Chiang et al. 2012, Rakhlin and Sridharan 2013] for its regret bound analysis).
> > >
> > > In the EG method, the “true update” of the learner happens on every second interaction with the opponent.  The opponent can cause high regret to the learner by playing very different strategies on odd and even interactions. This largely limits the application of EG to online learning in adversarial environments. This observation is formalized in [Golowich et al. 2020b, Appendix A.3]. More motivation of having this kind of robustness against adversarial opponents and the limitation of EG can be found in e.g. [Bowling 2005, Syrgkanis et al. 2015, Lei 2020,Golowich et al. 2020b].
> > >
> > > Therefore, single-call methods are more widely applicable and should be given independent attention.
> > > In fact, Gidel et al. 2019 also show that OGDA requires less gradient calls than EG and demonstrate that the former converges faster than the latter empirically (see their Figure 3).
> > >
> > > **6. The smaller stepsize choice of $1/(8L)$ compared with $1/(2L)$ Popov's algorithm and FORB**
> > >
> > > In fact, we can replace the learning rate for OGDA by any real value smaller than $1/(2L)$. This only changes some universal constants in our bounds. See Footnote 3 in the updated version. We still keep the learning rate $1/(8L)$ to be consistent with the one for OMWU, where some technical lemmas indeed require $\eta \le 1/8$.
> > >
> > > **7. The possibility of relaxing the assumption on bounded feasible domain (metric subregularity (maybe also SP-RSI, though authors should verify this) holds with polyhedral sets)**
> > >
> > > About your comments on Lemma 4, in fact we did not use the boundedness assumption in the proof of Lemma 4, so Lemma 4 is unchanged. In the revised version, we remove one of our main assumption (the assumption that $\mathcal{Z}$ is bounded), and slightly change (weaken) Definition 1. With this change, our main convergence theorem (Theorem 8) now does not rely on the boundedness assumption.
> > >
> > > As we have shown the equivalence between MS and our condition SP-MS, and as you pointed out, MS holds for polyhedrons, linear convergence for bilinear games in unbounded polyhedron should also happen in our case. The intuition is also easy to see: notice that with an application of our Lemma 1 on $t=1, 2, \ldots, T-1$ with $z^* = \Pi_{\mathcal{Z}^*}(\hat{z}_1)$, by summing up all $T-1$ inequalities and noting that $\hat{z}_1 = z_0$ by definition, we get
> > >
> > > $$\\|\\hat{z}\_T - \\Pi_{\mathcal{Z}^*}(\hat{z}_1) \\|^2\\leq \\|\hat{z}_1 - \Pi _{\mathcal{Z}*}(\hat{z}_1)\\|^2.$$
> > >
> > > In other words, $\hat{z}_T$ all lie within a bounded region given the initial point $\hat{z}_1$ (for all $T$). Therefore, we can view all updates only happening within a bounded subset of the polyhedron, and we can apply our conclusion for polytopes to argue the linear convergence for unbounded polyhedrons.
> > >
> > > Reference:
> > >
> > > [Tseng 1995] On linear convergence of iterative methods for the variational inequality problem
> > >
> > > [Malitsky 2019] Golden ratio algorithms for variational inequalities
> > >
> > > [Chiang et al. 2012] Online Optimization with Gradual Variations
> > >
> > > [Golowich et al. 2020a] Last Iterate is Slower than Averaged Iterate in Smooth Convex-Concave Saddle Point Problems
> > >
> > > [Golowich et al. 2020b] Tight last-iterate convergence rates for no-regret learning in multi-player games
> > >
> > > [Rakhlin and Sridharan 2013] Optimization, Learning, and Games with Predictable Sequences
> > >
> > > [Hsieh et al. 2019]: On the Convergence of Single-Call Stochastic Extra-Gradient Methods
> > >
> > > [Gidel et al. 2019]: A Variational Inequality Perspective on Generative Adversarial Networks
> > >
> > > [Bowling 2005] Convergence and No-Regret in Multiagent Learning
> > >
> > > [Syrgkanis et al. 2015] Fast Convergence of Regularized Learning in Games
> > >
> > > [Lei et al., 2020]: Last iterate convergence in no-regret learning: constrained min-max optimization for convex-concave landscapes
> > >
> > > [Gilpin et al. 2008] First-Order Algorithm with $O(\ln(1/\epsilon))$ Convergence for -Equilibrium in Two-Person Zero-Sum Games

---

> > > ### Author Response · Authors · 2020-11-24
> > > **Response to AnnoReviewer3 (Part 1/2)**
> > >
> > > Thanks again for the detailed comments. We are glad that you recognize our contribution in the OMWU part, and give many helpful suggestions for the OGDA part.  Below we provide detailed response to your questions in the latest post.
> > >
> > > **1. how important the ''caveat'' mentioned by the authors to compare constrained OGDA and FORB algorithms.**
> > >
> > > The subtle difference between OGDA and FORB may not affect the convergence property of the two algorithms. However, in our paper, we adopt a repeated game protocol as we describe in the second paragraph of Section 1, where the players only access the gradient at the point $(x_t, y_t)$. On the other hand, reflected gradient methods (including the forward-back algorithm) assume access to the gradient at $2x_t - x_{t-1}$, which may fall outside the feasible set (please see Section 3 of [Hsieh et al. 2019]). Therefore, they should be used with care in this setting.
> > >
> > >
> > >
> > > **2. The authors should clarify why their averaging of duality gaps is comparable to averaging of operator norms as in Golowich et al. Theorem 10.**
> > >
> > > To see why the duality gap bound is comparable to the operator norm bound, consider the following example where these two have the same order. Let $G$ be a bilinear game whose equilibrium is bounded away from the boundary of the feasible set (e.g., paper-scissor-rock). The duality gap of $(x_t,y_t)$ would be $\alpha_t = \max_{x’, y’} (x_tGy’ - x’Gy_t)=\max_{z’} F(z_t)^\top (z_t-z’) \leq \|F(z_t)\|$ (assuming the diameter is $1$ WLOG), where the last quantity is exactly the operator norm at time $t$.
> > >
> > > On the other hand, for a large enough $t$ such that $\|z_t-z\| \geq C$ for any $z$ on the boundary (always exists since the equilibrium is away from the boundary), we have
> > > $$\alpha_t = \max_z \left(\frac{ F(z_t)^\top (z_t-z) }{\|z_t-z\|}\times \|z_t-z\|\right) \geq \max_{z\in\text{boundary}} \left(\frac{ F(z_t)^\top (z_t-z) }{\|z_t-z\|}\times \|z_t-z\| \right)\geq \|F(z_t)\| \times C,$$
> > > where the last step is because $\|F(z_t)\|=\max_{z’} \frac{F(z_t)^\top (z_t-z’)}{\|z_t-z’\|}$ and the maximum is always attainable on the boundary since this is a bilinear game. This shows that $\alpha_t$ and $\|F(z_t)\|$ are of the same order.
> > >
> > > **3. From what I understand from remarks of the authors, Gilpin et al.'s condition is weaker than SP-RSI.**
> > >
> > > [Gilpin et al. 2008] only focused on two-player zero-sum matrix games in probability simplex, so besides the fact that their algorithm is different from ours, their result is also less general than ours (our condition with $\beta=0$ includes general polytopes, and $\beta>0$ includes other bilinear games with non-polytope feasible set like the example in Theorem 9).
> > >
> > > **4. I would also guess that metric subregularity is weaker than SP-RSI. Proper comparison and giving proper credit to previous works using metric subregularity. It is not hard to apply metric subregularity to standard extragradient algorithms to obtain linear rates.**
> > >
> > > We thank the reviewer for pointing out metric subregularity and related references. We found that it is indeed closely related to our conditions.
> > > Specifically, for the new form of our condition in the revision, we prove that the case with $\beta=0$ is equivalent to the metric subregularity of an operator defined in terms of the normal cone of the feasible set and the gradient of the objective.
> > > We have added the discussions and the proof of equivalence to the paper (please see Appendix F).
> > >
> > > Our condition is also similar to other error bound conditions in previous works on extragradient algorithms such as [Tseng 1995, Malitsky 2019]. We also included more discussions and citations on this in the revised version. Because error bound conditions and MS are closely related, we agree that MS should indeed lead to linear convergence rates for these algorithms. However, we are still not aware of any previous work that argues linear convergence for constrained single-call algorithms under the MS condition or other conditions comparable to or weaker than ours. We emphasize that the analysis of OGDA is not a straightforward extension of the analysis of EG and could be more challenging, as also observed by a concurrent work of [Golowich et al. 2020b] (e.g., discussions in their Section 1.1).

---

> ### Author Response · Authors · 2020-11-13
> **Response to AnnoReviewer3 (Part 2/3)**
>
> **4. Why does the rate of the average of the duality gap degrade in this paper? It is well known for VIs that the average of the duality gap has $1/T$ rate as in [Böhm, Axel, et al. 2020], then one can use convexity to convert this rate to a rate on the averaged iterate.**
>
> There is some mis-understanding here. The $1/T$ rate due to [Böhm, Axel, et al. 2020] is the "duality gap of the average update", but not "average duality gap".  More precisely, let $\alpha(x,y)$ be the duality gap of $(x,y)$. Then "duality gap of the average update" in [Böhm, Axel, et al. 2020] is
>
> $Q_1 = \alpha(\overline{x}, \overline{y})$, with $\overline{x} = \frac{1}{T} \sum_{t=1}^T x_t$ and $\overline{y} = \frac{1}{T} \sum_{t=1}^T y_t$,
>
> while the "average duality gap" in our paper is
>
> $Q_2 = \frac{1}{T}\sum_{t=1}^T \alpha(x_t, y_t)$.
>
> An upper bound of $Q_1$ does not imply an upper bound of $Q_2$. In fact, $Q_1 \leq Q_2$ by the convexity of $\alpha$.
>
> $Q_2$ is related to last-iterate convergence but $Q_1$ is not. To see this, consider the case when $x_t, y_t$ cycle around the equilibrium point. $Q_1$ can be vanishing because $(\overline{x}, \overline{y})$ can approach the equilibrium. But $\alpha(x_t, y_t)$ remains large for all $t$ so $Q_2$ is not vanishing.
>
> Our bound on $Q_2$ in Theorem 21 is indeed tight. This can be observed by the fact that the same average duality gap bound appears in an intermediate step of [Golowich et al, 2020] towards deriving their tight last-iterate bound (see their proof of Theorem 10).
>
> **5. It is necessary to see the relation of SPRSI with metric subregularity.**
>
> Thank you for pointing this out.  We are new to this term, but based on this keyword, we indeed found some related work that we omitted.
>
> We found that a condition that corresponds to the special case of our SPRSI1 in matrix games is proposed in [Gilpin et al. 2008, Lemma 3]. With this property, they showed that there is a variant of Nesterov’s first-order smoothing method which can find the equilibrium with linear convergence rate. This convergence rate matches ours, and this is also the polytope-constrained setting that we focus on. However, their algorithm does not belong to the single-call extragradient class that we are interested in, and cannot be implemented as a no-regret online learning algorithm.  A follow-up [Mordukhovich et al. 2010] drew further connection between this condition and the metric subregularity. Therefore, metric subregularity is indeed related to our SPRSI1, and we will incorporate related discussion into our paper. Thanks for helping us discover this line of research.
>
> On the other hand, we also want to emphasize that to our best knowledge, prior to our work neither metric subregularity nor the SPRSI1 condition was known to drive linear last-iterate convergence for single-call extragradient algorithms.
>
> **6. How tight is the rate, and how good the condition SPRSI for detecting structure. Compute the linear rate explicitly for some toy problems and then compare with the performance in practice to see tightness.**
>
> For theoretical lower bound of the last-iterate convergence, we are only aware that [Golowich et al., 2020] that shows a $1/\sqrt{T}$ lower bound for general smooth convex-concave functions for the extragradient algorithm in the unconstrained case. But their techniques are not readily transferred to our setting.
>
> Comparing our theory and our experiments, we feel that the theoretical bound we can obtain might be too pessimistic for randomly generated games. For example, some leading constants in OMWU appear to be generally large in theory (e.g., $1/\epsilon$ with $\epsilon$ defined in Definition 4), but in experiments, the performance does not seem to suffer from such large constants and still converge fast.  Please also refer to our response to AnnoReviewer1 for more details.
>
> On the other hand, it is also possible that there exist worst cases such that our bound is unimprovable. Therefore, at this point we are unable to conclude whether our bound is tight or not, but this is definitely an interesting question to investigate.
>
> [Gilpin et al. 2008] First-Order Algorithm with $O(\ln(1/\epsilon))$ Convergence for $\epsilon$-Equilibrium in Two-Person Zero-Sum Games
> [Mordukhovich et al. 2010] Applying metric regularity to compute a condition measure of a smoothing algorithm for matrix games
> [Golowich et al. 2020] Last Iterate is Slower than Averaged Iterate in Smooth Convex-Concave Saddle Point Problems

---

> ### Author Response · Authors · 2020-11-13
> **Response to AnnoReviewer3 (Part 1/3)**
>
> We thank the reviewer for the valuable comments. We will first answer your individual questions.
>
> **1. What are the different tools that the authors use to improve [Daskalakis and Panaegas, 2019]?**
>
> The similarity of our analysis and [Daskalakis and Panaegas, 2019] only lies in that both utilize a “two-stage” argument to show last-iterate convergence (i.e., in the first stage, a slower convergence rate is shown, and in the second stage, a faster convergence rate is shown). However, the proof details and the results are largely different.
>
> Specifically, [Daskalakis and Panaegas, 2019] utilize tools of ”spectral analysis” that are along the same line as [Liang and Stokes, 2018]. To show last-iterate convergence, they show that the OMWU update can be viewed as a “contraction mapping” with respect to a matrix whose eigenvalue is smaller than 1.
>
> Our analysis, on the other hand, leverages analysis of online mirror descent. We start from the “one-step regret bound” (Lemma 1) used in online mirror descent analysis, but make use of the two negative terms that are typically dropped in online learning analysis to show that KL$(z^*,z_{t})$ will sufficiently decrease in each round.
> Importantly, our analysis does not need an exponentially small learning rate required in [Daskalakis and Panaegas, 2019].
>
> **2. A constant exponential in $T_0$ might be too large. For example, how does this rate compare to the standard sublinear rate of OMWU?**
>
> In the paper, we push $T_0$ to the exponent just for simplicity to show our linear convergence. An alternative is to say that there are two stages of convergence, where in the first stage the convergence rate is $O(1/t)$ and this stage lasts for the first $T_0$ steps; and in the second stage the convergence rate is $O(e^{-\lambda t})$ (linear convergence). With this interpretation, the bound does not pay for a constant exponential in $T_0$. To keep the conciseness of the presentation, we use the original expression in the theorem, but we will add more explanations on the above two-stage view.
>
> We also want to stress that in terms of the last-iterate convergence, we are not aware of any “standard sublinear rate” for OMWU.  The results in [Daskalakis and Panaegas, 2019] did not provide an explicit rate. In fact, they listed “finding exact rates of convergence” as an open problem. To the best of our knowledge, we give the first last-iterate convergence rate for OMWU.
> As for the average-iterate, sub-linear rates of convergence with smaller leading constants are indeed well known, but the results are not comparable to ours as we focus on last-iterate convergence.
>
> **3. The main reference the authors are missing is FORB. How does the constrained OGDA the authors propose in this paper differ from FORB which is already given in a constrained setting? I suspect using MS in FORB analysis can directly yield similar linear convergence rates to this paper. Then, what is the advantage of SPRSI and the new constrained OGDA compared to FORB?**
>
> Sorry for missing FORB in our reference list (we will add it later). However, we are aware of this algorithm because it is reconsidered in [Hsieh et al. 2019] with a different name.
>
> We want to first clarify that we did not “propose” constrained OGDA, because this is an existing algorithm dating back to [Popov, 1980] (please see the literature review in Part 3). However, as we will mention in Part 3, while the single-call variants of the Extra-Gradient algorithms including constrained OGDA and constrained FORB are well known algorithms, their behavior regarding last-iterate convergence remains unclear, especially in bilinear games.
>
> Due to the similar updates of constrained OGDA and constrained FORB, we actually believe that the SPRSI condition will also give similar results shown in our paper for FORB. This requires more theoretical verification though. We are also not aware of any previous results that study MS or SPRSI in FORB or other single-call extragradient algorithms.
>
> (Side note: there are advantages of OGDA over FORB when applying them to no-regret learning, as we will point out in Part 3)
>
> [Popov 1980] A modification of the Arrow–Hurwicz method for search of saddle points
> [Dakalakis and Panaegas 2019] Last-Iterate Convergence: Zero-Sum Games and Constrained Min-Max Optimization
> [Hsieh et al. 2019] On the Convergence of Single-Call Stochastic Extra-Gradient Methods
> [Liang and Stokes, 2018] Interaction Matters: A Note on Non-asymptotic Local Convergence of Generative Adversarial Networks

---

### Author Response · Authors · 2020-11-24
**General Response for the Revision**

We thank all reviewers for their valuable comments, from which we learn a lot.  We have made our first revision that addresses most of the major concerns.  One notable change is that we slightly modify the convergence conditions for OGDA (including its name), so that it has a better connection with metric subregularity and other error bound conditions that have been used in the literature to show linear convergence for other algorithms (addressing issues raised by R3 and R4). Note that the new condition is **weaker** than the original one (thus making our results stronger), and it also covers the case of unbounded feasible sets (addressing issues raised by R3 and R4). We have also included more discussions and references related to this condition.

In more detail, our revision incorporates the following:

**Section 1 (Introduction):**
- Added more motivations of studying OGDA or single-call algorithms (R3)

**Section 2 (Related Work):**
- Added a few paragraphs discussing previous works on error bound methods and metric subregularity (R3)
- Included other related references suggested by R3 and R1

**Section 4 (OMWU):**
- Added more comparisons between our techniques and those of [Daskalakis and Panaegas, 2019b] (R1 and R3)
- Explained that the large leading constant in the bound for OMWU is only for simplicity (R3)

**Section 5 (OGDA):**
- Removed the boundedness assumption (R3 and R4)
- Weakened the condition in Definition 1 and renamed it as “SP-MS”
- Discussed more on the relation between Definition 1 and previous works, and pointed out our novelty (R2 and R3)
- Modified the statement of Theorem 8 (using the new SP-MS condition)
- Mentioned that the step size for OGDA can be as large as $1/(2L)$ (R3)
- Added the statement ``SP-MS holds with $\beta=3$ in Theorem 9 (R4)


**Appendix A (Experiment)**
- Added one more experiment subsection that investigates OMWU with non-unique equilibria. It still shows linear convergence. (R1)

**Appendix F**
- Showed the equivalence between our SP-MS with $\beta=0$ and metric subregularity (R3)

**Appendix G and Appendix H**
- Updated the proofs for all examples we present to incorporate the new SP-MS condition  (note that all changes are very minor)

**Appendix I (Proof of Theorem 8)**
- Proved Theorem 8 based on our new SP-MS condition (the idea is the same as before)

**Appendix J (Proof of Theorem 9)**
- Added a proof to show that the example satisfies our new condition with $\beta=3$

---

### Decision · Program_Chairs · 2021-01-07
**Final Decision**

**Decision:**

Accept (Poster)

**Comment:**

The authors propose to provide fast convergence results for the OGDA and OMWU algorithms based on a reinterpretation of the metric subregularity in the saddle point problem setting. During the rebuttal period, the paper improved significantly, not only due to the diligence of the authors but also due to reactive reviewers that provided extremely constructive comments. The technical developments are quite nice: Lemma 2 allows constant step-size parameter as compared to Daskalakis and Panageas, followed by Theorem 3, which establishes the first linear rate under the saddle point metric subregularity. The numerical demonstrations are also helpful in driving the point home. Although it is not surprising that the shape of the polytope matters, it is still impactful to see the linear rate.


ps. The authors should consider including a related work comparison to the reflected FB algorithm in [1] since it reduces to the FoRB and it also provides convergence analysis for the sequence in the general monotone inclusions.

[1] Cevher and Vu, "A reflected forward-backward splitting method for monotone inclusions involving Lipschitzian operators,''
https://arxiv.org/pdf/1908.05912.pdf